# Natural variation of *DROT1* confers drought adaptation in upland rice

Xingming Sun[1,2,7], Haiyan Xiong[1,3,7], Conghui Jiang[1], Dongmei Zhang [4], Zengling Yang [5], Yuanping Huang[5], Wanbin Zhu[1,6], Shuaishuai Ma[1], Junzhi Duan[1], Xin Wang[1], Wei Liu[1,6], Haifeng Guo[1], Gangling Li[1], Jiawei Qi[1], Chaobo Liang[1], Zhanying Zhang[1], Jinjie Li[1], Hongliang Zhang[1], Lujia Han[5], Yihua Zhou [4], Youliang Peng [2] & Zichao Li [1,6 ✉]

Upland rice is a distinct ecotype that grows in aerobic environments and tolerates drought stress. However, the genetic basis of its drought resistance is unclear. Here, using an integrative approach combining a genome-wide association study with analyses of introgression lines and transcriptomic profiles, we identify a gene, *DROUGHT1* (*DROT1*), encoding a COBRA-like protein that confers drought resistance in rice. *DROT1* is specifically expressed in vascular bundles and is directly repressed by ERF3 and activated by ERF71, both drought-responsive transcription factors. DROT1 improves drought resistance by adjusting cell wall structure by increasing cellulose content and maintaining cellulose crystallinity. A C-to-T single-nucleotide variation in the promoter increases *DROT1* expression and drought resistance in upland rice. The potential elite haplotype of *DROT1* in upland rice could originate in wild rice (*O. rufipogon*) and may be beneficial for breeding upland rice varieties.

[1] State Key Laboratory of Agrobiotechnology/ Beijing Key Laboratory of Crop Genetic Improvement, College of Agronomy and Biotechnology, China Agricultural University, 100193 Beijing, China. [2] Ministry of Agriculture Key Laboratory of Pest Monitoring and Green Management, College of Plant Protection, China Agricultural University, 100193 Beijing, China. [3] National Key Laboratory of Crop Genetic Improvement and National Center of Plant Gene Research (Wuhan), College of Life Science and Technology, Huazhong Agricultural University, 430070 Wuhan, China. [4] State Key Laboratory of Plant Genomics, Institute of Genetics and Developmental Biology, The Innovative Academy of Seed Design, Chinese Academy of Sciences, 100101 Beijing, China. [5] College of Engineering, China Agricultural University, 100083 Beijing, China. [6] Sanya Institute of China Agricultural University, 572025 Sanya, Hainan, China. [7] These authors contributed equally: Xingming Sun, Haiyan Xiong. ✉email: lizichao@cau.edu.cn

Drought is a persistent impediment to crop production, especially in the context of global climate change and growing water demand for agriculture[1]. Rice is a staple food for more than half of the world's population, but in China, for example, 70% of agricultural water is devoted to rice production alone[2]. Availability of irrigation water will increasingly become a limiting factor for rice cultivation. Thus, breeding rice cultivars that require less water and have excellent drought resistance is an urgent necessity.

Upland rice was domesticated in rain-fed regions and grows under aerobic soil conditions, whereas lowland rice is grown in paddy fields with its basal section covered by water. There are obvious genetic differences between upland and lowland rice in regard to drought resistance and productivity-related traits[3]. However, the genetic basis of aerobic drought-adaptive traits remains unclear, which greatly limits the potential utilization of gene variations in breeding. Linkage analyses have been performed using multiple drought-related traits to elucidate the genetic mechanism of drought resistance in rice[4–6]. However, cloning genes in such ways is challenging because drought resistance is a complex agronomic trait that is regulated by multiple loci with minor effects[7]. Recently, genome-wide association studies (GWASs) have been used to explore the genetics of drought resistance in maize and rice, and several drought-related genes have been functionally elucidated[8–11]. However, further research is still needed to determine the genetic basis of drought-related trait differentiations between upland and lowland rice.

As the first line of defense against various abiotic and biotic environmental stresses, cell walls play important roles in plant resistance to adversity[12]. Recent studies have revealed that mutations in genes affecting cell wall composition and/or structure can alter resistance to drought or other abiotic stresses in maize and rice[13–16]. Plants can monitor their cell wall architecture under normal growth and changing environments[17]. During normal plant cell growth, cell expansion is controlled by the directional organization of cellulose microfibrils in the cell wall, which is regulated by COBRA proteins[18,19]. In rice, the COBRA family comprises 11 members, among which *BC1* is crucial for cell wall formation; its functional mutation causes brittle culm due to a reduction in cell wall thickness and cellulose content[20]. *OsBC1L4*, a homolog of *BC1*, is also associated with cell wall formation and plant height[21]. Although the COBRA family genes are important for plant cell wall formation, their roles in responding to drought stress, balancing cell growth and resistance to abiotic stress are still unknown.

Drought resistance involves multiple physiological responses and is associated with the expression of numerous genes that are regulated by a complex transcriptional regulatory network[22]. Among the transcription factors responding to drought stress, ERF family members are important regulators that act as transcriptional activators or repressors of GCC box-mediated gene expression[23]. *ERF3* is a transcriptional repressor, and its overexpression can reduce drought tolerance in rice, while overexpression of *ERF3* with a mutated EAR motif has no significant effect on drought tolerance[24,25]. Similarly, overexpression of *OsAP2-39* (closely related to *ERF3*) can also inhibit the drought resistance of transgenic lines[26]. In contrast, several other ERF genes act as positive regulators of drought resistance. For example, *OsLG3* (*OsERF62*) increases drought tolerance by inducing ROS scavenging[27]. *ERF71* improves drought resistance by regulating the expression of genes involved in cell wall lignification[28,29]. However, the *ERF* genes that regulate drought resistance by affecting cell wall structure are still poorly understood.

In this work, we conduct a GWAS on a group of natural accessions composed of both upland and lowland rice and clone a drought resistance gene, *DROT1*, that encodes a COBRA family protein. *DROT1* enhances drought resistance by adjusting cell wall structure and is directly repressed by ERF3 and activated by ERF71, in a mechanism that may help control the balance between plant growth and drought resistance. Additionally, natural variation in the promoter of *DROT1* confers divergent drought resistance on upland and lowland rice, showing potential value for breeding.

## Results

**Mining QTLs for drought resistance in *japonica* rice.** To investigate the genetic diversity of rice drought resistance, we evaluated the drought stress phenotypes of 271 rice germplasms including 59 upland rice and 212 lowland rice accessions under field drought conditions (Supplementary Data 1). The rolling degree and the color of seedling leaves are two important traits that showed obvious differences among rice accessions under drought stress (Supplementary Fig. 1a). We determined drought resistance of each germplasm by evaluating the Leaf Rolling Index (LRI) and Leaf Color Index (LCI) and then calculating the Drought Resistance Index (DRI) based on appropriate weighting of the LRI and LCI (see Methods). Under drought stress, varieties with different DRI values generally exhibited significantly different growth performances, indicating that the DRI can be used to reflect the drought resistance of rice seedlings (Fig. 1a). The DRI values of upland rice accessions were significantly higher than those of lowland rice accessions, which is consistent with the better drought resistance of upland rice (Fig. 1b). The high phenotypic diversity and continuous variation of DRI and LRI indicated that these germplasms contained multiple drought-resistant QTLs (Supplementary Fig. 1b, c). Principal component analysis (PCA) of single-nucleotide polymorphisms (SNPs) in linkage equilibrium showed that temperate *japonica*, tropical *japonica* and sub-tropical *japonica* accessions formed distinct clusters, indicating that the set of 271 germplasm represents a structured population (Supplementary Fig. 1d). Using 2,070,333 SNPs covering the entire rice genome, we performed a GWAS for drought resistance under a Compressed Mixed Linear Model (CMLM), which identified seven loci as significantly associated with DRI (Fig. 1c, Supplementary Fig. 2a, and Supplementary Data 2). High contribution rate of LRI to DRI (89.5%) and the significantly different LRI between upland and lowland rice subgroups indicate that LRI is the main component of DRI. Therefore, we performed a GWAS on LRI and identified seven loci that were significantly associated with LRI (Fig. 1d, Supplementary Fig. 2b, Supplementary Data 2). Among them, *qDR4a*, *qDR4b* and *qDR10b* also contributed to DRI.

To further confirm the drought resistance loci, we simulated water-deficit stress on a set of 88 introgression lines (ILs) with segments from IRAT109 (an upland tropical *japonica* accession) in the background of Yuefu (a lowland temperate *japonica* accession). After these lines were germinated on 15% polyethylene glycol (PEG) 6000 (w/v) for 10 days, the relative shoot lengths of 32 ILs were significantly increased compared with that of Yuefu ($P < 0.001$) (Supplementary Data 3). Among these, IL349, carrying *qDR10b*, and IL10, carrying *qDR4a* and *qDR4b*, showed greater relative shoot length than Yuefu (Fig. 1e, f), which further confirmed that *qDR4a*, *qDR4b*, and *qDR10b* are important in rice drought resistance.

**Identification of candidate genes at the *qDR10b* region using an integrated approach.** To explore the genes underlying *qDR10b*, we performed a local linkage disequilibrium (LD) analysis and inferred the candidate interval around the lead SNP. By comparing the DNA sequences of IL349 and Yuefu in the

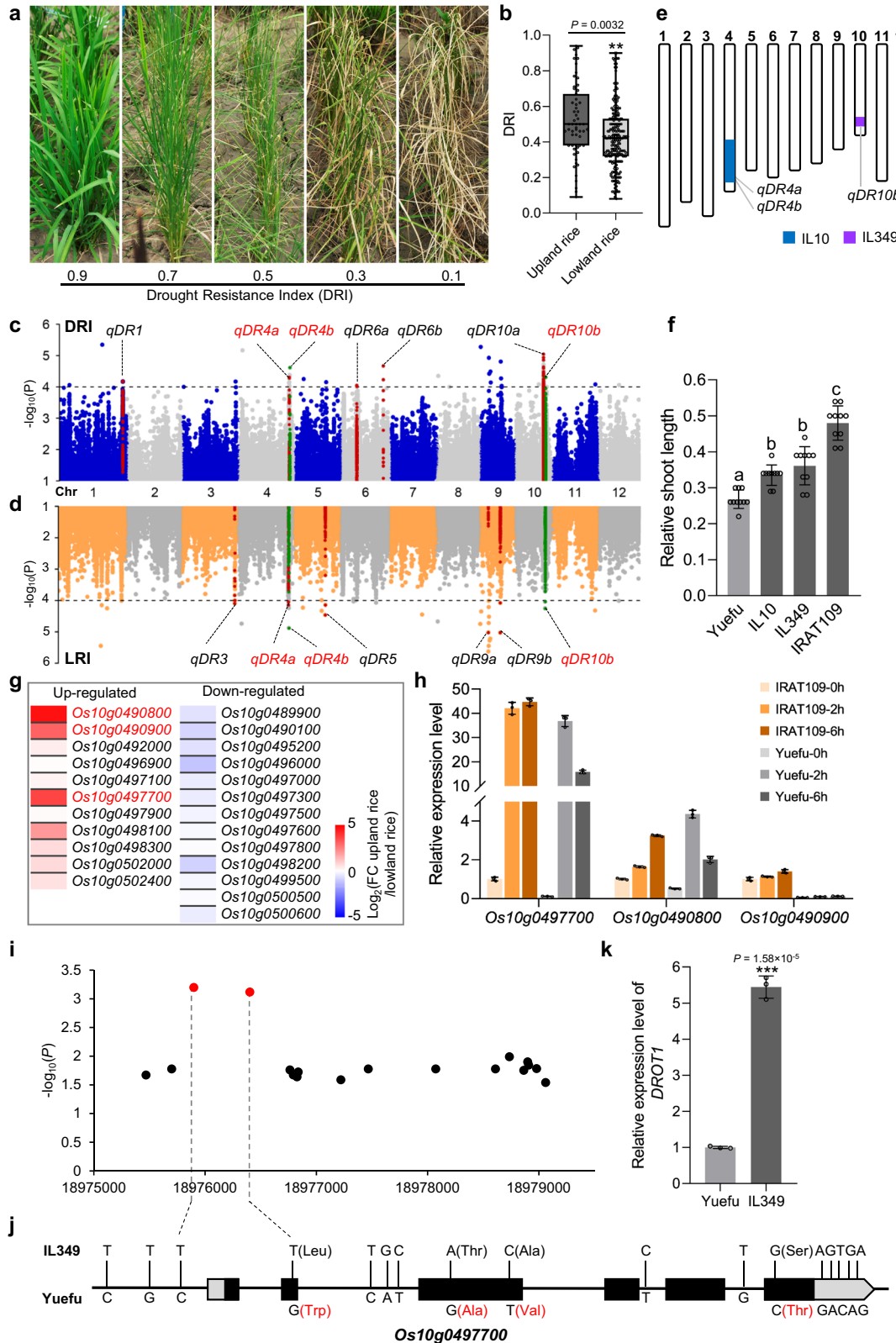

candidate interval of *qDR10b*, 3,326 SNPs were found, of which only 349 were associated with DRI ($P < 0.01$). Ninety-four of these were distributed in the regions of 25 predicted genes. Twenty-two of these 25 genes showed nucleotide differences in the promoter or 3′ UTR, while the other three showed differences in introns (Supplementary Data 4). By comparing the expression levels of the 25 genes in seedling leaves of upland rice (IRAT109

and Haogelao) and lowland rice (Yuefu and Nipponbare)[11], we found that *Os10g0490800*, *Os10g0490900*, and *Os10g0497700* showed significantly higher expression levels in upland rice than in lowland rice (Fig. 1g and Supplementary Data 5). Among them, *Os10g0497700* was strongly induced by dehydration (Fig. 1h). A gene-based association study showed that the two SNPs of *Os10g0497700*, s18975900 in the promoter and

**Fig. 1 Dissecting the drought resistance inheritance and cloning of DROT1. a** Phenotypes of accessions corresponding to different DRI. **b** DRI of upland rice ($n = 59$ accessions) and lowland rice ($n = 212$ accessions) subgroups. In each box plot (drawn by GraphPad Prism 8 software), the center line indicates the median, the edges of the box represent the first and third quartiles, and the whiskers extend to span a 1.5 interquartile range from the edges. **c, d** Manhattan plot of GWAS for DRI (**c**) and LRI (**d**). **e** Chromosome diagram of introgression lines IL349 and IL10. Blank area represents the recipient genome background of Yuefu (lowland rice). Purple rectangle indicates the donor segment of IRAT109 (upland rice) from 18.7 to 19.3 Mb on chromosome 10, and blue rectangle indicates the segment of IRAT109 from 26 to 31 Mb on chromosome 4. Each introgression line contains single segment of IRAT109. **f** Relative shoot length of seedlings treated with 15% PEG6000 for 10 days. Data represent means ± s.d. ($n = 10$ plants). **g** Heat map of 25 up- or down-regulated genes in upland rice compared with lowland rice in the candidate region based on the transcriptome data. FC upland rice/lowland rice represents the average expression value of two upland rice divided by that of two lowland rice. **h** Dehydration stress induced expression of genes in IRAT109 and Yuefu. Two-week-old seedlings were dehydrated under room condition for 0, 2 and 6 h. Data represent means ± s.d. ($n = 3$ biological replicates). **i** Association analysis of genetic variation in Os10g0497700 with DRI. Red dots indicate the most significant SNPs. **j** Allelic variation of Os10g0497700 between IL349 and Yuefu. **k** Expression of DROT1 in IL349 and Yuefu detected in leaf tissues of two-week-old seedlings grown under normal conditions. Data represent means ± s.d ($n = 3$ biological replicates). In **b**, **k**, significant differences were determined by two-tailed Student's t-tests (**$P < 0.01$, ***$P < 0.001$). In **f**, different letters indicate significant differences ($P = 0.01$, one-way ANOVA). Source data are provided as a Source Data file.

---

s18976403 in the second exon, were significantly associated with DRI (Fig. 1i). Sequence comparison of Os10g0497700 between IL349 and Yuefu revealed three SNPs in the promoter region and four nonsynonymous SNPs in the exons (Fig. 1j). The expression level of Os10g0497700 in IL349 was significantly higher than that in Yuefu (Fig. 1k). All the above results indicated that Os10g0497700 is a likely candidate gene for the drought resistance locus qDR10b, and we named it DROUGHT1 (DROT1).

DROT1 encodes a COBRA-like protein. A neighbor-joining tree analysis showed that COBRA-like proteins in plants can be classified into five groups (Supplementary Fig. 3a). Rice BC1 and its homologs in maize (ZmBK2) and sorghum (SbBC1) are grouped in a single clade, and all loss-of-function mutants of these three genes exhibit brittle stems and leaves[30,31], suggesting that the biological functions of COBRA-like proteins are highly conserved among species. Quantitative RT-PCR analysis of the 11 COBRA members in rice showed that the expression of five genes in groups I and II was induced by drought stress, suggesting that COBRA genes may be involved in drought stress response (Supplementary Fig. 3b).

**DROT1 positively regulates drought resistance in rice.** To verify the drought resistance function of DROT1, we knocked out DROT1 in IL349 using a CRISPR/Cas9-based mutagenesis and overexpressed it in Nipponbare with the coding sequence (CDS) of DROT1 amplified from IRAT109 (OEI) and Yuefu (OEY). We then subjected one loss-of-function mutant (drot1-1) with a 143-bp deletion in the third exon of DROT1 and four homozygous overexpression lines to various stress treatments (Supplementary Fig. 4a, b).

First, seedlings of transgenic plants and their controls were exposed to a soil drought treatment in the pot. Compared with IL349, drot1-1 exhibited significantly reduced drought resistance (Fig. 2a, c). Similarly, another knockout mutant with Zhonghua 11 (ZH11) background, drot1-zh11, was less resistant to drought than ZH11 (Supplementary Fig. 4a, c, d). In contrast, the drought resistance of OEI and OEY lines was significantly greater than that of negative transgenic control (NT) plants (Fig. 2b, d). Second, we performed drought treatment under aerobic drought field conditions in a rain-proof shed. After growing under severe drought field conditions without water supply for nearly three months (Supplementary Fig. 5a), the plant height and biomass of drot1-1 were significantly lower than those of IL349; in the paddy field, however, drot1-1 and IL349 had similar growth performance (Fig. 2e and Supplementary Fig. 5b, c). Therefore, the relative plant height and aboveground biomass of drot1-1 were significantly lower than those of IL349 (Fig. 2f, g). In contrast, the plant heights of OEI and OEY plants were significantly higher than that of NT in both paddy and drought field, while the

aboveground biomasses of OEI and OEY were significantly higher in the drought field (Fig. 2h and Supplementary Fig. 5d, e). More importantly, the relative plant height and aboveground biomass of OEI and OEY plants were significantly higher than those of NT (Fig. 2i, j). At maturity, the aboveground biomass of over-expression lines or knockout mutant in the paddy field showed no differences compared to control. In the severe drought field, however, drot1-1 had significantly lower biomass than IL349, while both OEI and OEY plants showed significantly increased biomass compared with NT (Fig. 2k, l). Third, the seedlings of overexpressed transgenic plants and their controls were subjected to simulated drought stress using 20% PEG6000. After recovery, both OEI and OEY plants showed significantly higher survival rates than NT (Supplementary Fig. 6a, b). In addition, the water loss rate of drot1-1 leaves was higher than that of IL349, while both OEI and OEY leaves have a lower water loss rate than NT leaves (Supplementary Fig. 7a, b). Based on these results, we speculate that DROT1 positively regulates drought resistance, and the CDSs of DROT1^{IRAT109} and DROT1^{Yuefu} may have similar drought resistance functions.

To test whether the expression level of DROT1 affects drought resistance, we generated transgenic lines with RNA interference (RNAi) of DROT1. The expression of DROT1 in RNAi transgenic plants was significantly lower than that in NT (Supplementary Fig. 8a). After treatment with 20% PEG6000 for 5 days followed by re-watering for 10 days, the survival rate of RNAi lines was significantly lower than that of NT (Supplementary Fig. 8b, c). This further suggests that the expression level of DROT1 is important for drought resistance.

To investigate whether DROT1 contributes to grain yield and yield-related traits under moderate drought conditions, different transgenic lines were planted in a rain-proof shed and watered eight times throughout their life cycle (Supplementary Fig. 9a–c). The grain yield per plant of drot1-1 was significantly lower than that of IL349 in the moderate drought field, but no difference in the paddy field (Fig. 2m). Furthermore, the grain yield per hectare of drot1-1 was significantly lower than that of IL349 in the moderate drought field (Fig. 2n). Both OEI and OEY lines had a significantly higher grain yield per plant than NT in the paddy field, but only OEI had a higher grain yield per plant than NT in the moderate drought field (Supplementary Fig. 9d). The grain yield per hectare of OEI and OEY did not differ from that of NT in the moderate drought field (Supplementary Fig. 9e). However, the plot biomasses of OEI and OEY were significantly higher than that of NT, and the plot biomass of drot1-1 was significantly lower than that of IL349 under moderate drought (Supplementary Fig. 9f, g). These results indicate that DROT1 can maintain rice yield or yield-related traits under moderate drought conditions.

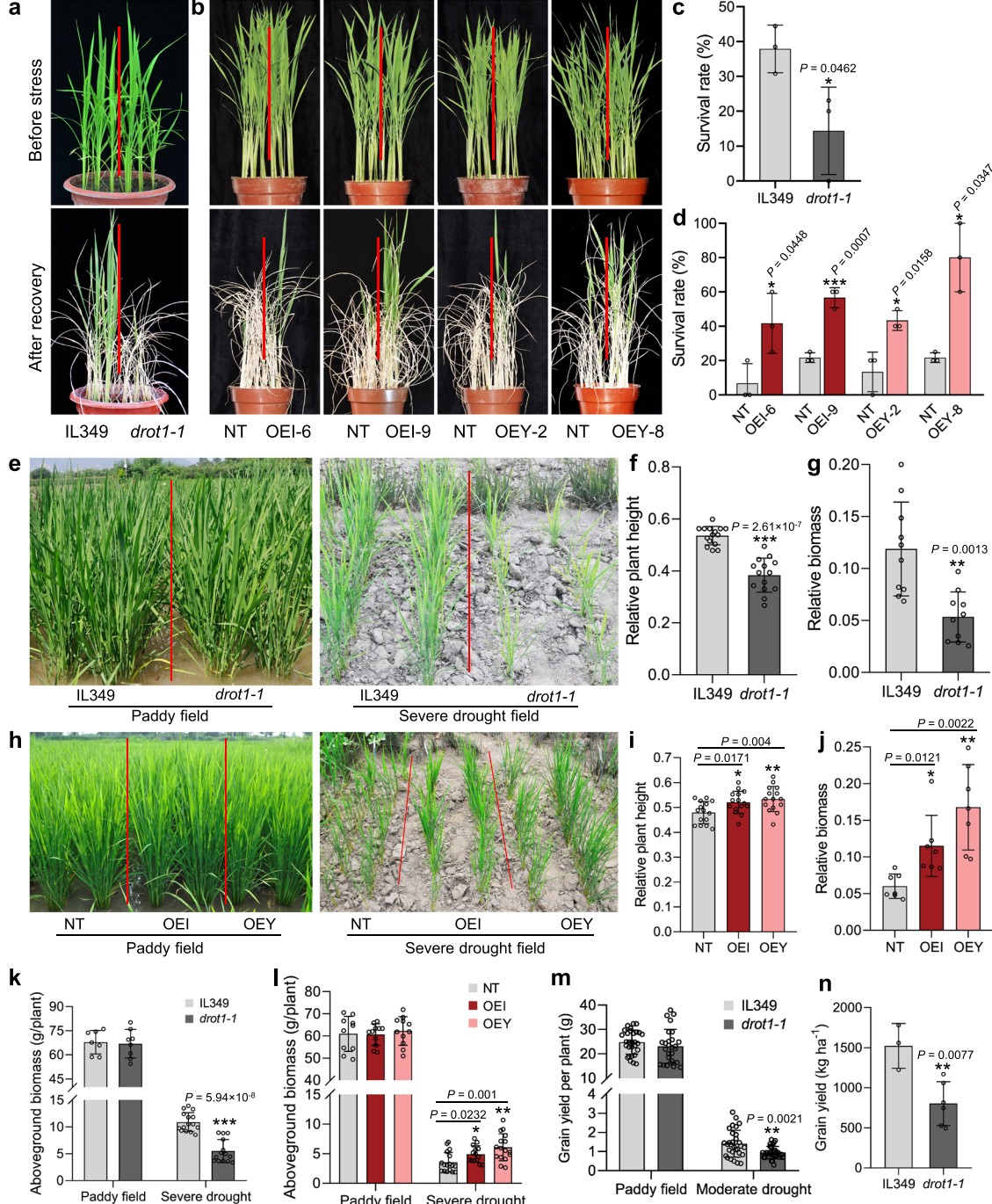

**Fig. 2 Functional validation of *DROT1* in drought resistance. a** Drought resistance of *drot1-1* compared to IL349. **b** Drought resistance of *DROT1*-overexpressing transgenic lines compared to NT (negative transgenic control). The seedlings grown for 4 weeks under normal conditions (top in **a**, **b**) were treated by drought stress for 15 days, followed by re-watering for 10 days (bottom in **a**, **b**). **c** Survival rates of IL349 and *drot1-1* seedlings after re-watering. Data represent means ± s.d. (*n* = 3 biological replicates). **d** Survival rates of NT, OEI and OEY seedlings after re-watering. Data represent means ± s.d. (*n* = 3 biological replicates). **e** Growth performance of *drot1-1* and IL349 grown in paddy field and severe drought field for 90 days. **f** Relative plant height of *drot1-1* and IL349. Data represent means ± s.d. (*n* = 15/14 plants). **g** Relative aboveground biomass of *drot1-1* and IL349. Data represent means ± s.d. (*n* = 10 plants). **h** Growth performance of NT and OE lines grown in paddy field and drought field for 90 days. **i** Relative plant height of NT, OEI and OEY. Data represent means ± s.d. (*n* = 15 plants). The values of relative plant height (**f**, **i**) were calculated by plant height in severe drought field divided by those in paddy field. **j** Relative aboveground biomass of NT, OEI and OEY. Data represent means ± s.d. (*n* = 7 plants). The values of relative biomass (**g**, **j**) were calculated by plant biomass in severe drought field divided by those in paddy field. **k** Aboveground biomass of *drot1-1* and IL349 at mature stage. Data represent means ± s.d. (*n* = 7/8, 15/14 plants). **l** Aboveground biomass of NT, OEI and OEY at mature stage. Data represent means ± s.d. (*n* = 10/11/11, 17/14/16 plants). **m** Grain yield per plant of *drot1-1* and IL349 under paddy field and moderate drought field. Data represent means ± s.d. (*n* = 29/27, 30/30 plants). **n** Grain yield per hectare of *drot1-1* and IL349 under moderate drought field. Data represent means ± s.d. (*n* = 3/6 plots). Asterisks indicate statistical significance by two-tailed Student's *t*-tests (**P* < 0.05, ***P* < 0.01, ****P* < 0.001). Source data are provided as a Source Data file.

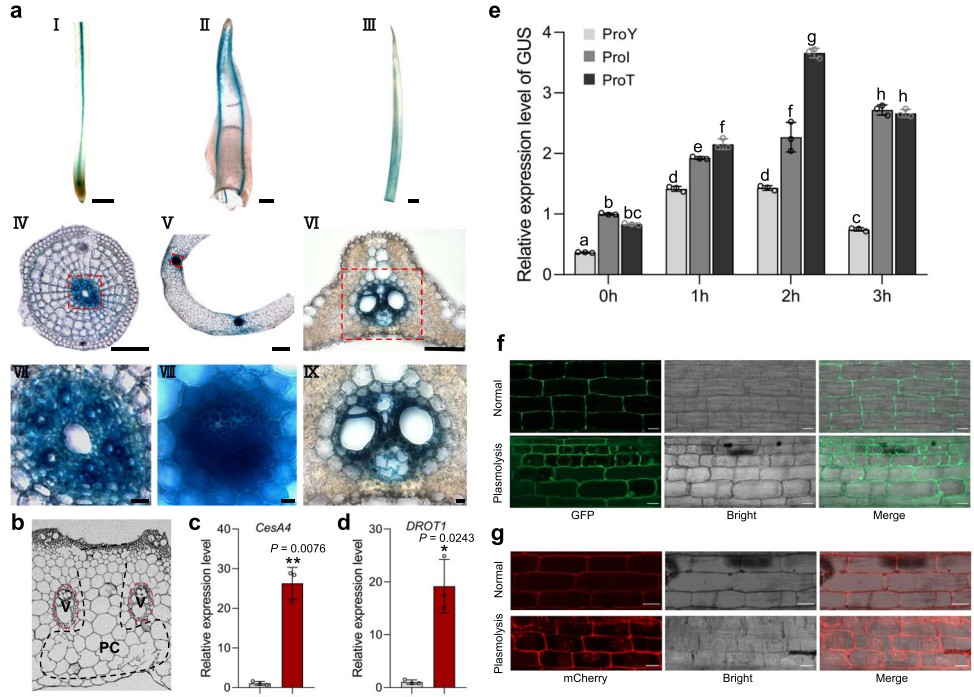

**Fig. 3 Expression patterns of *DROT1* and its subcellular localization. a** GUS staining. (I) Root. (II) Coleoptile. (III) leaf of seedling. (IV, V, and VI) Cross-sections of I, II and III separately. (VII, VIII, and IX) Magnification of the region framed by red box in IV, V, and VI, respectively. Scale bar, 500 μm in I-III, 200 μm in IV-VI, 20 μm in VII-IX. $n = 5$ biological replicates. **b** Schematic diagram of parenchyma cells (PC, indicated by black dashed line) and vascular bundles (V, indicated by red dashed line) in the cross-sections of rice internodes for microdissection. **c, d** qRT-PCR analysis of the vascular bundle-specific marker gene *CesA4* (**c**) and *DROT1* (**d**) in harvested cells or tissues as indicated in **b**. Data are means ± s.d. ($n = 3$ biological replicates). Asterisks indicate statistical significance by two-tailed Student's *t*-tests (\**P* < 0.05, \*\**P* < 0.01). **e** Expression of GUS gene promoted by different types of *DROT1* *cis*-elements. Three vectors containing GUS reporter gene driven by the promoter ProY (the promoter of *DROT1* from Yuefu), ProI (the promoter of *DROT1* from IRAT109) or ProT (the promoter of *DROT1* from Yuefu with the SNP s18975900 changed from C to T) were introduced into Nipponbare, respectively. Data are means ± s.d. ($n = 3$ biological replicates). Different letters indicate statistically significant differences at $P = 0.01$ by one-way ANOVA. **f, g** Subcellular localization of DROT1. Seedlings root tissues of transgenic rice containing vector proDROT1::DROT1-GFP (**f**) or proDROT1::DROT1-mCherry (**g**) were observed by confocal laser-scanning microscopy, respectively. $n = 3$ independent experiments. The plasmolysis in the cells occurred by treated root tissues with 1 M sorbitol for 15 min. Bars = 20 μm. Source data are provided as a Source Data file.

**DROT1 is specifically expressed in vascular bundles and induced by abiotic stress.** *DROT1* is widely expressed in various organs at the seedling and reproductive stages, and IRAT109 had higher expressions than Nipponbare (Nipponbare shares the same DNA sequence of *DROT1* with that of Yuefu) (Supplementary Fig. 10a). To visualize the spatial expression pattern of *DROT1*, we performed GUS staining on transgenic plants containing a *proDROT1*[IRAT109]::GUS construct, finding that *DROT1* was specifically expressed in the vascular bundles of the root, shoot, and leaf (Fig. 3a). To further confirm the tissue-specific expression of *DROT1*, we collected vascular bundles and parenchyma cells separately from young internodes by laser capture microdissection (LCM) (Fig. 3b). Expression analysis of *CesA4*, a vascular bundle-specific marker gene, confirmed the reliability of the data (Fig. 3c). The expression of *DROT1* in vascular bundles was significantly higher than that in parenchyma cells, confirming the specific expression of *DROT1* in vascular bundles (Fig. 3d).

PEG stress induced *DROT1* expression in both IRAT109 and Nipponbare, but the induction was much higher in IRAT109 than in Nipponbare (Supplementary Fig. 10b). To explore *DROT1* expression levels over short or long-term drought stress, seedlings of IRAT109 and Yuefu were subjected to dehydration and drought stress in pots. Under dehydration, *DROT1* was induced instantly in both IRAT109 and Yuefu, and its expression was much higher in IRAT109 (Supplementary Fig. 10c). Under drought stress, the expression of *DROT1* increased rapidly when soil moisture declined to a severe drought level, and IRAT109 had

higher expression levels than Yuefu (Supplementary Fig. 10d, e). These results suggest that differently induced expression of *DROT1* is responsible for the differentiation of drought resistance between upland and lowland rice.

Although there are three SNPs between IL349 and Yuefu in the *DROT1* promoter, only s18975900 was significantly associated with drought resistance. To determine whether s18975900 is responsible for the differentiated expression of *DROT1* between upland and lowland rice, we transformed three constructs with the GUS gene driven by different promoters- ProY, ProI, or ProT into Nipponbare. We found that GUS expression was induced by dehydration stress in all three transgenic lines, but the ProI transgenic plants had higher GUS expression than the ProY plants, suggesting that the *DROT1*[IRAT109] promoter has stronger transcription activity than the *DROT1*[Yuefu] promoter. Moreover, GUS expression was also higher in ProT than that in ProY (Fig. 3e), indicating that converting the single-nucleotide of s18975900 in the *DROT1* promoter from C to T significantly enhanced its transcription activity.

**DROT1 enhances drought resistance by affecting cell wall properties.** To examine the subcellular localization of DROT1, we fused the CDS of *DROT1* from IRAT109 with GFP and mCherry reporter gene under the native *DROT1* promoter and transformed into Nipponbare. Transgenic rice roots were treated with 1 M sorbitol to separate cell walls and protoplasts, and were

visualized with a fluorescence microscope. The results showed that DROT1 protein is primarily located on the periphery of the cell, especially on the cell wall (Fig. 3f, g).

To verify whether *DROT1* affects cell wall composition under normal and/or drought conditions, we examined the cellulose, hemi-cellulose, and lignin contents in leaf tissues of different transgenic lines. OEI and OEY plants showed significantly higher cellulose contents than NT under the drought condition, while there was no difference between these lines under the normal growth condition; in contrast, the cellulose content of *drot1-1* was significantly lower than that of IL349 under both drought and normal conditions (Fig. 4a and Supplementary Fig. 11a). Both OEI lines and knockout lines had increased hemi-cellulose compared with the control lines under drought conditions (Fig. 4b). The lignin content was increased only in OEY lines under drought conditions (Fig. 4c). Under normal conditions, hemi-cellulose and lignin content were not changed in both OE and knockout lines compared to controls (Supplementary Fig. 11b, c). These results suggest that *DROT1* may specifically promote cellulose synthesis under drought stress, but is not essential for the synthesis of hemi-cellulose and lignin.

Since *DROT1* is mainly expressed in vascular bundles, which are the infrastructure for water retention and transportation, its structural changes may have a great impact on drought resistance. To verify whether *DROT1* is involved in cell wall formation in vascular bundles, we conducted in situ Fourier transform infrared (FTIR) microscopy imaging analysis to semi-quantify the cell wall composition of specific tissues in the leaves of *DROT1* transgenic and control plants (Fig. 4d). The results showed that the spectral signals were mainly concentrated in the vascular bundle and sclerenchyma of the lateral veins, and the differences in cell wall components between transgenic and control plants could be distinguished by subtle differences in intensity of the spectral signal (Fig. 4e and Supplementary Fig. 11d). Under the drought condition, the whole lateral vein of OEI and OEY plants showed significantly higher cellulose and hemi-cellulose contents than that of NT, and OEY lines also had a higher lignin content than NT (Fig. 4f). However, IL349 and *drot1-1* had similar contents of all three compounds in the whole lateral veins (Fig. 4g). Through separate semi-quantitative analyses of the cell wall components in the vascular bundles by fast non-negativity-constrained least-squares (NNLS) fitting, we found that OEI and OEY plants had significantly higher cellulose and lignin contents than NT (Fig. 4h); compared with IL349, *drot1-1* had a significantly lower cellulose content in the vascular bundles under the drought condition (Fig. 4i). Under the paddy field, the contents of three cell wall components in the whole lateral veins and vascular bundles were similar in the OEI, OEY, and NT plants (Supplementary Fig. 11e, f). Similarly, the abundances of cell wall components did not differ significantly between *drot1-1* and IL349 under the paddy field condition (Supplementary Fig. 11g, h). Combining the results obtained from the two different quantitative methods, we found that only the cellulose content is changed consistently with an increase in OE lines and decrease in knockout lines under drought conditions. Therefore, we speculate that *DROT1* can affect the cellulose content in vascular bundles under drought stress, thereby regulating drought resistance.

Interestingly, the cellulose content is increased in OE lines under drought, but not under normal conditions. This may have been caused by an increase in individual glucan chains under the drought condition, supplying more materials for the assembly of cellulose microfibrils, and thereby improving the organization and increasing the cellulose content in the cell wall with the participation of excess DROT1 proteins. This is further supported by our findings that drought induces the expression of genes *CesA4*, *CesA7*, and *CesA9*, which encode members of the cellulose

synthesis complex (CSC) for the synthesis of glucan chains (Supplementary Fig. 12a–c)[32].

In addition to vascular bundles, sclerenchyma tissues also play important roles in preventing apoplast water movement and cortex collapse under drought stress[33]. Through transmission electron microscope (TEM) analysis of 45-day-old leaves, we found that the cell wall thickness of sclerenchyma cells in both OEI and OEY plants was significantly greater than that of NT (Supplementary Fig. 13a–g). Scanning electron microscopy (SEM) of sclerenchyma cells in the mature stem revealed that the cell wall thickness of sclerenchyma was greater in OEI and OEY lines than that in NT (Supplementary Fig. 13h–k).

The assembly of cellulose, especially the degree of crystallinity, is critical to the structure of cell walls. To determine whether DROT1 regulates the crystallinity of cellulose microfibrils, we measured the relative crystallinity index (RCI) of cellulose in leaf samples using X-ray diffraction (XRD). As expected, the RCI values of cellulose was generally lower under drought condition than those under normal growth condition. The RCI values of OEI and OEY plants were significantly higher than that of NT under paddy or drought field (Fig. 4j). In contrast, the RCI value of *drot1-1* was significantly lower than that of IL349 under drought field despite *drot1-1* having a similar RCI value to IL349 under the normal condition (Fig. 4k). This demonstrates that DROT1 may contribute to maintaining high-cellulose crystallinity under drought stress. Therefore, we speculate that DROT1 may affect drought resistance by regulating the cellulose content of vascular bundles and maintaining cell wall structure.

**DROT1 is directly regulated by ERF3 and ERF71.** *DROT1* is strongly induced by drought stress, indicating that its promoter may have *cis*-elements involved in the drought stress response. We found three GCC box motifs (the binding sites of AP2/ERF transcription factors) in the promoter of *DROT1*, one of which contains the functional SNP s18975900. To identify upstream regulators of *DROT1*, both transcription repressor and activator of *ERF* family were studied. *ERF3* is a transcription repressor that negatively regulates drought resistance in rice[24], this is further confirmed as the survival rates of *ERF3* overexpression lines were significantly lower than those of NT under drought stress (Supplementary Fig. 14a–c). We found that the expression of *DROT1* was significantly down-regulated in the *ERF3*-overexpressing lines (Fig. 5a, b). *ERF71* is a transcription activator that positively regulates drought resistance[28]. The expression of *DROT1* was upregulated in *ERF71*-overexpressing transgenic lines (Fig. 5c, d). Further analysis of isolated tissues revealed that the expression of both *ERF3* and *ERF71* was higher in vascular bundles than in parenchyma tissues (Fig. 5e, f). These results suggest that *ERF3* and *ERF71* may regulate drought resistance by manipulating the expression of *DROT1*.

To further confirm the transcriptional regulation on *DROT1* by ERF3 and ERF71, we performed transient transcriptional activation experiments in rice protoplast. The results indicated that ERF3 could inhibit the transcription activity of the *DROT1* promoter from both IRAT109 and Yuefu. Conversely, ERF71 could activate the promoter of *DROT1* and did so more strongly for *DROT1*[IRAT109] than for *DROT1*[Yuefu] promoters (Fig. 5g–i). Yeast one-hybrid (Y1H) assay also demonstrated that both ERF3 and ERF71 can directly bound to the *DROT1*[IRAT109] and *DROT1*[Yuefu] promoters (Fig. 5j). Chromatin immunoprecipitation (ChIP) analysis revealed that ERF3 could significantly enrich two of the three GCC box-containing fragments (F2 and F3), and ERF71 could promote enrichment of all three fragments in the *DROT1* promoter (Fig. 5k–m). Electrophoretic mobility-shift assay (EMSA) further demonstrated that both ERF3 and ERF71 could bind to the three GCC box-containing fragments

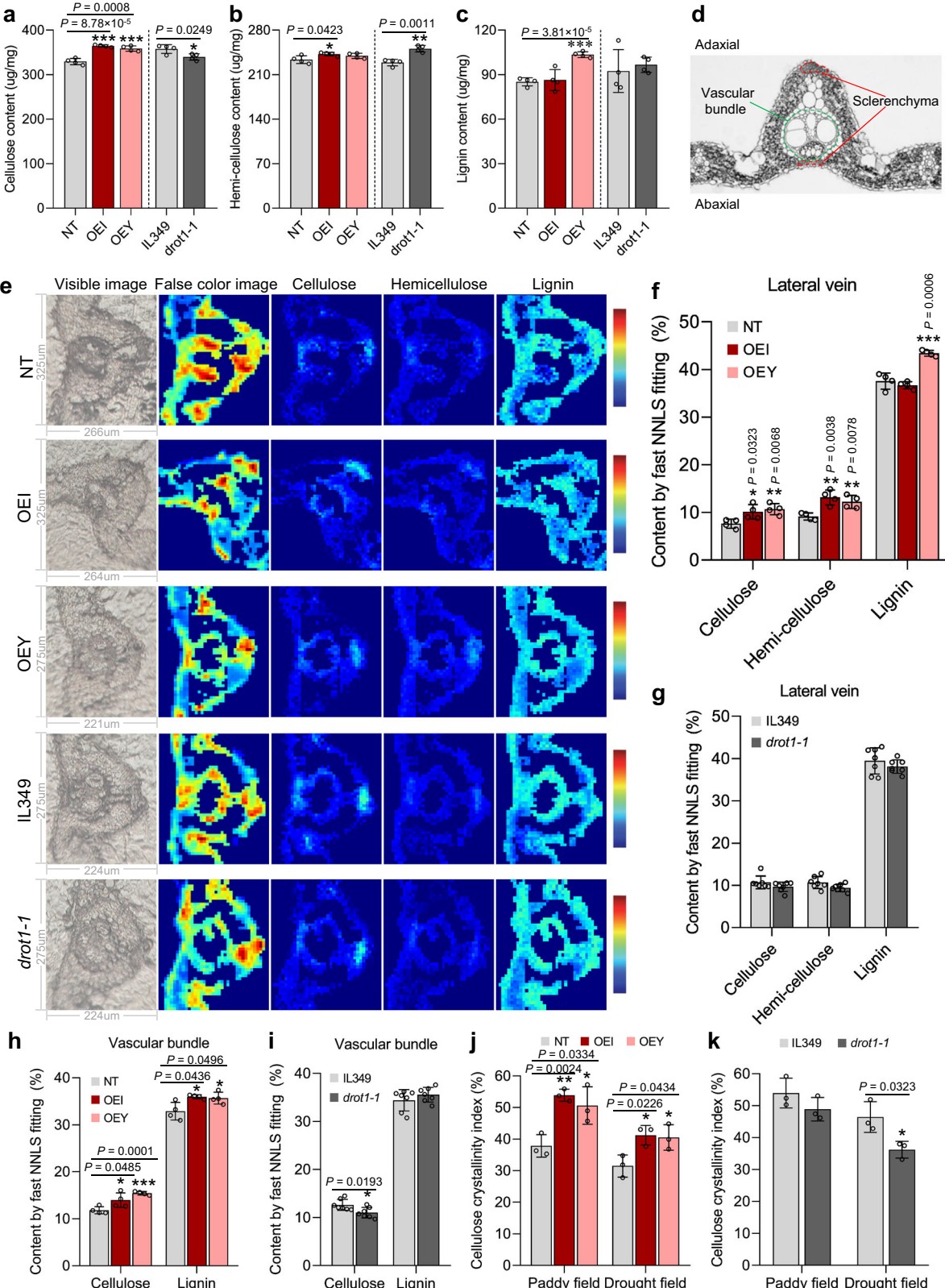

(denoted P1, P2, and P3) in the *DROT1* promoter (Fig. 5n, o and Supplementary Fig. 15a–c). These results prove that the expression of *DROT1* is directly regulated by ERF3 and ERF71.

To test whether *ERF3* acts with *DROT1* in a common genetic pathway, we crossed *ERF3*-overexpressing lines with *DROT1* overexpression lines to generate OE-ERF3/OEI lines. After treatment with 20% PEG for 5 days and recovery for 7 days,

the OE-ERF3/OEI lines showed a similar survival rate to OEI (Fig. 6a, b). This indicates that *ERF3* and *DROT1* might function in a common pathway to regulate drought resistance. To decipher the genetic relevance of *ERF71* and *DROT1*, we generated *erf71* and *drot1-n* mutants in a Nipponbare background using CRISPR/ Cas9 system (Fig. 6c). *drot1-n* was crossed with *erf71* to generate the *erf71/drot1-n* double mutant. The survival rate of the *erf71/*

**Fig. 4 DROT1 affects cell wall properties under drought stress condition. a–c** Quantification of cell wall components of rice plants by chemical analysis. The contents of cellulose (**a**), hemi-cellulose (**b**), and lignin (**c**) obtained from leaf tissues of the indicated plants grown in drought field. Data are means ± s.d. ($n = 4$ biological replicates). **d** Microstructural diagram of cross section of lateral vein in rice leaf. Green-circled area represents vascular bundle, red-circled area represents sclerenchyma. **e** Fast NNLS fitting images of lateral veins in cross section of leaves grown in drought field. From left to right are visible images, false color images (reflecting the location and density of spectral distribution in the range of $1800-800\ cm^{-1}$ throughout the lateral vein, comparisons cannot be made between different rice plants vertically), spectrogram of characterized cellulose, hemi-cellulose and lignin content. The colors of these three columns reflects the relative content of each target component. In this arrangement, comparisons can be made between various components horizontally as well as between different plants vertically. **f–i** Semi-quantitative analysis of cell wall components. The contents of cellulose, hemi-cellulose and lignin determined by fast NNLS fitting for lateral veins of **e** (**f**, **g**). The cellulose and lignin content based on fast NNLS fitting for the segmented vascular bundle from **e** (**h**, **i**). Data are means ± s.d. ($n = 4$ biological replicates for **f**, **h**, and $n = 7$ biological replicates for **g**, **i**). **j, k** Relative crystallinity index (RCI) of cellulose in leaf samples of NT, OEI and OEY (**j**), drot1-1 and IL349 (**k**) from paddy field and drought field, respectively. It is presented as percentage of crystalline in total cell wall components. Data are means ± s.d. ($n = 3$ biological replicates). Asterisks indicate statistical significance by two-tailed Student's $t$-tests (*$P < 0.05$, **$P < 0.01$, ***$P < 0.001$). Source data are provided as a Source Data file.

drot1-n double mutant was similar to that of drot1-n (Fig. 6d, e), suggesting that ERF71 and DROT1 may regulate drought resistance in the same genetic pathway.

Plants respond to drought by going through two stages: rapid growth inhibition followed by recovery and adaptation to a new stable environment[34]. In the rapid drought-response stage, both ERF3 and ERF71 can be quickly induced[25,28]. Our results confirmed that both ERF3 and ERF71 responded quickly to dehydration and drought stress, showing a similar expression pattern to DROT1 (Supplementary Fig. 10f–i). To investigate the expression patterns of ERF3, ERF71, and DROT1 at the drought-adaptation stage, we collected leaf RNA samples of Yuefu, IRAT109, and IL349 grown in the drought field for 90 days, which considered to be under a stable, persistent drought stress condition. Meanwhile, plants grown in a paddy field were used as controls. DROT1 expression levels were the same in plants grown under the persistent drought condition and those grown in the paddy field, indicating that the expression was readjusted during adaptation to long-term drought (Supplementary Fig. 10j). More importantly, its expression in IRAT109 and IL349 was always higher than that in Yuefu, which was associated with the greater drought resistance of IRAT109 and IL349. Interestingly, at this drought-adaptation stage, the expression of ERF3 decreased in all three lines, while the expression of ERF71 increased in IRAT109 and IL349, compared with their expression levels in plants grown in the paddy field (Supplementary Fig. 10k, l). This indicates that the expression of these two regulators may change during the drought-adaptation stage and that their coordinated regulation maintains the expression of DROT1 to rebalance growth and drought resistance.

**Natural variation of DROT1 might improve drought resistance in upland rice.** To investigate the natural variation of DROT1 in germplasm resources and identify elite alleles, haplotype analysis was performed using 743 rice samples (Supplementary Data 6). There are eight haplotypes, among which Hap1-4 mainly occurs in cultivated rice and Hap5-8 mainly in wild rice (Fig. 7a). We found that Hap1 is a japonica-specific haplotype, while Hap4 is mainly found in indica. The proportion of upland rice in Hap3 is much higher than that in the other haplotypes (Fig. 7b). Although Hap5 also showed a higher proportion of upland rice, it is not a major haplotype for cultivar. In indica or japonica, the proportion of upland rice with Hap3 was also higher than the proportion of all upland rice in these two subgroups (Supplementary Fig. 16a, b). Correspondingly, in the whole population, the proportion of accessions with Hap3 in upland rice population was much higher than that in lowland rice population with 73% of the upland rice accessions being tropical japonica and 23% being indica (Fig. 7c). Hap3 is the only haplotype with the nucleotide "T" at position s18975900 of

the DROT1 promoter, which plays an essential role in the high expression of DROT1. By investigating the expression of DROT1 in germplasm under drought stress, we found that the expression level in $DROT1^T$-type accessions was significantly higher than that in $DROT1^C$-type accessions, which indicating Hap3 should be associated with higher expression of DROT1 in natural germplasm (Fig. 7d and Supplementary Fig. 17a). It also showed that the expression level of DROT1 was significantly higher in upland rice than that in lowland rice (Supplementary Fig. 17b). These further suggest that the expression of DROT1 might be important for drought resistance of upland rice.

The expression of DROT1 was significantly higher in upland rice with Hap3 than that in lowland rice with Hap3 (Supplementary Fig. 17c). The accessions with same haplotype of DROT1 showed different expression levels, which reminding us that upstream regulators may diverse between upland and lowland rice. We then investigated the haplotypes of ERF3 and ERF71. ERF3 has 6 haplotypes, among which Hap1-2 mainly occurs in japonica and Hap3-6 are found in indica. (Supplementary Fig. 18a). ERF3 has no specific haplotypes for upland rice. ERF71 has 10 haplotypes, among which Hap1/2/3/6 are japonica-specific haplotypes, and Hap5/7 are indica-specific haplotypes (Supplementary Fig. 18b). Importantly, Hap6 of ERF71 is mainly composed of upland rice, and Hap4 also has a high proportion of upland rice. Through combined haplotype analysis of DROT1-ERF3-ERF71, 12 main combined haplotypes were found, among which $DEE^{1-1-6}$ and $DEE^{3-1-4}$ are upland-specific haplotypes (Fig. 7e). Rice accessions with Hap3 of DROT1 were further divided into four types- $DEE^{3-1-4}$, $DEE^{3-1-6}$, $DEE^{3-3-5}$ and $DEE^{3-5-5}$. There was no significant difference in DROT1 expressions between upland and lowland rice accessions in each subgroup of Hap3 (Supplementary Fig. 17d–f). This indicates that the differentiated expression of DROT1 between upland and lowland rice with Hap3 may be caused by different haplotypes of ERF regulators. To further clarify whether accessions with Hap3 have better drought resistance than those with other haplotypes, we compared the expressions of DROT1 and survival rates among accessions with same haplotypes of ERF3/71 but different haplotypes of DROT1. The results showed that, in japonica, accessions with $DEE^{3-1-4}$ had a significant higher expression level of DROT1 and survival rate than that of $DEE^{1-1-4}$ (Fig. 7f, g). In indica, $DEE^{3-3-5}$ type accessions had significantly higher expression of DROT1 and survival rate than that of $DEE^{2-3-5}$ and $DEE^{4-3-5}$ type accessions. Taken japonica and indica together, the accessions with Hap3 had higher expression of DROT1 and survival rate than accessions with other haplotypes (Supplementary Fig.17g, h). These indicate that Hap3 may be a potential drought resistance haplotype. To sum up, we speculate that the expression level of DROT1, which could be enhanced by the promoter of Hap3 and may be determine by the combined haplotypes of ERFs-DROT1, might be important for drought resistance in rice.

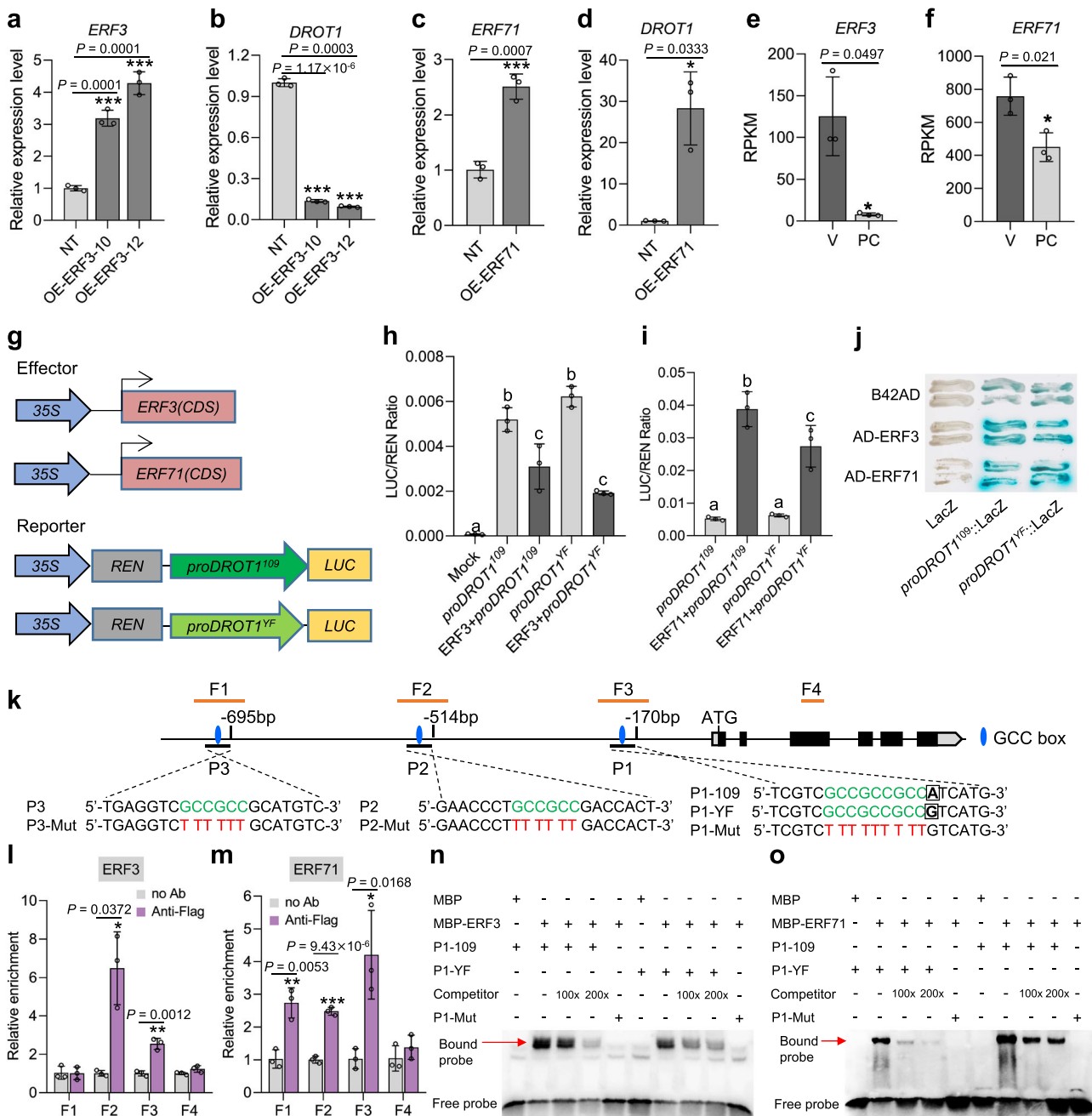

**Fig. 5 DROT1 is directly regulated by ERF3 and ERF71. a, b** Expression of *ERF3* (**a**) and *DROT1* (**b**) in *ERF3*-overexpressing transgenic lines compared with that in NT (negative transgenic control with Nipponbare background). Data are means ± s.d. (*n* = 3 biological replicates). **c, d** Expression of *ERF71* (**c**) and *DROT1* (**d**) in *ERF71*-overexpressing transgenic lines compared with that in NT. Data are means ± s.d. (*n* = 3 biological replicates). **e, f** The differentiated expression of *ERF3* (**e**) and *ERF71* (**f**) in vascular bundles and parenchyma cells. Data are means ± s.d. (*n* = 3 biological replicates). **g–i** Transient expression assays of dual-luciferase by co-transfecting rice protoplasts with the vectors shown in **g**. The transcription activities of the reporters were significantly repressed by ERF3 (**h**), but were strikingly activated by ERF71 with a higher activation activity for the promoter of *DROT1* from IRAT109 (**i**). Mock, co-transfected with an empty reporter construct and an empty effector construct. *proDROT1^109^/ proDROT1^YF^*, co-transfected with a reporter construct and an empty effector construct. 109, IRAT109; YF, Yuefu. Data are means ± s.d. (*n* = 3 biological replicates). Different letters indicate statistically significant differences at *P* = 0.01 by one-way ANOVA test. **j** Y1H assay. ERF3 and ERF71 can bind to the promoter of *DROT1*. *n* = 3 independent experiments.
**k** Schematic diagram showing the three GCC box regions used for EMSA in the promoter of *DROT1*, namely P1, P2 and P3. Green letters in the three probe sequences indicate the core motif of GCC box, red letters indicate the substituted nucleotide sequences in the mutated probes. Black boxes indicate the functional SNP s18975900. F1-F4 are the four fragments used in ChIP-qPCR. **l, m** ChIP-qPCR analysis of ERF3 (**l**) and ERF71 (**m**) enrichment of fragments in the promoter of *DROT1*. Amplified fragments by qPCR are marked with F1-F4 in **k**. Values are means ± s.d. (*n* = 3 biological repeats). **n, o** EMSA shows ERF3 and ERF71 directly bind to the GCC box of P1 in the promoter of *DROT1*. Unlabeled wild type probes were used as competitors. *n* = 4 independent experiments. Asterisks indicate statistical significance by two-tailed Student's *t*-tests (**P* < 0.05, ***P* < 0.01, ****P* < 0.001). Source data are provided as a Source Data file.

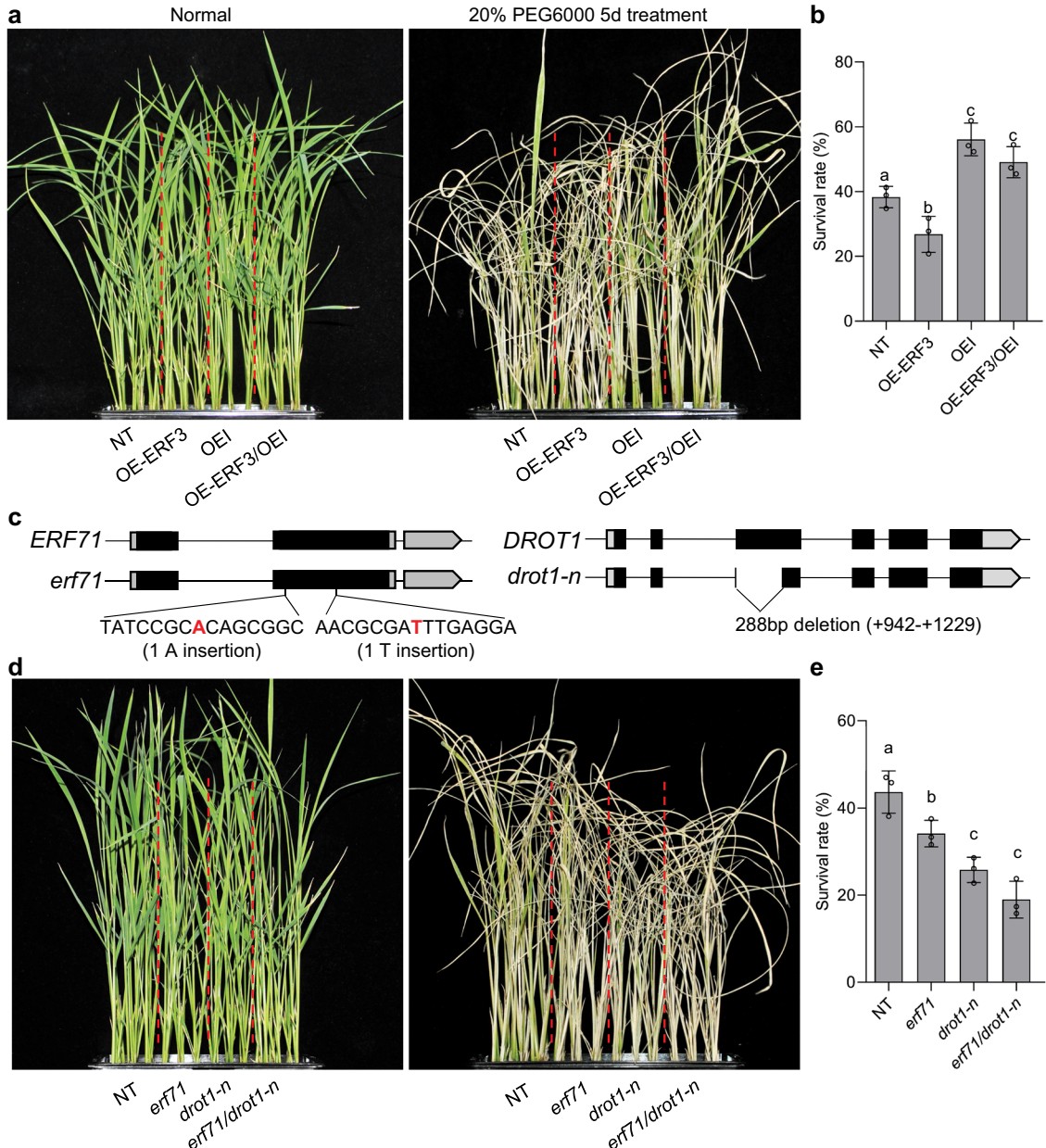

**Fig. 6 *DROT1* genetically interacts with *ERF3* and *ERF71* to regulate drought resistance. a**, **b** Resistance of NT, OE-ERF3, OEI, and OE-ERF3/OEI plants to drought stress simulated by 20% PEG6000. Seedlings grown for 2 weeks under normal conditions (**a**, left) were treated by 20% PEG for 5 days, followed by re-watering for 7 days (**a**, right). Statistical analysis of seedling survival rates after re-watering (**b**). Data are means ± s.d. (*n* = 3 biological replicates). **c** Diagram of *ERF71* and *DROT1* knockout lines. **d**, **e** Resistance of NT, *erf71*, *drot1-n*, and *erf71/drot1-n* double mutant plants to drought stress simulated by 20% PEG6000. Data are means ± s.d. (*n* = 3 biological replicates). Different letters indicate statistically significant differences at *P* = 0.05 by one-way ANOVA with Duncan test. Source data are provided as a Source Data file.

To explore the origin and spread of Hap3, the lineage and geographical distribution of Hap3 were analyzed. The evolutionary relationship of the eight haplotypes indicated that Hap3 evolved on the basis of other haplotypes (Fig. 7h). The haplotype network of *DROT1* showed that Hap3 evolved from wild rice-specific haplotypes, suggesting its late appearance in wild rice (Fig. 7i). By investigating the origin of germplasm with Hap3 and its neighbor haplotype-Hap8, we found that most accessions are densely distributed in Southeast Asian, both in wild rice and cultivated populations (Fig. 7j). This suggests that Hap3 might firstly originate in Southeast Asian. Phylogenetic analysis showed that the tropical *japonica* accessions with Hap3 were grouped with *indica* but not with temperate *japonica* (Supplementary Fig.

19a). The *indica*-specific SNPs exhibited a high frequency in tropical *japonica* between 18.5 and 19.9 Mb, while tropical *japonica*-specific SNPs showed a lower frequency in *indica* in this region (Supplementary Fig. 19b, c), which demonstrated that Hap3 of tropical *japonica* might introgressed from *indica*.

Combining the above results, we propose that *DROT1^Hap3^* of *indica* might originate from *indica*-type wild rice and was subsequently introduced into *indica* during the domestication and differentiation of cultivated rice. After diversification, *DROT1^Hap3^* was possibly introgressed into tropical *japonica* from *indica* to improve its drought resistance and gradually accumulated in tropical *japonica* varieties, thereby increasing the proportion of upland rice in tropical *japonica* subpopulation (Supplementary Fig. 19d).

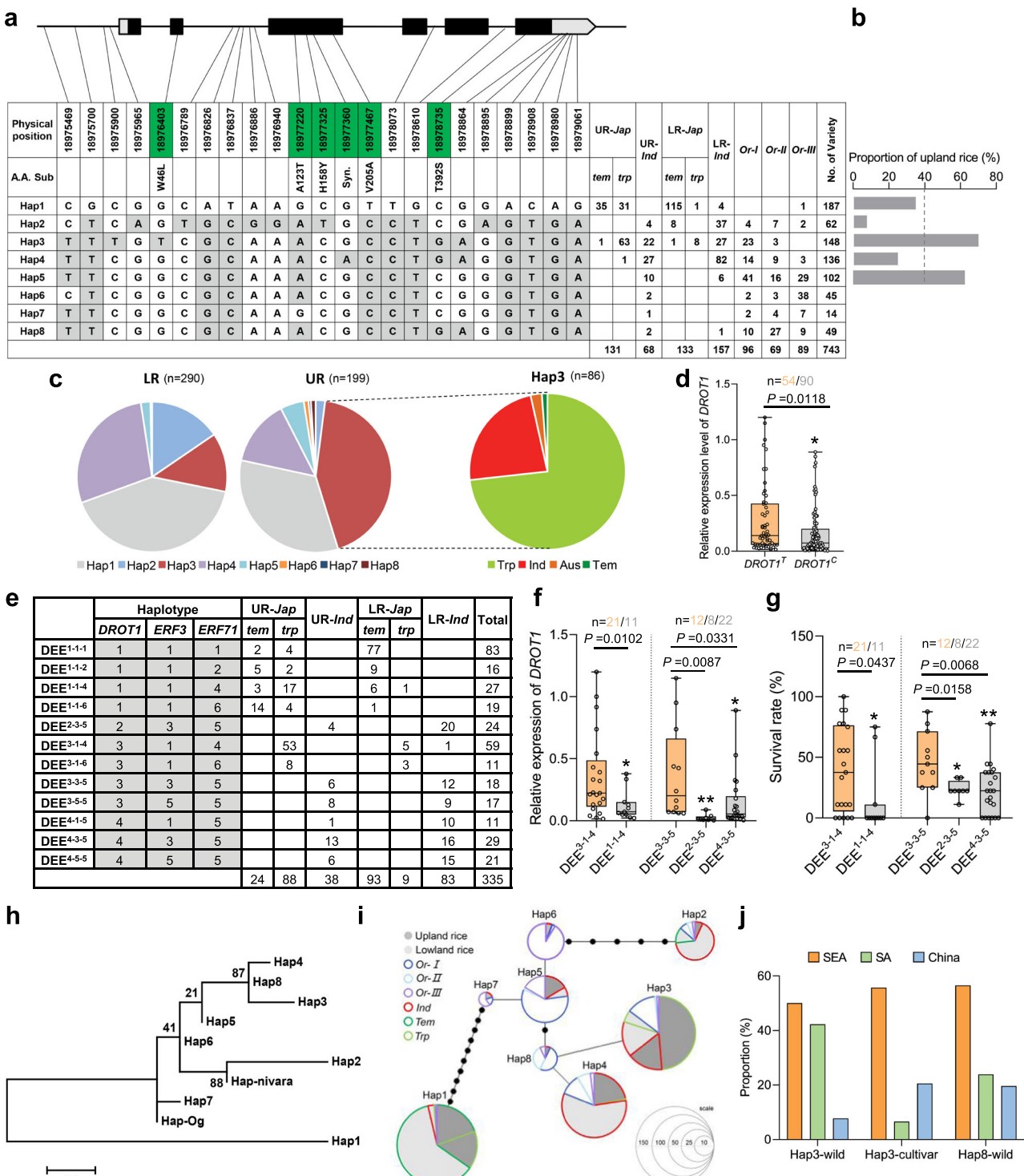

## Discussion

Drought resistance is a complex trait with low heritability and is strongly influenced by both genome background and environment[35]. Previous studies of drought resistance in rice have been mostly based on reverse genetics and involved measuring drought resistance phenotypes in controlled lab conditions or other non-field environments, which could not fully reflect the circumstance of actual drought in the field. To explore genes underlying field drought resistance in rice, linkage analyses and GWAS were conducted[36,37], but it is inefficient and difficult to clone genes with minor effects. With the development of omics technology, multiple approaches can be combined to perform forward genetic studies on field drought resistance of rice. To validate the functions of drought resistance genes in rice, both survival rate and relative growth performance (such as plant height, biomass, and grain yield) should be used to evaluate the overall impact of drought stress, because the former can be used for evaluating the drought resistance of plants under severe arid conditions, while the latter could well reflect the drought resistance of plants under mild drought stress[38]. In this study, we used an integrative approach combining genomes, transcriptomes, and introgression lines to clone a drought resistance gene, *DROT1*. To

**Fig. 7 Haplotype analysis and origin of *DROT1*. a** Haplotypes of *DROT1* in natural population. Nucleotide variations in the coding region are labeled in green boxes. The *DROT1* nucleotide sequences of accessions in the germplasm were compared with that of Nipponbare (Hap1). The number of varieties for each haplotype (Hap1–8) is shown in the right column. The upland rice IRAT109 belongs to Hap3, and the lowland rice Yuefu shares same haplotype with Nipponbare. UR, upland rice. LR, lowland rice. **b** Proportion of upland rice in cultivated accessions with Hap1-Hap5. The dashed line represents the average proportion of upland rice in the entire cultivar population. **c** Distribution of *DROT1* haplotypes in lowland rice and upland rice. **d** Relative expression of *DROT1* in *DROT1^T* and *DROT1^C* type accessions under drought stress. *DROT1^T* indicates the accessions with Hap3 as the SNP site s18975900 is "T"; *DROT1^C* indicates the accessions with s18975900 is "C" (non-Hap3 haplotype). Data are means ± s.d. ($n = 3$ biological replicates). **e** Combined haplotypes of *DROT1-ERF3-ERF71*. **f** Relative expression of *DROT1* in accessions with same haplotype of *ERF3* and *ERF71*. Data are means ± s.d. ($n = 3$ biological replicates). **g** Survival rates of accessions with same haplotype of *ERF3* and *ERF71*. Data are means ± s.d. ($n = 3$ biological replicates). In each box plot of **d**, **f**, and **g**, the center line indicates the median, the edges of the box represent the first and third quartiles, and the whiskers extend to span a 1.5 interquartile range from the edges. **h** Evolutionary relationship of eight haplotypes. *O.nivara* and *O.glaberrima*-specific haplotypes were used as outgroups. The scale bar indicates the average number of substitutions per site for different haplotypes. **i** Haplotype network of *DROT1* shown by the minimum-spanning tree. Circle size is proportioned to the number of accessions with a given haplotype. Circle colors represent different rice subspecies. The dark gray inside the circle indicates upland rice, while the light gray indicates lowland rice. **j** Geographical distribution of rice accessions with Hap3 and Hap8. SEA, South East Asia. SA, South Asia. Statistical significance was calculated by two-tailed Student's *t*-test and *P*-values were indicated in **d**, **f**, and **g** (*$P < 0.05$, **$P < 0.01$). Source data are provided as a Source Data file.

identify its function, we investigated the survival rate of transgenic lines in pot experiments and relative growth performance in drought field. Through the evaluation of these two indicators, the drought resistance function of *DROT1* was confirmed.

Drought stress reduces plant growth, which is considered not only a passive consequence of adverse environments, but also an adaptation to stress[39]. Understanding the balance between growth and survival under abiotic stress is critical to control the stress and growth trade-off for breeding drought-resistant and high-yielding rice varieties. Plant growth relies on cell division and cell expansion, which require coordination between cell wall loosening and biosynthesis[40]. Under abiotic stress, the composition and architecture of the plant cell wall play important roles in influencing the rate and direction of cell and tissue growth[12]. A recent transcriptome analysis also demonstrated that the expression of genes related to cell expansion and cell division was strongly decreased after severe drought stress in rice[41]. However, the mechanism of coordination between drought stress and cell growth remains unclear. In this study, we demonstrate that *DROT1* improves drought resistance by strengthening cell wall structure through increased cellulose content and maintenance of high-cellulose crystallinity under drought stress conditions (Supplementary Fig. 20). Both cellulose synthesis and its crystalline structure influence cell wall formation and plant growth[42]. We found that overexpression of *DROT1* not only enhances drought resistance, but also increases plant height and grain yield under normal conditions, indicating that *DROT1* may improve the balance between growth and stress resistance. *ERF3* negatively regulates drought resistance by inhibiting *DROT1*, but positively manipulates crown root development by regulating the expression of genes involved in cytokinin signaling[43]. Hence, we speculate that *ERF3* may also coordinate plant growth and stress resistance. Under normal growth conditions, plants need continuous cell division to promote growth, during which the cell walls should remain relatively relaxed. Thus, the genes related to the formation of secondary cell walls (such as *DROT1*) should be suppressed to a certain extent. However, under long-term drought stress, cell walls are strengthened to protect cells and maintain cell turgor, requiring more DROT1 protein. Therefore, the inhibition of ERF3 on *DROT1* needs to be removed. We found that the expression of *ERF3* was significantly down-regulated under long-term drought stress (Supplementary Fig. 10k). This indicates that ERF3 coordinates plant growth and drought stress by manipulating the expression of *DROT1*. In addition, during the rapid response stage of drought stress, the expression of *DROT1* was rapidly upregulated, which may be activated by the induced expression of *ERF71*. However, the

expression of *ERF3* was also rapidly induced by instant drought stress (Supplementary Fig. 10f). We speculate that there may be a coordination between *ERF71* and *ERF3* in regulating the expression of *DROT1*. This regulatory mechanism should be an adaptive strategy to balance growth and maximize survival under drought stress.

During the long process of adaptation to arid environments, the typical upland *japonica* rice ecotype became genetically differentiated from that of cultivated lowland rice[44]. Upland *japonica* rice adapts to drought by developing a robust (long and thick) root system[11]. The genes responsible for robust root are considered drought-adapted genes of upland rice. To evaluate whether *DROT1* is also an aerobic drought-adaptation gene of upland rice, we examined the genetics, expression/regulation patterns, and evolution of this gene. First, the causal SNP s18975900 in the promoter of *DROT1* confers higher expression of *DROT1* in upland rice under drought stress. This key SNP is the DNA basis for *DROT1* adapting to drought in upland rice. Second, ERF3 and ERF71 regulate *DROT1* expression by directly binding to the P1 site. This nucleotide variation from C to T enables ERF71 to enhance the activation ability of the *DROT1* promoter (Fig. 5i). *DROT1* not only responds to transient dehydration stress, but also maintains a higher expression level in upland rice under long-term drought stress. By comparing the expression profiles of 220 cultivated rice varieties growing in paddy fields and dry fields, we found that the average expression level of *DROT1* in the dry fields was higher than that in the paddy fields for both *indica* and *japonica*[45]. This indicates that the induced expression of *DROT1* in natural accessions is universal. This fine-tuned response to different stages of drought stress also reflects that *DROT1* has evolved an adaptive response mechanism to drought. Third, Hap3 is the only haplotype in the natural germplasm that harbors potential elite variation of *DROT1*, and the proportion of Hap3 in upland rice is much higher than that in lowland rice. Temperate *japonica* and tropical *japonica* rice share a common ancestor and diversified during a global cooling event about 4,200 years ago[46]. Therefore, tropical *japonica* is genetically closer to temperate *japonica* than to *indica*. However, *DROT1^Hap3* in tropical *japonica* was grouped with *indica*. We speculated that this haplotype might introgress into tropical *japonica* from *indica* for adapting to drought. This is consistent with fact that most *japonica* upland rice are tropical *japonica*, which is distributed in the arid and semi-arid environment. Therefore, we postulate that *DROT1* is also a possible drought-adaptation gene for upland rice. As an important issue, it remains unclear whether the expression of *DROT1* is positively correlated with the degree of drought resistance in rice germplasm. Owing to the complexity and

difficulty in evaluating the degree of drought resistance for germplasm, the survival rate can only partially reflect the degree of drought resistance. Other traits such as relative biomass or plant height are also important for assessing the degree of drought resistance, which remains to be further explored. In addition, given the deficiency of knowledge on upland rice evolution, and limitations of the methods used in this study, it remains an opening question that how did Hap3 originate and disperse during *DROT1* adapting to aerobic drought environments.

## Methods

**Plant materials.** A panel of 271 *japonica* accessions used for GWAS were collected from a rice core collection. Sequence data were obtained from the 3000 Rice Genome Project[47]. A total of 270 distinct introgression lines ($BC_5F_3$) were constructed by crossing and backcrossing IRAT109 with Yuefu, followed by self-crossing[48]. The 88 ILs used for PEG treatment were selected as they have no more than three genome-wide introgression segments from IRAT109.

For generation of the *drot1* mutant by CRISPR/Cas9, two target site fragments were cloned into the binary plasmid pHUE411. The recombinant plasmid was transformed into calli of IL349 and Nipponbare mediated by *Agrobacterium tumefaciens*[49]. The *drot-zh11* was purchased from a knockout mutant library. To construct *erf71* and *drot1-n* mutants, two target site fragments of each gene were cloned into the binary plasmid pHUE411, respectively. The recombinant plasmid was transformed into calli of Nipponbare. The *erf71/drot1-n* double mutant was developed by crossing *drot1-n* with *erf71*. To construct the *DROT1* overexpression plasmids, the full coding sequences were amplified from the cDNA of IRAT109 and Yuefu, and inserted into binary vector pMDC32, separately. Two plasmids were transformed into Nipponbare, respectively. To generate *DROT1* RNAi vector, a 508 bp fragment spanning the third to fifth exon of *DROT1* was amplified from the cDNA of IRAT109 and linked into pTCK303, and then transformed into Nipponbare. Primer sequences used in this study are listed in Supplementary Data 7.

All rice plants were cultivated in the experimental field in Beijing under normal paddy field or drought field conditions as indicated below.

**Investigation of the drought resistance phenotype.** For investigating the drought resistance phenotype of 271 *japonica* rice accessions, rice seeds were directly sown in the soil in a rain-proof shed and irrigated for germination. Three weeks after planting, the seedlings were irrigated thoroughly and then grown naturally without water supply. The soil moisture content gradually dropped from the highest level to about 10% and then kept in drought for nearly 40 days. The grade of leaf rolling and leaf color of each independent repeat were visually inspected by evaluating the overall performance of a row of 12–15 plants. All the materials were planted for two biological repeats with one row for each accession in each repeat. According to a standard evaluation system for rice[50], leaf rolling was scored with five grades. Grade 1 means no curling leaf, grade 2 means curling leaves no more than 25% of total plants, grade 3 means moderate curling leaves accounting for nearly half of the numbers, grade 4 means severe curling leaves shown in 3/4 of the plants, and grade 5 means almost all plants exhibit completely curly leaves. Similarly, leaf color is also scored by five grades. Grade 1 is completely green, and grade 5 is completely dry and yellow. The LRI and LCI were then calculated based on the degree of leaf rolling and leaf color of two repeats, respectively. The best linear unbiased prediction (BLUP) of LRI and LCI was calculated through the linear mixed effect function lmer in the lme4 package of R Version 3.0.1. In addition, the BLUP values of LRI and LCI of all materials were evaluated using principal component analysis and the membership function method. Finally, the drought resistance of each variety was characterized by comprehensive drought-resistant index (DRI), which was calculated based on the weights of LRI and LCI.

To investigate the drought-resistant function of *DROT1* under field conditions, rice plants were grown in both severe drought and paddy fields. Seeds were sown in the seedbed or directly sown in the soil of a rain-proof shed for germination. After 21 days, seedlings grown in the seedbed were transplanted into the paddy field. At the same time, the germinated seedlings in the rain-proof shed were watered once. Then, rice plants in the paddy field were grown under well-watered conditions with normal field management, while rice plants in the rain-proof shed were grown under severe drought stress conditions. The water potential was shown in Supplementary Fig. 5a. The height and aboveground biomass of the plants grown in two environments were investigated at 90 days after sowing. The plant height was measured from the basal part of the plant to the top of the leaf blade. The aboveground parts were collected and dried to measure the weight. The relative plant height and biomass were calculated as values in the drought field divided by the values in the paddy field.

To evaluate grain yield and yield-related traits under moderate drought conditions, rice plants were grown in a rain-proof shed and watered properly throughout the life cycle to ensure plant flowering and grain filling. At least three replicates of each material were randomly planted in a square area of 1.9 meters in

length and width of the nine grids as shown in Supplementary Fig. 9b, c. Replicates were planted in nine rows with row spacing of about 25 cm. Rice seeds were sown directly, leaving 50 plants in each row after germination with plant spacing at the distance of 4 cm. After maturity, all the plants in a single plot were harvested together and the plot biomass were weighted. Then, the fully filled seeds of each plot were collected and dried to weigh the plot yield. The grain yield per hectare was converted from the plot yield. To analyze the grain yield per plant, each line was planted in the paddy field (plant spacing: 15 cm, row spacing: 25 cm) and moderate drought field (similar planting and growth conditions as those for the plot yield experiment) at the same time. More than 30 plants were randomly selected and the dry seeds of a single plant were weighed.

To assess the sensitivity of ILs to drought stress, the surface-sterilized seeds were germinated at 28–30 °C under 15% PEG6000, which considered to be a simulation of drought stress. Seeds germinated in sterilized water were used as control. Ten days later, the shoot length was measured, and the relative shoot length of each IL was calculated as the ratio of shoot length under PEG treatment to that under normal condition.

To simulate water stress treatment for *DROT1*-overexpressing transgenic lines, the seedlings grown in nutrient solution for 1 month were treated with 20% PEG6000 for 3 days, followed by recovering for 7 days. The surviving and dead seedlings of each line were counted, and the survival rate was calculated. Transgenic lines and control lines were planted on the same plate with at least three repeats.

To evaluate drought resistance of *DROT1* at the seedling stage, seeds were soaked in water at 28–30 °C for 2 days and then germinated seedlings with similar vigor were planted in pots containing a mixture of nutrient soil and vermiculite. The pots were put in a water container and seedlings were grown under sunlight for 1 month in summer (25–35 °C). Then, the seedlings were subjected to drought stress without water supply until the leaves and stem were wilted (about 15 days), followed by re-watering for 10 days. The surviving and dead seedlings of each line were counted, and the survival rate was calculated. Transgenic lines and control lines were planted in the same pot. At least 3 pots and 30 plants were evaluated for each line.

**Genome-wide association study.** A total of 2,070,333 SNPs with a minor allele frequency (MAF) of 0.05 were used for the GWAS under CMLM in the GAPIT package (version 2) operated in R environment. A kinship matrix was used to evaluate the parental relatedness among accessions, and the principal components were analyzed to estimate the population structure. The genome-wide significance threshold was determined by permutation tests with 1000 replications ($P < 0.0001$)[11]. A region containing more than two consecutive significant SNPs was considered a single associated signal, and the SNP with minimum $P$-value in the associated signal considered to be the lead SNP.

Linkage disequilibrium (LD) was investigated based on standardized disequilibrium coefficients ($D'$), and squared allele-frequency correlations ($r^2$) for pairs of SNP loci were determined using the TASSEL5.0 program. The distances of LD decay in the regions surrounding the lead SNPs identified in this study were then calculated[51].

**Gene expression analysis.** Transcriptome data used for gene expression analysis in the candidate region of *qDR10b* were obtained in a previous study[11]. Briefly, RNA sequencing was performed on the young leaves of two upland rice (IRAT109 and Haogelao) and two lowland rice (Yuefu and Nipponbare). The RPKM value of each gene in the candidate region with difference between IL349 and Yuefu were compared between upland rice and lowland rice.

To analyze gene expression induced by dehydration, 2-week-old seedlings of IRAT109 and Yuefu were dehydrated for 24 h at room temperature. RNA Samples were collected every 2 h for a total of 12 times. To analyze gene expression under drought stress in pots, rice seeds were germinated and transplanted into pots containing a mixture of nutrient soil and vermiculite. Seedlings were grown for 25 days under well-watered conditions and then subjected to drought stress for 10 days without water supply followed by rewatering. The soil moisture was measured at around 17:00 every day and leaf samples were collected for RNA extraction and expression analysis.

To investigate *DROT1* expression at cellular level, young stem internodes from Nipponbare were fixed in glacial acetic acid and ethanol (1:3) and then embedded in wax (Sigma-Aldrich). The vascular bundles and parenchymal tissues of internodes were sectioned and harvested by LMD 7000 (Leica) laser microdissection system. Total RNA of isolated tissues was extracted for expression analysis.

To investigate the expression of *DROT1* in rice germplasm, 144 accessions including 57 upland rice and 87 lowland rice were selected according to the haplotype of *DROT1-ERF3-ERF71*. The germinated seeds were grown under laboratory conditions in nutrient soil with normal water supply. After grown for 3 weeks, seedlings were subjected to drought stress without water supply for 5 days. Leaf samples were then collected for RNA extraction and gene expression analysis. After an additional three days of drought stress for some of the accessions, the seedlings were re-watered for 5 days. The surviving and dead seedlings of each accession were counted, and the survival rate was calculated.

**GUS assay**. For the construction of ProI and ProY GUS plasmids, the promoter of *DROT1* (840 bp upstream of ATG) from IRAT109 or Yuefu was amplified and inserted into the pCAMBIA1301 vector between *Eco*RI and *Bgl*III restriction sites, respectively. To construct the ProT (*ProDROT1*$^{Yuefu(s18975900C>T)}$::GUS) vector, the promoter of *DROT1* from Yuefu was firstly cloned into pMD19-T, and then the recombinant vector was amplified by PCR using specific primer pairs with T instead of C at s18975900. Next, the restriction enzyme *Dpn*I was used to digest the plasmid template of the PCR products. After purification, the mutated promoter was digested with *Eco*RI and *Bgl*III and cloned into the pCAMBIA1301 vector. The constructs were introduced into Nipponbare by *Agrobacterium*-mediated transformation. GUS histochemical staining assays were performed in the $T_3$ homozygous transgenic plants. Roots, coleoptiles and leaves of germinated seedlings were stained, and the cross-sections of the tissues were observed and photographed with a stereomicroscope. GUS gene expression was analyzed in the leaves of single-copy homozygous $T_2$ generation lines under normal and dehydrated conditions.

**Subcellular localization**. The CDS of *DROT1* without stop codon was amplified from the cDNA of IRAT109 and then combined with the native *DROT1* promoter amplified from the DNA of IRAT109. The combined fragment was inserted into the modified pCAMBIA super1300-GFP and pCAMBIA super1300-mCherry vector. The generated constructs *proDROT1::DROT1-GFP* and *proDROT1::DROT1-mCherry* were transformed into Nipponbare by *Agrobacterium*. Fluorescence was examined in the root epidermis tissues of 2-week-old transgenic plants under a confocal laser-scanning microscope (ZEISS LSM900) and images were collected by ZEN blue 3.1 software. For plasmolysis, the roots were treated with 1 M sorbitol for 15 min. The following excitation (Ex) and emission (Em) wavelengths were used for detection: GFP (Ex = 488, Em = 500–550) and mCherry (Ex = 550, Em = 610).

**Cell wall component examination by the chemical method**. Leaf tissues were harvested from at least 10 plants of each line grown in both paddy and severe drought field conditions. Alcohol-insoluble residues (AIRs) were prepared by treating ground samples with alcohol. For the measurement of cellulose and hemicellulose content, AIRs were destarched with α-amylase, and the residues were hydrolyzed by 72% and 4% (v/v) sulfuric acid in turn. Then, the liquid supernatant was passed through a 0.22 μm filter. The monosaccharide composition was determined by HPLC (Shimadzu LCMS-2020, equipped with an Aminex HPX-87H column). The column temperature was 40 °C with a mobile phase of 0.05 M $H_2SO_4$ and a flow rate of 0.6 mL/min. Cellulose content was quantified by the glucose content, and hemi-cellulose content was converted by the sum of xylose and arabinose content. The lignin content was measured using the acetyl bromide method[52]. In brief, 2 mg of AIRs were dissolved in 0.5 mL acetyl bromide solution (25% v/v acetyl bromide in glacial acetic acid) for 2 h at 50 °C. After adding 2 M sodium hydroxide and 0.5 M hydroxylamine hydrochloride, the absorbance of the supernatant was measured at 280 nm.

**FTIR micro-spectroscopic image analysis**. For visually observing the distribution of the main cell wall components in rice leaf tissues, the middle parts of the flag leaves from rice plants grown under paddy field and drought field for 90 days were fixed in 50% FAA and then embedded in paraffin and employed for FTIR micro-spectroscopic imaging[53]. In brief, 15-μm-thick transverse sections were prepared and transferred onto a ZnS window for FTIR micro-spectroscopic imaging (Spotlight 400, PerkinElmer Ltd., Beaconsfield, Bucks, UK). Visible images were obtained using a charge-coupled device (CCD) camera. All FTIR images were taken using Spectrum IMAGE R1.7.1 Software (PerkinElmer) and collected in transmission mode in the range of 1800–800 cm$^{-1}$ at 4 cm$^{-1}$ spectral resolution and 6.25 × 6.25 μm spatial resolution. The acquired images were subjected to a series of data processing, including noise reduction, background removal, and baseline correction. The fast NNLS algorithm was employed with InSituAnalyze for characterizing the concentration (%) of target components in situ[54]. For comparison between different images in a vertical column, the target component concentration was calibrated based on a calibration coefficient of each sample (the leaf samples from the same part of the plant were ground and weighed; the dry powder was w0, the dry weight after 80% ethanol extraction was w1, and the chemical calibration coefficient was equal to w1/w0). After the above calculation, a maximum-minimum normalization (adjusted to [0,1]) was carried out on the chemical images. In addition, the contents of cellulose and lignin in the vascular bundles segmented and extracted by *K*-means cluster analysis were calculated based on fast NNLS fitting.

**X-ray diffraction**. Leaf samples grown in paddy field or drought field for 90 days were collected and ground in liquid nitrogen with mortar and pestle. After filtering with a 100-mesh nylon membrane, the powder was used for synchrotron X-ray diffraction[55]. The experiments were performed on a six-circle diffractometer (Huber 5020) in the Beijing Synchrotron Radiation Facility Center using CuKα radiation with a wavelength of 0.154 nm. Data were recorded and integrated using the Origin v 9.1 software (Origin). The relative crystallinity index (RCI) was determined as follows: RCI = $(I_{002} − I_{am})/I_{002}*100$, where $I_{002}$ represents crystalline and amorphous materials ($2\theta = 22.7°$), and $I_{am}$ represents amorphous materials ($2\theta = 18°$) using commercial crystalline cellulose as a control.

**TEM and SEM**. The leaves of 45-day-old seedlings (for TEM) and the 2nd internode of flowering rice plants (for SEM) were fixed in a 2.5% (w/v) glutaraldehyde in 0.1 M PBS at 4 °C. The samples were dehydrated through a graded series of ethanol. Then, leaf samples were embedded with Spurr Kit and cut for ultrathin sections. After placing them on a copper mesh and staining with uranyl acetate and lead citrate, the prepared sections were observed under a Hitachi H7500 transmission electron microscope. For SEM, the dehydrated internodes were dried with a carbon dioxide critical-point dryer. Afterwards, the samples were coated with gold and palladium and observed using an S-3000N scanning electron microscope (Hitachi, Tokyo). The cell wall thickness of sclerenchyma cells was measured using ImageJ v1.8.0 software.

**Transactivation activity analysis**. The full-length CDSs of *ERF3* and *ERF71* were amplified and cloned into the vector pGreenII 62-SK to generate effectors. To construct the reporter, the promoter of *DROT1* (728 bp upstream of ATG) from IRAT109 and Yuefu was amplified and cloned into the vector pGreenII 0800-LUC[56]. Transactivation analysis was performed in the protoplasts extracted from 3-week-old leaf sheath of Nipponbare. Transfected protoplasts were incubated in the dark for 16 h at 28 °C, then cells were lysed, and the luciferase activity was detected using the Dual-Luciferase Reporter Assay System (Promega, E1960). The *Renilla luciferase* (REN) gene was used as an internal control.

**Yeast one-hybrid assay**. The *DROT1* promoter amplified from IRAT109 and Yuefu was inserted into pLacZi2μ vector to generate *proDROT1*$^{109}$:: *LacZ* and *proDROT1*$^{YF}$:: *LacZ* reporter constructs, respectively. The full-length CDSs of *ERF3* and *ERF71* were amplified from the cDNA of Nipponbare and inserted into pB42AD to generate AD-ERF3 and AD-ERF71, respectively. The fused AD and LacZ plasmids were co-transformed into yeast strain EGY48. The transformants were grown on SD/-Trp-Ura dropout media containing 20 mg/mL X-gal (5-bromo-4-chloro-3-indolyl-β-d-galactopyranoside) according to the manufacturer's instructions (Clontech).

**Chromatin immunoprecipitation-quantitative PCR (ChIP-qPCR)**. Two-week-old seedings of transgenic lines containing 35 S:ERF3-Flag and 35S:ERF71-Flag were collected for the ChIP assay. In brief, approximately 1 g of leaf samples was ground into powder in liquid nitrogen and cross-linked with 1% formaldehyde for 10 min at 4 °C, and then the reaction was stopped by 0.1 M glycine for 5 min. After nucleus lysis and chromatin isolation, the genomic DNA was sonicated to produce fragments with a size of ~300 bp. The supernatants were pre-incubated with protein A/G agarose beads (Roche) for 2 h. Immunoprecipitation was carried out with Flag M2 Affinity Gel (Sigma, A2220) added into supernatants and incubated at 4 °C overnight. Chromatin immunoprecipitated without antibody was used as a control. The precipitated DNA was purified and quantified by qPCR.

**Electrophoretic mobility-shift assay**. The full-length CDSs of *ERF3* and a truncated CDS of *ERF71* (600 bp starting at ATG) were amplified and cloned into the vector pMAL-c5x and transformed into *E. coli* BL21. MBP-ERF3 or MBP-ERF71$^{1-200}$ proteins were purified using amylose resin (NEB, E8021V). DNA probes containing the GCC box motif were synthesized and 5′-end were labeled with biotin. Probes with the GCC motif replaced by the same number of thymine were used as negative control. Unlabeled oligonucleotides (100- and 200-fold of labeled probes) were used as the competitors. Purified proteins were incubated with DNA probes at room temperature (24–28 °C) for 20 min. DNA gel shift assays were conducted according to the manual for the LightShift Chemiluminescent EMSA Kit (Thermo Fisher Scientific, 20148).

**Haplotype and phylogenetic analysis of *DROT1***. For investigating natural variation of *DROT1* in germplasm, a panel of 743 rice accessions consisting of 254 wild rice and 489 cultivated rice (160 temperate *japonica*, 104 tropical *japonica* and 225 *indica*) were subjected to haplotype analysis. The SNPs of wild rice were obtained from published data[57]. The SNPs of 371 cultivar accessions were obtained from 3000 Rice Genome Project (https://snp-seek.irri.org/), and those of the remaining 118 cultivars were from deep sequencing data of our lab (data not released). A total of 23 SNPs in the 3.6-kb region surrounding *DROT1* were used for haplotype classification.

To construct the maximum-likelihood tree of *DROT1*, haplotypes of 545 rice accessions (140 *O. rufipogon*, 140 temperate *japonica*, 90 tropical *japonica*, 164 *indica* and 11 *aus*) derived from 743 accessions were genotyped by MEGA 6 software (1,000 replications of bootstrap tests). The phylogenic tree was visualized and annotated using EvolView. To construct the minimum-spanning tree, Arlequin version 3.5[58] was used to calculate the distance among haplotypes based on variations of *DROT1* in 545 rice accessions. Arlequin's distance matrix output was used in Hapstar-0.6[59] to draw a minimum-spanning tree.

For combined haplotype analysis of *DROT1-ERF3-ERF71*, 402 rice cultivar accessions were obtained based on qualified genotypic data for all three genes. By classification, 12 main combined haplotypes with a total of 335 accessions were found. The rest were rare types with less than 5 accessions, which were not considered.

**Introgression analysis**. For identification of introgressed region of *DROT1*$^{Hap3}$ from *indica* into tropical *japonica*, 23 *indica* and 57 tropical *japonica* possessing Hap3 of *DROT1* were selected. The SNPs of all 80 varieties from 18.1 to 20.5 Mb on chromosome 10 were obtained from the 3000 Rice Genome Project. The specific SNPs of *indica* or tropical *japonica* were determined by calculating the proportion of each SNP in a certain subgroup, and the nucleotide with higher frequency (SNP frequency > 0.9) is considered the specific SNP types of this group (In fact, not all these specific SNPs are necessarily true subspecies-specific sites, but the subspecies-specific sites must be included in them). These specific SNPs were then subjected to calculate the nucleotide frequency in the other subgroup.

**Reporting summary**. Further information on research design is available in the Nature Research Reporting Summary linked to this article.

## Data availability

Data supporting the findings of this work are provided in the paper and its Supplementary Information file. The genetic materials supporting the findings of current study are available from the corresponding authors upon request. Source data are provided with this paper.

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

## Acknowledgements

We would like to thank R. F. Huang (Chinese Academy of Agricultural Sciences) for kindly providing seeds of *ERF3* overexpressed transgenic lines, Y.H. Pan (GKLRGB-SKLAB Union Laboratory, Guangxi Academy of Agricultural Sciences) for the help of field work. We also thank F. Qin and Y. Guo (China Agricultural University) for valuable suggestions to the work, R. D. Qu (North Carolina State University) for critical reading and suggested revisions for the manuscript. This work was supported by grants from the Ministry of Science and Technology of China (2021YFD1200502), the National Natural Science Foundation of China (31861143007, 32001521 and 31601278), China Post-doctoral Science Foundation (2019M650902), The Project of Sanya Yazhou Bay Science and Technology City (SYND-2022-16) and Project of Hainan Yazhou Bay Seed Lab (B21HJ0508).

## Author contributions

Z.L. conceived the project; X.S. and H.X. designed the research and analyzed the data under the supervision of Z.L.; X.S. and C.J. conducted GWAS; X.S. and H.X. performed genetic transformation test and drought resistance evaluation; X.S. performed expression analysis and subcellular localization assay; D.Z. performed LCM; X.S., Z.Y. and Y.H. performed FTIR micro-spectroscopic image analysis; X.S. performed SEM, TEM, HPLC, XRD experiment, transactivation activity, EMSA and ChIP assay; X.S. performed haplotype and evolution analyses; S.M., J.D., X.W., W.L., H.G., G.L., J.Q. and C.L. assisted with part of experiments and data analysis; W.Z., L.H., Y.Z., Y.P., H.Z., J.L. and Z.Z. provided advice on the experiments; X.S., H.X. and Z.L. wrote the manuscript.

## Competing interests

The protein sequence of DROT1 is the same as OsCOBL4, which has been granted patents in China (Z.L. 2013 1 0322634.4) and the United States (US 10, 190, 133, B2). For patent granted in China, the inventors are Z.L., H.X., P.L., J.L., H.Z. The title of the patent is "Plant drought resistance-related protein OsCOBL4 and its applications". The invention provided a plant drought tolerance-related protein OsCOBL4 and its encoding gene and application. For patent granted in the United States, the list of inventors is Z.L., H.X., P.L., J.L., H.Z. The title of the patent is "Compositions and methods for improving abiotic stress tolerance". The invention provided abiotic stress tolerant plants, as well as methods and compositions for identifying, selecting and/or generating abiotic stress tolerant plants. Other authors claim no competing interests.
