## [Peer Review File · Nature Communications]

Natural variation of DROT1 confers drought adaptation in upland riceReviewers' Comments:

Reviewer #1:

Remarks to the Author:

This study identified a drought-tolerance gene, DROT1, in upland rice using integrative approaches joining GWAS, introgression line analysis, and transcriptome profiling. This gene encodes a COBRA-like protein involved in the adjustment of cell wall structure under drought. Its expression is regulated by ERF3 and ERF71, both of which are drought responding transcription factors. This study also investigated the origin and spread of DROT1 within *Oryza sativa*. Many drought-tolerance genes have been identified in rice, but genes and mechanisms conferring aerobic drought tolerance in upland rice are still largely unknown. The findings in this study could be important to advance our understanding of drought adaptation mechanisms and its evolutionary origin in upland rice. However, there are still several comments on this article.

- 1) The expression pattern of DROT1 in IRAT109 and Yuefu is shown in Fig. 1f, but this is only short-term expression data (0-6 h). How about the long-term expression of this gene (days)? A major conclusion in this study is that the expression level of DROT1 determines the degree of drought tolerance in upland rice. Thus, genotypic (statistical) comparison of DROT1 expression is critical (both short and long-term).
- 2) It is indicated that the balance between ERF3 and ERF71 expression determines the expression of DROT1. However, time-course expression data of ERF3 and ERF71 along with DROT1 are not shown. These data must be compared in IRAT109 and Yuefu.
- 3) Drought tolerance was evaluated under severe drought conditions in pots and fields. Does DROT1 contribute to maintained grain yield and yield-related traits under sub-lethal drought conditions?
- 4) What is the haplotype of NT used in Fig. 5a, b, c, and d? Does the haplotype affect the expression of DROT1 mediated by ERF3 and ERF71?
- 5) It is mentioned that ERF3 and ERF71 directly regulate the expression of DROT1. Transient expression assays do not determine whether an induced expression is directly or indirectly regulated. EMSA can provide evidence showing direct regulation, but it is an *in vitro* experiment system. I feel additional *in vivo* evidence is required to support this. Also, the EMSA data is not quantitative. Because the direct regulation by ERF3 and ERF71 is a key finding in this study, qualitative data along with statistical evaluation should be provided.
- 6) The major haplotype in tropical upland rice is Hap3. However, there are still several lowland rice accessions which possess Hap3. Are the expression levels of Hap3 in lowland rice lower than those in upland rice under drought?
- 7) In Supplementary Fig. 13, these expression data were not statistically evaluated. For example, ERF71 in Yuefu may not be induced in a drought field relative to a paddy field. Also, the expression levels of DROT1 are similar in paddy and drought fields. It is mentioned in lines 359-359 that this DROT1 level is necessary to improve the resistance of rice under long-term (90-day) drought stress. How does non-induced DROT1 confer drought tolerance in upland rice? Also, what is the developmental stage when rice was grown for 90 days in paddy and drought fields? Which data should we see to find IRAT109 is still tolerant to drought after 90 days of drought treatment?
- 8) The position of asterisks in Supplementary Fig. 6b should be fixed.
- 9) Fig. 1k: What tissue, developmental stage, growth conditions were used for analysis?

Reviewer #2:

Remarks to the Author:

The authors first used GWAS and analysis of an ILs to identify three loci (i.e., qDR4a, qDR4b and qDR10b) that associated with two traits (DRI and LRI) and played important roles in drought resistance of japonica rice. They then identified a candidate gene (DROT1) that conferred drought resistance at qDR10b by multiple approaches including local LD, transcriptome and gene-based association analyses and validated the drought resistant function of DROT1 by transgenic experiments. They further demonstrated that DROT1 improved drought resistance by adjusting cell wall structure and was regulated directly by two drought responsive TFs (ERF3 and ERF71). Finally, they suggested that the elite haplotype of DROT1 in upland rice originated from wild rice and introgressed into tropical japonica from indica. Overall, this is very good study with substantial work behind. The results or findings are of significance in the isolation and functional investigations of drought-stress genes in rice or crops in general. The manuscript is well organized except for Discussion (see the comments below) and written clearly. Despite these, I have a few of concerns and uncertainties listed below.

1. I am confused with the section "Natural variation of DROT1 improves drought resistance in upland rice". First, the descriptions on 8 haplotypes such as those associated with "unique", "major", "proportion" did not make much sense at all. No clearcut exists among groups. Second, I am curious about how the NJ tree (Fig. 6c) was generated. Did the authors used Hap1 as the outgroup? If yes, why choose Hap1? If no outgroup was used, the argument "... Hap3 evolved on the basis of other haplotypes, suggesting its late appearance in the rice ancestors" would be problematic. Third, it's not appropriate to mention "the tropical japonica of Hap3 was grouped with indica, inferring it introgressed from indica" because Hap3 occurred in almost all groups, including japonica upland and lowland rice, indica upland and lowland rice (Fig. 6a). I believe that the key is to address the origin of Hap 3 (lineage/group and geographical area) and its dispersal process rather than to focus exclusively on "the Hap3 of tropical japonica".

2. Given that the authors "hypothesize that Hap3 is a typical drought-resistant haplotype in upland rice" (p. 376-377), how to explain the presence of Hap3 in lowland rice? Or is the gene drought-resistant or not in lowland rice? Similarly, if "DROT1 may be an aerobic drought adaptation gene in upland ecotype rice" (p. 492), why not be the same outcome of this gene in lowland rice that has Hap3 too. In this sense, the section "DROT1 may be an aerobic drought adaptation gene in upland ecotype rice" in Discussion should be reconsidered.

3. In the third section of Discussion, the authors raised an interesting issue associated with the stress and growth trade-off that is worthwhile attention. However, the other three topics the authors chose seemed to me very superficial without concrete take-home message. In another word, the authors spent too much space to repeat what the Results have already mentioned. These sections should be shortened substantially, with more concise presentations and to the points. In another word, these sections should be reorganized with clear hypotheses raised and discussed, and otherwise might be removed entirely.

4. p. 150-152: Two cultivars were used in the expression analysis for both upland and lowland rice. So, does Figure 1g. show the average of expression values?

5. In the legend of Supplementary Fig.1, two photos showing "leaf curling (upper right) and leaf color (lower right)" did not present clearly the features that were described in the text. Are the enlarged photos in which leaf curling and leaf color could be demonstrated clearly? Also, the authors should provide the percentage of variation that explained by PC1 and PC2 in the figure.

Reviewer #3:

Remarks to the Author:

In this manuscript, the authors use GWAS in combination with genetics, molecular biology, and phenotypic and physiological characterization to isolate a COBRA homolog, DROUGHT1, as being responsible for drought resistance in upland rice. The fact that the authors isolate the exact SNP responsible for one of the loci identified in their GWAS analysis is impressive, and the fact that loss of DROT1 reduces drought tolerance whereas overexpression of the gene enhances drought tolerance, in combination with transgenic promoter-GUS analyses and phylogenetic tracing of the DROT1 gene, provides convincing evidence of the importance of this gene for adaptation to drought. However, I found the data reporting cell wall modifications in DROT1 mutant and overexpression lines to be incomplete, and the authors do not provide compelling mechanistic data that explain precisely how DROT1 functions at the molecular, cellular, or physiological levels. The COBRA family of proteins has long been enigmatic, and it is a shame that this work does not really elucidate their functions beyond what is known, at a mechanistic level at least. Specific comments follow.

The paper is poorly written, with a very high frequency of syntax and grammar errors – the errors are so numerous that I have not listed them in this review, but extensive language editing would be required for the manuscript to be suitable for publication. For example, cellulose microfibrils, which are part of the core message of the paper, are variously called “microfilaments” (which is used to refer to actin) and “microfibers” – neither of these terms is correct.

A major weakness of the paper is the cell wall analyses. In Figures 4 and S9, FTIR microscopy, for which peaks can be difficult to accurately assign, is used as the main analytical technique, but these data should be corroborated with immunolabeling, staining, and/or biochemical analysis of cell wall composition in the different lines. As the data stand now, they only allow rough speculation as to the mechanistic connection between loss/gain of DROT1 expression, changes in cell wall composition, and drought tolerance, as evinced by the authors in both their Results and Discussion sections dealing with the cell wall data. As they stand now, the data are too preliminary to draw firm conclusions.

In constructing the phylogenetic tree of DROT1, the authors should use maximum likelihood, which will provide a more accurate tree than neighbor joining.

Figure 2 f, g, i, j: the y axes are confusingly labeled (relative to what? no explanation in the legend), and should all start at zero.

Subcellular localization in Figure 3: there is a strong potential of artifacts from using a double 35S promoter, and using the native promoter would be much better.

Figure 3F: it is not clear that multiple cells are plasmolyzed in this image. Using 10% NaCl is not a standard method of plasmolysis, and could lead to cell death, causing artifacts; using 1 M sorbitol is preferable. If the protein is apoplasmic as depicted in Figure S15, tagging it with a pH-robust fluorescent protein such as mCherry would be preferable to GFP.

A few examples of writing errors:

L31: “jointing” should be “joining” or “combining”

L33: should be “that confers...”

L281: what are “cellulose monomers”? Do you mean UDP-glucose, the substrate for cellulose synthesis, and why would more of it be produced under drought?

L282: should be “cellulose microfibrils”, not microfilaments (actin)

Reviewer #4:

Remarks to the Author:

The manuscript "Natural variation of DROT1 confers drought adaptation in upland rice" by Sun, et al described a COBRA-like protein, DROT1, plays an important role in drought regulation. The SNP alteration in the promoter of DROT1, affects the expression level, thereby confers divergent drought resistance in upland rice and lowland rice, showing a potential value in breeding. In this paper, the results are interesting, but this story is a little bit simple and molecular evidences are weak.

Therefore, I think that it can not be accepted on "Nature Communications" in current form.

Major Concerns:

1, The CRISPR experiment generally is very efficient for gene knock-out and create frameshift mutation. The paper mentioned only one line obtained showing a similar phenotype. It is a bit unusual.

2, the different expression level of DROT1 induces divergent drought resistance in upland rice and lowland rice, so I think that RNAi is more suitable than CAS9, or two experiments should be performed.

3, DROT1 confers drought adaptation by affecting cell wall properties including cellulose content and cellulose crystallinity. This finding is valuable, but How DROT1 regulate cell wall component, and its interaction genes should be explored.

4, Two regulators ERF3 and ERF71 regulate DROT1 expression by binding its promoter. The EMSA result is not perfect, and yeast assay and ChIP should be performed.

5, Genetic interactions are needed between DROT1 and ERF3/ERF71.

Minor Concerns:

Phylogenetic analysis should be performed using the maximum likelihood method.

Reviewers' Comments:

Reviewer #1:

Remarks to the Author:

The resubmitted manuscript contains new data and revisions, which have made this article better. However, I still have the following questions which should be addressed before consideration for publication in Nature Communications.

(1) Detailed analyses of transgenic lines with altered expression of DROT1 demonstrated that DROT1 is a drought tolerance regulator in rice. However, I feel that the data presented here are insufficient to conclude that the 'natural variation' of DROT1 confers drought adaptation in upland rice. A major reason for this is that there are no tolerant haplotypes in rice. It may be possible that the expression level of DROT1 is more critical for drought tolerance than its haplotype.

(2) Expression levels of DROT1 are compared among upland and lowland rice accessions in Supplemental Figure 18. The expression levels of DROT1 in many lowland rice accessions are higher than those in upland rice accessions. Is the expression level of DROT1 positively correlated with the degree of drought tolerance among these accessions? If so, drought tolerance is determined by the expression level of DROT1, rather than its haplotype.

(3) The data about the origin and spread of DROT1 are insufficient to conclude the significance of Hap3 in drought tolerance in upland rice. I agree that DROT1 is a drought tolerance regulator in rice, but it appears that its haplotype is not directly associated with drought tolerance. Instead of its haplotype, the factors regulating DROT1's expression may characterize drought tolerance in upland rice. I believe that this part is critical for the quality and uniqueness of this study, but it will still need extensive work to make some conclusive statement regarding a determinant that distinguishes between upland and lowland rice.

(4) In Fig. 6b, the phenotype of OE-ERF3 has been rescued by crossing the transgenic line with OEI. The data demonstrate that ERF3 acts upstream of DROT1. However, the phenotype of erf71 is not crossed with OEI in Fig. 6e.

Reviewer #2:

Remarks to the Author:

The authors have tried hard to address the points that all reviewers raised, including the major concerns and minor points in my review. The revised version is improved in numerous areas by adding new data and analyses as well as interpretations. Particularly, I appreciate very much the hard work and detailed explanations provided by the authors in responding to the major concerns and minor points by all reviewers. Despite these, I am still not satisfied with the revisions on the last section of Results, as I mentioned in my original reviewing report.

1. The new Supplementary Fig. 17a (the haplotype network) is very informative and more important than the Fig. 7c. The figure indicates that Hap3 originated from wild rice because the haplotype nearest to Hap3 consists mainly of wild rice. So, I agree that "Hap3 evolved from wild rice-specific haplotypes" (please indicate clear all the haplotypes in supplementary Fig. 17 not just Hap3!). However, the descriptions from lines 413-420 in revised text, alongside with the explanations in responding letter did not solve my concerns. In responding letter regarding origin of Hap3, the authors mentioned "We found that Hap3 is mainly composed of Or-I wild rice, indica and tropical japonica. This indicates that Hap3 originated from indica-type wild rice. ..., we speculate that Hap3 of indica-type wild rice was simultaneously introduced into indica of China and SEA." In fact, Or-II (japonica-type wild rice) also contained Hap3 (though low frequency) and so could not be ruled out as ancestral lineage. Moreover, it's hard to understand why Hap3 was present in LR with pretty high frequency

(36/148=24.3%) (Figs. 7a and 7b), as I curious in my original reviewing report.

2. To explore the origin and spread of a haplotype or allele, haplotype network analysis is one of effective approaches. In this sense, the new Supplementary Fig. 17a (the haplotype network) is very informative and more important than the Fig. 7c. However, to look at the components of Hap4 is not enough because introgression of Hap3 from cultivated rice into wild rice cannot be ruled out. On the other hand, identification of the components of the wild haplotype nearest to Hap3 is critical. In this sense, I agree that "Hap3 evolved from wild rice-specific haplotypes" because the haplotype nearest to Hap3 consists mainly of wild rice in Supplementary Fig. 17a (please indicate clear all the haplotypes in supplementary Fig. 17 not just Hap3!). Nevertheless, all three wild lineages (Or-I, Or-II Or-III) are present in the haplotype nearest to Hap3 (Supplementary Fig. 17a) and Hap3 was found in all areas (Fig. 7d). In a word, where the Hap3 originated exactly remains unclear. The sentence "Phylogenetic analysis showed that the tropical japonica accessions with Hap3 were grouped with indica but not with temperate japonica, implying that this haplotype had introgressed from indica (Fig. 7e)." (Lines 424-426) is problematic. I would stress that the UR was scattered across the entire tree (Fig. 7e) and no evidence is available so far to support that Hap3 emerged in tropical japonica ! (Figs. 7f. g were not relevant to this issue). To summarize, it's not appropriate to say "The elite haplotype of DROT1 in upland rice was introgressed into tropical japonica from indica" (Abstract). "this haplotype had introgressed from indica (Fig. 7e)" (lines 425-426).

3. In responding my second concern, the authors presented "three aspects", which, as a matter of fact, all are sorts of speculations and questionable. Say, where is the evidence to prove "This indicates that the upstream regulation pathways of DROT1 may have diversified between upland and lowland rice, which caused the differential expression of DROT1 in upland and lowland rice accessions with Hap3 under drought stress." To my knowledge, these rebuttals or interpretation are largely speculations or pretty weak without solid evidence. In a word, I still do not understand why Hap3 was present or what function Hap3 has in lowland rice if it contributes to drought tolerance although the authors mentioned "The above analysis helps us understand why Hap3 is present in lowland rice."

Reviewer #3:

Remarks to the Author:

The authors have addressed my concerns to a large degree, and I commend them for their careful revision of the manuscript.

Reviewer #4:

Remarks to the Author:

In the new manuscript, the authors verified that DROT1 is directly regulated by ERF3 and ERF71 through yeast one-hybrid, ChIP-qPCR, and EMSA experiments, and further confirmed that ERF3/ERF71 acts with DROT1 in a common genetic pathway. Although the molecular mechanism between ERF3/ERF71 and DROT1 has been supplemented, there are still some concerns based on the current results. My main concerns and uncertainties are listed below.

1. I am confused by the inconsistent results presented by the Chip experiment and the EMSA experiment. ChIP analysis revealed that ERF3 enrich two of the three GCC box containing fragments (Fig. 5k-i), while EMSA analysis demonstrated that ERF3 could bind to the three GCC box-containing fragments in the DROT1 promoter (Fig. 5n, and Supplementary Fig. 15a).
2. Given that the authors "DROT1 is directly repressed by ERF3 and activated by ERF71" (p. 37-38), how to explain the OE-ERF3/OEI lines showed a similar survival rate to OEI (new Fig. 6a, b). To more rigorously verify whether ERF3 and DROT1 function in a common genetic pathway, *erf3* and *erf3/drot1* double mutants should be generated.

Point-by-point Response to Reviewers' comments

Reviewer #1 (Remarks to the Author):

This study identified a drought-tolerance gene, *DROT1*, in upland rice using integrative approaches joining GWAS, introgression line analysis, and transcriptome profiling. This gene encodes a COBRA-like protein involved in the adjustment of cell wall structure under drought. Its expression is regulated by *ERF3* and *ERF71*, both of which are drought responding transcription factors. This study also investigated the origin and spread of *DROT1* within *Oryza sativa*. Many drought-tolerance genes have been identified in rice, but genes and mechanisms conferring aerobic drought tolerance in upland rice are still largely unknown. The findings in this study could be important to advance our understanding of drought adaptation mechanisms and its evolutionary origin in upland rice. However, there are still several comments on this article.

Response : Thank you for your positive comments.

1) The expression pattern of *DROT1* in IRAT109 and Yuefu is shown in Fig. 1f, but this is only short-term expression data (0-6 h). How about the long-term expression of this gene (days)? A major conclusion in this study is that the expression level of *DROT1* determines the degree of drought tolerance in upland rice. Thus, genotypic (statistical) comparison of *DROT1* expression is critical (both short and long-term).

Response: Thank you for your valuable comments and suggestions.

For investigating the time-course expression of *DROT1* in short-term response to drought, two-week-old seedlings of IRAT109 and Yuefu were dehydrated under room conditions. RNA Samples were collected every two hours for a total of 12 times during the 24 hours dehydration. Since the seedling plants are completely dry after being dehydrated for 24 hours, we consider that further gene expression analysis on the samples after that is meaningless. We found that *DROT1* is induced immediately by dehydration in both IRAT109 and Yuefu, and its expression reached the highest at 8-10 hours after dehydration (new Supplementary Fig.10c). Moreover, the expression of *DROT1* in IRAT109 was much higher than that in Yuefu. This indicated that *DROT1* has stronger drought-response ability in upland rice than that in lowland rice.

To explore the long-term expression of *DROT1* under drought, seedlings of IRAT109 and Yuefu were grown in pots for 25 days with well-watered conditions, and then subjected to drought stress for 10 days by stopping water supply, followed by

re-watering for 4 days. We measured soil moisture at the same time every day and meanwhile collected leaf samples for RNA extraction and expression analysis. During the process of drying out of the soil, the expression of *DROTI* did not change much in the early days. However, on the 9th day, when soil moisture dropped to 3%, the expression of *DROTI* induced instantly with a higher level in IRAT109 than in Yuefu (new Supplementary Fig.10d, e). After the seedlings were re-watered, the expression of *DROTI* decreased immediately in Yuefu, while it maintained in a higher level in IRAT1109 for a period of time. We also found the expression of *DROTI* decreased in the early days, which maybe related to the stimulus response of plants or other mechanisms during the process of water potential decline. However, when soil moisture declined to drought levels, the expression of *DROTI* increased rapidly. This indicates that the induced expression of *DROTI* in response to drought requires a certain threshold of water potential below which it can be activated immediately. Therefore, we suggest that the induced expression of *DROTI* is positively correlated with drought resistance.

Related results have been added to the main text in line **243-251** and new Supplementary Fig.10 c-e. Because we supplemented the expression data of *DROTI* between IRAT109 and Yuefu under dehydration for 0-24h, the expression data of IRAT109 and Nipponbare dehydrated for 0-8h in previous version has been deleted in this new version.

2) It is indicated that the balance between *ERF3* and *ERF71* expression determines the expression of *DROTI*. However, time-course expression data of *ERF3* and *ERF71* along with *DROTI* are not shown. These data must be compared in IRAT109 and Yuefu.

Response: Thank you for raising these important points.

To elucidate the time-course expression of *ERF3* and *ERF71*, we performed the expression analysis on the same RNA samples used for *DROTI* expression analysis. Under dehydration, both *ERF3* and *ERF71* showed induced expression patterns and reached the first peak at 8th -10th hour, which is similar to that of *DROTI* (new Supplementary Fig.10f, g). This suggests that *ERF3* and *ERF71* could quickly

respond to drought, which may coordinately activate the expression of *DROT1*. After that, the expression of *ERF3* and *ERF71* continuously increased and reached the highest level at 18 hours, while the expression of *DROT1* decreased during this time. It implies that *ERF3/ERF71* may also regulate other drought-related genes in the late stage of dehydration stress. Comparing the expression level between IRAT109 and Yuefu, we found that the expression of *ERF3* was higher in IRAT109 than in Yuefu almost in the whole stage, and *ERF71* also showed higher expression level in IRAT109 in the first stage. It is unusual that *ERF3* exhibited a continuous high expression under dehydrate stress, which should repress the expression of *DROT1*. We hypothesized that the expression of *DROT1* driven by *ERF71* should be stopped by *ERF3* at any time if needed, which is just like a speeding car needs to be ready to brake in case of an emergency. In this case, post-transcriptional modification of *ERF3* may be more important in regulating *DROT1* expression.

In the long-term drought stress, *ERF3* and *ERF71* showed similar expression pattern to *DROT1* with their expression induced from 9th day after drought treatment (new Supplementary Fig.10h, i). This indicates that the expression of *DROT1* is strongly correlated with *ERF3* and *ERF71*. The expression of *ERF3* and *ERF71* was also higher in IRAT109 than in Yuefu.

In short, we suggest that the drought induced expressions of *ERF3* and *ERF71* resulted in the increased expression of *DROT1*, and their differentiations between upland and lowland rice lead to the different expressions of *DROT1* in the two ecotypes. We consider that the balancing mechanism between *ERF3* and *ERF71* in determining the expression of *DROT1* is complicated, but worthy of further study in the future.

The related results were added in the main text in line **381-383**, and new Supplementary Fig. 10 f-i.

3) Drought tolerance was evaluated under severe drought conditions in pots and fields. Does *DROT1* contribute to maintained grain yield and yield-related traits under sub-lethal drought conditions?

Response: Thank you for raising this question.

To evaluate whether *DROTI* contributes to grain yield and yield-related traits under sub-lethal drought conditions, grain yield and plant biomass were investigated under normal and moderate drought conditions.

Moderate drought field was under rain-prove shed with watering for 8 times throughout the life cycle to ensure plant flowering and filling (new Supplementary Fig.9a). For analyzing grain yield per plant, each line was grown in paddy field and moderate drought field at the same time. More than 30 plants were randomly selected and dry seeds of single plant were weighted in both paddy field and moderate drought field. The grain yield per plant of *drot1-1* was significantly lower than that of IL349 under moderate drought field, while there is no difference between them under paddy field (new Fig. 2m). We found both OEI and OEY lines had a significant higher grain yield per plant under paddy field, but only OEI exhibited a higher grain yield per plant than NT under moderate drought field (new Supplementary Fig.9d). It is possible that this moderate drought condition was still severe for materials with Nipponbare background.

To obtain grain yield per hectare in moderate drought condition, each material was directly sown in the field plots with equal size. Planting schematic diagram and growing performance are shown in new Supplementary Fig.9b, c. After maturity, grains in each plot were harvested and grain yield per plot were weighed and converted into grain yield per hectare. The results showed that grain yield per hectare was significant lower in *drot1-1* than in IL349 under moderate drought field (new Fig. 2n). The grain yield per hectare of OEI and OEY also showed no difference with NT (new Supplementary Fig.9e). As a yield related trait, plant biomass per plot was investigated. Both OEI and OEY lines had significantly higher plant biomass per plot than NT, and *drot1-1* showed a significantly lower plant biomass per plot than IL349 (new Supplementary Fig.9f, g). Thus, *DROTI* helps maintain rice yield or yield-related trait under moderate drought conditions.

To distinguish the two different drought conditions, the original ‘drought field’ was changed to ‘severe drought field’, and sub-lethal drought conditions used for yield trails was named as ‘moderate drought’. The related results were added in the main

text in line 213-227, new Fig. 2m, n and Supplementary Fig. 9. The detailed methods for field trials were added in the Methods in line 587-600.

4) What is the haplotype of NT used in Fig. 5a, b, c, and d? Does the haplotype affect the expression of *DROT1* mediated by ERF3 and ERF71?

Response: The NT used in Fig.5 and the whole manuscript is non-transgenic plants isolated from heterozygous transgenic lines which used Nipponbare as the recipient of transformation. The haplotype of *DROT1* in NT is same as that of Nipponbare and Yuefu. We have added statement in the legend of Fig. 5.

Through transactivation activity analysis, we found that ERF3-mediated downregulation of *DROT1* expression was not affected by different haplotypes, while ERF71 activated the promoter of *DROT1* with a significantly stronger activating activity on the haplotype of IRAT109 than that of Yuefu (Fig.5h, i).

5) It is mentioned that ERF3 and ERF71 directly regulate the expression of *DROT1*. Transient expression assays do not determine whether an induced expression is directly or indirectly regulated. EMSA can provide evidence showing direct regulation, but it is an in vitro experiment system. I feel additional in vivo evidence is required to support this. Also, the EMSA data is not quantitative. Because the direct regulation by ERF3 and ERF71 is a key finding in this study, qualitative data along with statistical evaluation should be provided.

Response : Thank you for your valuable comments.

To demonstrate that ERF3 and ERF71 directly regulates *DROT1 in vivo*, we performed yeast one-hybrid and ChIP assays (as Reviewer 4 also suggested in the comment ‘4, Two regulators ERF3 and ERF71 regulate *DROT1* expression by binding its promoter. The EMSA result is not perfect, and yeast assay and ChIP should be performed’). In Y1H assay, both ERF3 and ERF71 could bind to *DROT1* promoters of IRAT109 and Yuefu (new Fig.5j). Although the transformants of empty B42AD vector together with *DROT1* promoter also shows weaker binding signal, it may be due to the activation of GCC box by yeast endogenous transcription factors. ChIP-qPCR revealed that both ERF3 and ERF71 could significantly enrich fragments containing GCC box in *DROT1* promoter (new Fig.5l, m). These results further indicate that ERF3 and ERF71 directly regulate the expression of *DROT1*.

We also improved the EMSA results by adding unlabeled oligonucleotides as competitors. The results further confirmed that both ERF3 and ERF71 could bind to the P1 site (new Fig.5n, o). In addition, we added the result of ERF71 binding to P2 and P3 fragments in new Supplementary Fig.15b, c. The original Fig.5l showing ERF3 binding to P2 and P3 fragments has been moved to new Supplementary Fig.15a.

Related results were added in the main text in line **359-366**, new Fig. 5j-o and Supplementary Fig.15a-c. The methods used for Y1H and ChIP assays were added in line **751-770**.

6) The major haplotype in tropical upland rice is Hap3. However, there are still several lowland rice accessions which possess Hap3. Are the expression levels of Hap3 in lowland rice lower than those in upland rice under drought?

Response : Thank you for your insightful comment.

As you mentioned, there are several lowland rice accessions with Hap3. According to your suggestion, we randomly selected 15 upland rice and 18 lowland rice accessions with Hap3 for expression analysis. Three-week-old seedlings were dehydrated for 2 hours and then sampled for RNA extraction and *DROT1* expression analysis. The result showed that *DROT1* in upland rice accessions had a significantly higher expression level than that in lowland rice under dehydrate stress (new Supplementary Fig.18).

We speculate that there may be three reasons why some lowland rice accessions also possess Hap3 genotype. **First**, it may be due to the differentiation of upstream regulatory genes which lead to the differential expression of *DROT1* between upland and lowland rice with Hap3 under drought stress. Although Hap3 is present in some lowland rice, due to the restriction of upstream regulators, the expression of *DROT1* cannot reach the same level as that in upland rice, which ultimately limit it to become upland rice. **Second**, as upland rice evolved on the basis of lowland rice, through long term breeding and selection, upland rice has accumulated a higher proportion of Hap3 than lowland rice. But Hap3 may also retain in lowland rice, because the contribution of one dominant haplotype or gene is limited to enhance drought resistance of the

whole population. We speculate that the differentiated regulators and specific haplotype of *DROT1* jointly determined the drought resistance of upland rice. **Third**, as suggested by Reviewer 2, geographical distribution of accessions with Hap3 were investigated. We found that most of the cultivars with Hap3 are distributed in Southeast Asian and are upland *tropical* or upland *indica* rice, while the rest of Hap3 accessions are lowland *indica* rice that mainly distribute in China (new Fig.7d). The accessions harboring Hap3 had two distinct ecotypes in different regions, indicating that the dispersal of this haplotype was closely related to the local environment. During the evolution of *indica* rice in Southeast Asia, Hap3 of *DROT1* was accumulated in upland rice of *indica* under arid or seasonal drought environment. However, in the evolution process of *indica* rice in China, accessions with Hap3 may have not undergone specific drought-adaptative selection, and eventually the *indica* rice in China was dominated by lowland rice.

We have added this result in the main text in line **417-424**.

7) In Supplementary Fig. 13, these expression data were not statistically evaluated. For example, *ERF71* in Yuefu may not be induced in a drought field relative to a paddy field. Also, the expression levels of *DROT1* are similar in paddy and drought fields. It is mentioned in lines 359-359 that this *DROT1* level is necessary to improve the resistance of rice under long-term (90-day) drought stress. How does non-induced *DROT1* confer drought tolerance in upland rice? Also, what is the developmental stage when rice was grown for 90 days in paddy and drought fields? Which data should we see to find *IRAT109* is still tolerant to drought after 90 days of drought treatment?

Response: Thank you for pointing this out.

We re-analyzed the expression data and statistically evaluated (new Supplementary Fig.10j-l). We found that the expression of *ERF71* was not induced in Yuefu under drought field compared to paddy field. However, its expression increased significantly in *IRAT109* and *IL349*, which is essential for the higher expression of *DROT1* in these two varieties and better drought resistance compared to Yuefu.

The expression levels of *DROT1* are similar in paddy field and drought field. We assume that as rice plants adapt to drought stress, *DROT1* expression may not be continuously induced, but should be maintained at a certain level to re-balance growth

and drought resistance. Importantly, under long term drought stress, the expression of *DRO1* in IRAT109 and IL349 was significantly higher than that in Yuefu, which is positively correlated with their stronger drought resistance. We apologize for the confusion caused by not detailing this point in the previous manuscript. We have revised it and added detailed description in line **387-398**.

Rice grown for 90 days is at booting stage. Here, we provide a photo showing the phenotype of IRAT109 and Yuefu after 90 days drought treatment. The growth performance of IRAT109 (left) is relative well with green and non-curly leaves, while Yuefu (right) shows a yellowish green and severe curly leaves. This indicates that IRAT109 is still tolerant to drought after 90 days growing under drought treatment. We could report this in the manuscript if the reviewer felt this was helpful, but have chosen not to do so at the moment in order to maintain focus on the key findings.

Growth performance of IRAT109 and Yuefu under drought condition for 90 days

8) The position of asterisks in Supplementary Fig. 6b should be fixed.

Response: Thank you for your reminder. We have revised it.

9) Fig. 1k: What tissue, developmental stage, growth conditions were used for analysis?

Response: We apologize for not describing this clearly in the manuscript.

RNA samples used for expression analysis in Fig. 1k were collected from leaf tissues of two-week-old seedlings grown under normal conditions. We have added a description to the legend of Fig. 1k.

Reviewer #2 (Remarks to the Author):

The authors first used GWAS and analysis of an ILs to identify three loci (i.e., qDR4a, qDR4b and qDR10b) that associated with two traits (DRI and LRI) and played important roles in drought resistance of japonica rice. They then identified a candidate gene (DROT1) that conferred drought resistance at qDR10b by multiple approaches including local LD, transcriptome and gene-based association analyses and validated the drought resistant function of DROT1 by transgenic experiments. They further demonstrated that DROT1 improved drought resistance by adjusting cell wall structure and was regulated directly by two drought responsive TFs (ERF3 and ERF71). Finally, they suggested that the elite haplotype of DROT1 in upland rice originated from wild rice and introgressed into tropical japonica from indica. Overall, this is very good study with substantial work behind. The results or findings are of significance in the isolation and functional investigations of drought-stress genes in rice or crops in general. The manuscript is well organized except for Discussion (see the comments below) and written clearly. Despite these, I have a few of concerns and uncertainties listed below.

Response: Thank you very much for your positive comments of this work.

1. I am confused with the section “Natural variation of DROT1 improves drought resistance in upland rice”. First, the descriptions on 8 haplotypes such as those associated with “unique”, “major”, “proportion” did not make much sense at all. No clearcut exists among groups. Second, I am curious about how the NJ tree (Fig. 6c) was generated. Did the authors used Hap1 as the outgroup? If yes, why choose Hap1? If no outgroup was used, the argument “... Hap3 evolved on the basis of other haplotypes, suggesting its late appearance in the rice ancestors” would be problematic. Third, it’s not appropriate to mention “the tropical japonica of Hap3 was grouped with indica, inferring it introgressed from indica” because Hap3 occurred in almost all groups, including japonica upland and lowland rice, indica upland and lowland rice (Fig. 6a). I believe that the key is to address the origin of Hap 3 (lineage/group and geographical area) and its dispersal process rather than to focus exclusively on “the Hap3 of tropical japonica”.

Response: Thank you for your valuable comments and good suggestions.

For the first comment, we agree with you that no clear-cut haplotype exists among groups. The previous description of Hap3 was inaccurate. It has been clarified that upland and lowland ecotypes in *japonica* subgroup are clearly differentiated in

genomic level, but it is difficult to distinguish them in *indica* (ref.44 of the manuscript). Therefore, it is understandable that *indica* of Hap3 consists both upland and lowland rice. Furthermore, it is generally believed that the drought resistance of rice is determined by multiple genes with minor-effects, so it is unlikely to distinguish all upland rice by *DROT1* alone. There may be other unknown drought-resistance genes also differentiated between upland and lowland rice, and the combined haplotypes of these drought-resistance genes may be able to better classify two rice ecotypes. We have now revised the inaccurate words in the manuscript. For example, ‘typical’ in previous line 337 was changed as ‘important’ in line 411. ‘major’ in line 371 of previous version and ‘unique’ in previous line 526 were removed due to the re-description of the results.

For the second comment, we used Hap1 as outgroup in previous study. The haplotype analysis showed that Hap1 is a unique type that is rarely distributed in common wild rice, the vast majority of wild rice are Hap5-8. Therefore, we initially considered that Hap1 to be an independent group compared to other groups. This is inappropriate. We reconsidered it and used *O.nivara* and *O.glaberrima* specific haplotypes as out groups to rebuild phylogenetic tree. We found that *O.nivara* has three haplotypes, namely Hap5, Hap6 and Hap-*nivara* (a *nivara* specific haplotype), while *O.glaberrima* had only one haplotype, named Hap-*Og*. As Reviewer 3 and Reviewer 4 suggested (*Reviewer 3-‘In constructing the phylogenetic tree of DROT1, the authors should use maximum likelihood, which will provide a more accurate tree than neighbor joining’; Reviewer 4-‘Phylogenetic analysis should be performed using the maximum likelihood method’.*), the maximum likelihood method, instead of neighbor joining method, was used in constructing phylogenetic tree. The result shows that Hap1 is still clustered in a single clade (new Fig. 7c). The origin of this haplotype is a mystery, we speculate that it may come from rare wild rice. Importantly, Hap3 clustered into a clade with Hap4 and Hap8 (two *Or-I* wild rice haplotypes), but not in the same clade with the outgroups. In contrast, Hap5 and Hap6 are closer to outgroups. This suggests that Hap3 may have evolved on the basis of other haplotypes and appeared later in wild rice. To further illustrate the evolution path of *DROT1*, the

haplotype network of *DROT1* was generated. It clearly shows that Hap3 evolved from those wild rice-specific haplotypes, suggesting its late appearance in the rice ancestors (new Supplementary Fig. 17a). We have added the results in new Fig. 7c and Supplementary Fig. 17a, and modified the sentences in line **414-417**.

For the third comment, we agree with you that we should address more on the origin of Hap3 and its dispersal process. To answer this question, we analyzed the lineage and geographical distribution of Hap3. For the convenience of analysis, we subdivided *japonica* rice into temperate *japonica* and tropical *japonica* (new Fig.7a). We found that Hap3 is mainly composed of *Or-I* wild rice, *indica* and tropical *japonica*. This indicates that Hap3 originated from *indica*-type wild rice. Most of the cultivars with Hap3 are distributed in Southeast Asia (SEA) and are upland tropical or upland *indica* rice, while the remaining Hap3 accessions are lowland *indica* rice from China (new Fig.7d). Based on this result, we speculate that Hap3 of *indica*-type wild rice was simultaneously introduced into *indica* of China and SEA. Why most accessions with Hap3 in SEA are upland rice, but those in China are lowland rice? It has been reported that drought resistant rice accessions are mainly distributed in SEA (20-24° N, 80-100° E. Preferred area: Bangladesh and India), which may be closely related to the local environments (Bin Rahman and Zhang Rice, 2018). During the evolution of *indica* rice in Southeast Asia, Hap3 was accumulated in upland rice of *indica* in arid or seasonal drought environment. However, in the evolution process of *indica* rice in China, accessions with Hap3 may not experience specific drought-adaptative selection, and ultimately the *indica* in China are mostly lowland rice. We have added these results in the main text in line **417-420** and new Fig. 7d, and revised the Fig. 7a.

In addition to the origin and spread of Hap3, we also interested in the evolution of Hap3 in upland rice. Since more than half of the cultivars with Hap3 are upland tropical *japonica*, further study of this subgroup is necessary. Phylogenetic analysis showed that tropical *japonica* with Hap3 was clustered with *indica*, contradicting with the fact that tropical *japonica* is genetically closer to temperate *japonica* than to *indica* (ref.46 and ref.47 of the manuscript). Further analysis confirmed that Hap3 of

tropical *japonica* was introgressed from *indica*. As two-thirds of accessions in tropical *japonica* population harbor Hap3, we speculate that this haplotype is accumulated in tropical *japonica* owing to its better drought resistance. What we should know is that most of upland *japonica* is tropical type. This is consistent with the result that upland rice with *DROT1*^{Hap3} is mainly tropical *japonica*, suggesting that there is a correlation between tropical *japonica* and *DROT1*^{Hap3}. This finding also helps us understand why the tropical *japonica* subpopulations are generally more resistant to drought stress. So, we think this section should be retained in the manuscript. We hope that this is now clear and acceptable.

Reference: Bin Rahman ANMR, Zhang J. Preferential Geographic Distribution Pattern of Abiotic Stress Tolerant Rice. *Rice*. 2018 Feb 8;11(1):10. doi: 10.1186/s12284-018-0202-9.

2. Given that the authors “hypothesize that Hap3 is a typical drought-resistant haplotype in upland rice” (p. 376-377), how to explain the presence of Hap3 in lowland rice? Or is the gene drought-resistant or not in lowland rice? Similarly, if “DROT1 may be an aerobic drought adaptation gene in upland ecotype rice” (p. 492), why not be the same outcome of this gene in lowland rice that has Hap3 too. In this sense, the section “DROT1 may be an aerobic drought adaptation gene in upland ecotype rice” in Discussion should be reconsidered.

Response: Thank you for your suggestion.

First, we would like to explain why Hap3 is present in lowland rice from the following three aspects.

1. Limited gene effects. The contribution of one dominant haplotype or gene is limited to enhancing drought resistance of the whole population. We found that lowland rice accessions with Hap3 are mainly *indica*. It has been clarified that the upland and lowland ecotypes of *indica* could not be distinguished from the genome (as reference 11 and 44 of the main text). Due to the complex genetic structure of drought resistance, it is difficult to classify upland and lowland ecological types by using only one gene with limited effect.

2. Differential expression of *DROT1*. As Reviewer 1 suggested in comment 6 (*The major haplotype in tropical upland rice is Hap3. However, there are still several lowland rice accessions which possess Hap3. Are the expression levels of Hap3 in*

lowland rice lower than those in upland rice under drought?), we performed expression analysis of *DROTI* in 15 upland rice and 18 lowland rice accessions with Hap3 under dehydrated stress. The result showed that *DROTI* in upland rice accessions had a significantly higher expression level than that in lowland rice under dehydrate stress (new Supplementary Fig.18). This indicates that the upstream regulation pathways of *DROTI* may have diversified between upland and lowland rice, which caused the differential expression of *DROTI* in upland and lowland rice accessions with Hap3 under drought stress. Therefore, although some lowland rice accessions harbor Hap3, their *DROTI* expression level is lower than that of upland rice, resulting in relatively weak drought resistance and not sufficient to become upland rice.

3. Geographic origin and drought adaptive selection. By investigating the geographical distribution of rice accessions with Hap3, we found that most of lowland rice are *indica* from China, while other *indica* are upland rice from SEA. Rice accessions with Hap3 had two distinct ecotypes in different regions, indicating the dispersal of this haplotype is closely related to the local environment. During the evolution of *indica* rice in Southeast Asia, Hap3 was accumulated in upland rice of *indica* in arid or seasonal drought environment. However, during the evolution process of *indica* rice in China, accessions with Hap3 may have not undergone specific drought-adaptative selection, and ultimately the *indica* rice in China is almost lowland rice.

The above analysis helps us understand why Hap3 is present in lowland rice. We have added related results in line 417-424. We also think the word ‘typical’ described in previous line 337 is inappropriate and have changed it to ‘important’ in line 411.

Second, through the induced expression analysis and transformation test of *DROTI* from lowland rice (OEY), we infer that *DROTI* also has a drought resistant function in lowland rice but its effect is weaker than that in upland rice.

Third, to assess whether *DROTI* is an aerobic drought adaptation gene of upland rice, we further discussed in terms of genetics, expression/regulation patterns and evolution. We believe that the most important feature of a gene adapting to drought is

the variation in the DNA that enables it to optimize its resistance function in that environment. Through gene-based association analysis and promoter activity analysis, we consider that the causal SNP s18975900 in the promoter confers higher expression of *DROTI* in upland rice and is essential for enhancing the expression of *DROTI* under drought stress. Genetic transformation analysis of overexpression and RNAi lines showed that the expression level was positively correlated with drought resistance. Therefore, this key SNP is the DNA basis for *DROTI* adapting to drought in upland rice. From the expression/regulation pattern, ERF3/ERF71 regulate the expression of *DROTI* by directly binding to its promoter, especially at the site (P1) containing the causal SNP. This nucleotide variation from C to T enables ERF71 to enhance the activation ability of *DROTI* promoter and positively regulate drought resistance. Furthermore, Hap3 is the only haplotype in the natural germplasm that contains this functional variation, and its proportion is much higher in upland rice than in lowland rice. Hence, we consider that *DROTI* may be an aerobic drought adaptation gene in upland ecotype rice. We reorganized this section and made detailed modifications in the Discussion as shown in line **496-525**.

3. In the third section of Discussion, the authors raised an interesting issue associated with the stress and growth trade-off that is worthwhile attention. However, the other three topics the authors chose seemed to me very superficial without concrete take-home message. In another word, the authors spent too much space to repeat what the Results have already mentioned. These sections should be shortened substantially, with more concise presentations and to the points. In another word, these sections should be reorganized with clear hypotheses raised and discussed, and otherwise might be removed entirely.

Response: Thank you for your valuable comments and objective suggestions. We have revised the Discussion section comprehensively and highlighted the main viewpoints. In brief, we mainly discuss the following issues: 1, Methodology of drought resistance gene research in rice. 2, How plants balance growth and stress trade-off; 3, Is *DROTI* a drought adaptation gene? The detailed modifications are shown in the Discussion section of the main text.

4. p. 150-152: Two cultivars were used in the expression analysis for both upland and

lowland rice. So, does Figure 1g. show the average of expression values?

Response: Yes. The average expression value of two upland rice were compared with that of two lowland rice. We have added this description to the legend of Fig. 1g.

5. In the legend of Supplementary Fig.1, two photos showing “leaf curling (upper right) and leaf color (lower right)” did not present clearly the features that were described in the text. Are the enlarged photos in which leaf curling and leaf color could be demonstrated clearly? Also, the authors should provide the percentage of variation that explained by PC1 and PC2 in the figure.

Response: Thank you for your good suggestion.

We have added enlarged photos in the Supplementary Fig.1a. The percentage of variation explained by PC1 and PC2 were added in the parentheses in the Supplementary Fig.1d.

Reviewer #3 (Remarks to the Author):

In this manuscript, the authors use GWAS in combination with genetics, molecular biology, and phenotypic and physiological characterization to isolate a COBRA homolog, DROUGHT1, as being responsible for drought resistance in upland rice. The fact that the authors isolate the exact SNP responsible for one of the loci identified in their GWAS analysis is impressive, and the fact that loss of DROT1 reduces drought tolerance whereas overexpression of the gene enhances drought tolerance, in combination with transgenic promoter-GUS analyses and phylogenetic tracing of the DROT1 gene, provides convincing evidence of the importance of this gene for adaptation to drought. However, I found the data reporting cell wall modifications in DROT1 mutant and overexpression lines to be incomplete, and the authors do not provide compelling mechanistic data that explain precisely how DROT1 functions at the molecular, cellular, or physiological levels. The COBRA family of proteins has long been enigmatic, and it is a shame that this work does not really elucidate their functions beyond what is known, at a mechanistic level at least. Specific comments follow.

The paper is poorly written, with a very high frequency of syntax and grammar errors – the errors are so numerous that I have not listed them in this review, but extensive language editing would be required for the manuscript to be suitable for publication. For example, cellulose microfibrils, which are part of the core message of the paper, are variously called “microfilaments” (which is used to refer to actin) and “microfibers” – neither of these terms is correct.

Response: We apologize for the poor writing of the manuscript. We have checked the paper carefully and corrected syntax and grammar errors. In particular,

'microfilaments' has been changed to 'microfibrils' in line 313, and 'microfibers' in the discussion part was removed.

In order to improve the English of our manuscript, we invited a professional science editor to polish the language of the paper. We hope this revised version will be easier for reading.

A major weakness of the paper is the cell wall analyses. In Figures 4 and S9, FTIR microscopy, for which peaks can be difficult to accurately assign, is used as the main analytical technique, but these data should be corroborated with immunolabeling, staining, and/or biochemical analysis of cell wall composition in the different lines. As the data stand now, they only allow rough speculation as to the mechanistic connection between loss/gain of DROT1 expression, changes in cell wall composition, and drought tolerance, as evinced by the authors in both their Results and Discussion sections dealing with the cell wall data. As they stand now, the data are too preliminary to draw firm conclusions.

Response: Thank you for your valuable comments. We agree that more accurate assays should be used for the analysis of cell wall components.

In the original manuscript, we adopted the newly invented FTIR combined with *in-situ* semi-quantitative method, which is based on the fact that it is relatively simple to simultaneously detect the content of multiple cell wall components in different tissues, with high throughput, relatively easy operation and low cost compared to immunolabeling. However, as you mentioned, the accuracy of this method is limited. At present, what we achieve is 1-2 cell size per pixel, which cannot reach the resolution at the cell wall level. Nonetheless, we consider that this is a relatively suitable approach, which helps us to preliminarily found target cell wall components in specific tissues that regulated by DROT1.

In order to accurately detect the content of the main components of cell wall, we also used high performance liquid chromatography (HPLC) to determine the content of cellulose, hemicellulose, and used chemical method to determine the content of lignin. The cellulose contents OEI and OEY lines were significantly higher than that of NT under drought condition, while there was no difference among these lines under normal growth condition. In contrast, the cellulose content was significantly decreased in *drot1-1* than that of IL349 under both drought and normal conditions

(new Fig. 4a, new Supplementary Fig. 11a). Except for cellulose, the content of hemi-cellulose was increased in both OE lines and knock-out lines in comparison with that of control lines under drought conditions (new Fig. 4b); the lignin content was increased only in OEY lines under drought conditions (new Fig. 4c). Under normal conditions, hemi-cellulose and lignin content were not changed in both OE lines and knock-out lines compared to control lines (new Supplementary Fig. 11b, c). Therefore, we suggest that DROT1 may specifically promote cellulose synthesis under drought stress, but not responsible for the synthesis of hemi-cellulose and lignin.

According to your suggestion, we have tried to use immunolabeling method to further demonstrate the change of cellulose content in vascular tissue, but we have not succeeded yet. Because it is indeed difficult for us, we apologize for not being able to provide any experimental results. However, combining the results obtained from the two different methods (HPLC and FTIR), we found that only the cellulose content is changed consistently with an increase in OE lines and decrease in knockout lines under drought conditions. Therefore, we speculate that DROT1 can affect the cellulose content in vascular bundles under drought stress, thereby regulating drought resistance.

We have revised the conclusions and discussions related to cell wall component in the manuscript so that it can be supported by experimental data. Related results have been added in line **270-309**, chemical methods for cell wall component examination have been added in line **683-698**. Additional one reference was cited in line **694** and added as ref.52

In constructing the phylogenetic tree of DROT1, the authors should use maximum likelihood, which will provide a more accurate tree than neighbor joining.

Response: Thank you for your professional advice.

We have reconstructed the phylogenetic tree by using maximum likelihood method. As Reviewer 2 mentioned in the first comment (*Second, I am curious about how the NJ tree (Fig. 6c) was generated. Did the authors used Hap1 as the outgroup? If yes, why choose Hap1? If no outgroup was used, the argument "... Hap3 evolved on the*

basis of other haplotypes, suggesting its late appearance in the rice ancestors” would be problematic.) that outgroups should be used properly, we take *O.nivara* and *O.glaberrima* specific haplotypes as out groups in the new phylogenetic tree. It showed that Hap3 clustered into a clade with Hap4 and Hap8 (two Or-I wild rice haplotypes), but not in the same clade with the outgroups. In contrast, Hap5 and Hap6 are closer to outgroups (new Fig. 7c). This suggests that Hap3 may have evolved on the basis of other haplotypes and appeared later in wild rice. We have revised the sentences in line **414-417**.

Figure 2 f, g, i, j: the y axes are confusingly labeled (relative to what? no explanation in the legend), and should all start at zero.

Response: We apologized for the inappropriate description in the legend.

The relative plant height and biomass were calculated by plant height and biomass of severe drought field divided by those under paddy field, respectively.

All of the y axes have been changed to start at zero.

We have revised the figures and added the detailed descriptions in the legend of Fig. 2

Subcellular localization in Figure 3: there is a strong potential of artifacts from using a double 35S promoter, and using the native promoter would be much better.

Response: Thank you for your valuable suggestion.

To improve it, we separately amplified the promoter and cDNA of *DROT1* from IRAT109 and combined. The combined fragments of *DROT1* were used to generate proDROT1::DROT1-GFP and proDROT1::DROT1-mCherry constructs, respectively. Then the constructs were transformed into Nipponbare, respectively. Seedling roots of positive transgenic lines were used for examination. The results are shown in new Fig. 3f, g.

Figure 3F: it is not clear that multiple cells are plasmolyzed in this image. Using 10% NaCl is not a standard method of plasmolysis, and could lead to cell death, causing artifacts; using 1 M sorbitol is preferable. If the protein is apoplasmic as depicted in Figure S15, tagging it with a pH-robust fluorescent protein such as mCherry would be preferable to GFP.

Response: We appreciate your point.

We have performed plasmolysis by using 1M sorbitol instead of 10% NaCl and found that plasmolysis could be seen obviously in multiple cells (new Fig. 3f, g). As you suggested, both DROT1-GFP and DROT1-mCherry fusion proteins were used for fluorescent examination. The results showed that clear signals of GFP and mCherry fluorescence can be observed in the cell walls. In addition, green fluorescence signals were also evident in the cell membrane of cells expressing DROT-GFP, while sporadic red fluorescence were also observed in the cytoplasm of cells expressing DROT1-mCherry. In Arabidopsis, COB (a homolog of DROT1) was associated with the Golgi and abundant in the cell wall, supporting its transit along a Golgi-derived vesicle secretion pathway (ref. 19 of the manuscript). It is possible that DROT1 protein translocation is blocked by the spatial barrier between cell membrane and cell wall after plasmolysis, and thus some of the DROT1 proteins are trapped in the cytoplasm. Anyhow, localized signals can be observed in the cell walls using two different fusion proteins. Therefore, we considered that DROT1 is primarily located at the periphery of the cells, especially in the cell wall. Related results have been added in new Fig. 3f, g and revised in line **264-269** and line **673-682**.

A few examples of writing errors:

L31: “jointing” should be “joining” or “combining”

Response: Thank you. We have revised it.

L33: should be “that confers...”

Response: Thank you. We have revised it.

L281: what are “cellulose monomers”? Do you mean UDP-glucose, the substrate for cellulose synthesis, and why would more of it be produced under drought?

Response: we apologized for not describing it exactly.

The so called ‘cellulose monomer’ is not UDP-glucose. In fact, what we wanted to describe is that the synthesis of **individual glucan chains** by CSC (which we incorrectly refer to cellulose monomers) in overexpression (OE) lines is increased under drought conditions. These glucan chains, as the basic units of cellulose microfibrils, are used to assemble cellulose on the cell wall under the participation of

DROT1. We speculate that more glucan chains would be produced under drought stress in OE lines. Because the expression of *CesA* genes was significantly higher in OEI and OEY lines than that of NT under drought stress (new Supplementary Fig.12). The induced expression of *CesA* may lead to an increase in the content of individual glucan chains, which then increases the assembly of cellulose due to more substrates facing more DROT1 in the OE lines, and finally more cellulose is synthesized in OE lines. However, under normal growth conditions, the amount of *CesA* complex is sufficient to synthesize glucan chains for cell growth and development, and excess DROT1 is not required for assembling process, so overexpression of *DROT1* does not increase cellulose content in OE lines. We can only make a simple inference at present, and more experiments are needed to confirm it in our future work.

We have changed “cellulose monomer” to “individual glucan chains” in line **311**.

L282: should be “cellulose microfibrils”, not microfilaments (actin)

Response: Thank you for your kind remind. We have revised it in line **313**.

Reviewer #4 (Remarks to the Author):

The manuscript “Natural variation of DROT1 confers drought adaptation in upland rice” by Sun, et al described a COBRA-like protein, DROT1, plays an important role in drought regulation. The SNP alteration in the promoter of DROT1, affects the expression level, thereby confers divergent drought resistance in upland rice and lowland rice, showing a potential value in breeding. In this paper, the results are interesting, but this story is a little bit simple and molecular evidences are weak. Therefore, I think that it can not be accepted on “Nature Communications” in current form.

Major Concerns:

1, The CRISPR experiment generally is very efficient for gene knock-out and create frameshift mutation. The paper mentioned only one line obtained showing a similar phenotype. It is a bit unusual.

Response: Thank you for pointing this out.

In our previous work, we performed the knock-out of *DROT1* in IL349 which shares a genomic background of Yuefu. Since the induction and differentiation of callus in Yuefu is very difficult, although we have tried several times to transform the CRISPR/Cas9 construct into Yuefu, but failed to obtain enough transgenic lines. In

the end, we obtained only one *DROT1* knockout mutant.

In order to fully demonstrate that knockout of *DROT1* can reduce drought resistance in rice, we purchased a *DROT1* mutant, *drot1-zh11* with background of Zhonghua11. A single-base insertion in the third exon caused frame shift of *DROT1* in *drot1-zh11* (new Supplementary Fig.4a). The *drot1-zh11* mutant had a significantly lower survival rate than that of ZH11 (new Supplementary Fig.4c, d). In addition, we also obtained *DROT1* knock-out mutants with Nipponbare background. One knockout mutant line with 288bp deletion in the third exon of *DROT1* named as *drot1-n* and was subjected for simulated drought treatment (new Fig.6c). The results showed that the survival rate of *drot1-n* was significantly lower than that of Nipponbare (new Fig.6d, e).

These results been added in line **179-181**, new Fig.6c and Supplementary Fig.4c, d.

2, the different expression level of DROT1 induces divergent drought resistance in upland rice and lowland rice, so I think that RNAi is more suitable than CAS9, or two experiments should be performed.

Response: Thank you for your valuable suggestion.

We constructed RNAi vector of *DROT1* and transferred it into Nipponbare. Two independent transgenic lines with significantly reduced expression of DROT1 were used for analysis (new Supplementary Fig.8a). After treated with 20% PEG for 5 days, and then re-watering for 10 days, the survival rate of RNAi lines was significantly lower than that of NT (new Supplementary Fig.8b, c). This further suggests that the expression level of *DROT1* is important for drought resistance of rice.

We have added this result in main text in line **206-212** and new Supplementary Fig.8a-c.

3, DROT1 confers drought adaptation by affecting cell wall properties including cellulose content and cellulose crystallinity. This finding is valuable, but How DROT1 regulate cell wall component, and its interaction genes should be explored.

Response: Thank you for your constructive comments.

DROT1 may regulate cellulose content and crystallinity through binding to cellulose microfibrils. BC1, a homolog of DROT1, also functions in cellulose

assembly through binding to cellulose microfibrils (ref.55 of the manuscript). Both BC1 and DROT1 have a carbohydrate-binding module (CBM) at their N-terminus, which specifically interacts with crystalline cellulose. The conserved protein sequence among COBRA family suggest that DROT1 may also have a common function as BC1.

Exploring interacting proteins of DROT1 is very important for elucidating its molecular mechanism in regulating cell wall component. Based on your suggestion, we screened a cDNA library of membrane proteins from rice leaf and panicle samples using DROT1 cDNA as bait, and identified some possible interacting proteins. The candidates include Os10g0562700 (REL2, controlling rolled and erect leaf), Os08g0440800 (OsALDH11A3, Rice Aldehyde Dehydrogenase), Os04g0455900 (ER-Golgi intermediate compartment protein), Os01g0116400 (protein kinase), Os10g0355800 (ATP synthase beta subunit) and other proteins with unknown functions. However, there is no clear prediction that these proteins are involved in cell wall formation. Some of the candidates may be involved in drought (OsALDH11A3, see the reference at the end), but that may be another new story that different from the main conclusion of this manuscript. We are sorry for not being able to provide any protein interaction results here. Since this is an interesting issue, we will carry out in depth in our future work.

Reference: Gao C, Han B. Evolutionary and expression study of the aldehyde dehydrogenase (ALDH) gene superfamily in rice (*Oryza sativa*). *Gene*. 2009 Feb 15;431(1-2):86-94.

4, Two regulators ERF3 and ERF71 regulate DROT1 expression by binding its promoter. The EMSA result is not perfect, and yeast assay and ChIP should be performed.

Response: Thank you for your comments.

We have now improved the EMSA results by adding unlabeled oligonucleotides as competitors. The results further confirmed that both ERF3 and ERF71 could bind to the P1 site (new Fig.5n, o). In addition, we added the results of ERF71 binding to P2 and P3 fragments in new Supplementary Fig.15b, c. The original Fig.5l showing

ERF3 binding to P2 and P3 fragments has been moved to Supplementary Fig.15a.

As you suggested, we further performed yeast one hybrid assay and CHIP qPCR analysis. In Y1H assay, both ERF3 and ERF71 could bind to *DROT1* promoters of IRAT109 and Yuefu (new Fig.5j). Although the transformants of empty B42AD vector together with *DROT1* promoter also shows weaker binding signal, it may be due to the activation of GCC box by yeast endogenous transcription factors. In CHIP assay, seedlings of transgenic plants containing 35S: ERF3-Flag and 35S:ERF71-Flag were collected, respectively. ERF3 and ERF71 proteins were purified by using anti-Flag antibody. CHIP quantification revealed that both ERF3 and ERF71 could significantly enrich *DROT1* fragments containing GCC box (new Fig.5l, m). These results further indicate that ERF3 and ERF71 directly regulate the expression of *DROT1*.

Related results have been added in the main text in line **359-366** and new Fig. 5j, l and m. The methods used for Y1H and CHIP assays were added in line **751-770**.

5, Genetic interactions are needed between *DROT1* and ERF3/ERF71.

Response: To test whether *ERF3* acts with *DROT1* in a common genetic pathway, we crossed *ERF3* overexpression lines with *DROT1* overexpression lines to generate OE-ERF3/OEI lines. By treated with 20% PEG for 5d and recovered for 7d, the OE-ERF3/OEI lines showed a similar survival rate to OEI (new Fig. 6a, b). This indicates that ERF3 and *DROT1* may function in a common pathway to regulate drought resistance.

To test whether *ERF71* acts with *DROT1* in a common genetic pathway, *erf71* and *drot1-n* mutants in the Nipponbare background using CRISPR-Cas9 system (new Fig. 6c). Then *drot1-n* was crossed with *erf71* to generate *erf71/drot1-n* double mutant. The survival rate of *erf71/drot1-n* double mutant was similar to *drot1-n* (new Fig. 6d, e), which indicated that ERF71 and *DROT1* may regulate drought resistance in a same genetic pathway.

These results have been added in line **368-378** of the main text and new Fig. 6.

Minor Concerns:

Phylogenetic analysis should be performed using the maximum likelihood method.

Response: Thank you for your professional suggestion.

We rebuilt a phylogenetic tree using maximum likelihood method. As Reviewer 2 mentioned in the first comment (*I am curious about how the NJ tree (Fig. 6c) was generated. Did the authors used Hap1 as the outgroup? If yes, why choose Hap1? If no outgroup was used, the argument "... Hap3 evolved on the basis of other haplotypes, suggesting its late appearance in the rice ancestors" would be problematic*) that outgroups should be used properly, so we took *O.nivara* and *O.glaberrima* specific haplotypes as out groups in the new phylogenetic tree as shown in new Fig.7c. It showed that Hap3 clustered into a clade with Hap4 and Hap8 (two Or-I wild rice haplotypes), but not in the same clade with the outgroups. In contrast, Hap5 and Hap6 are closer to outgroups. This suggests that Hap3 may have evolved on the basis of other haplotypes and appeared later in wild rice. We have revised the sentences in line **414-417**.

Reviewer #1 (Remarks to the Author):

The resubmitted manuscript contains new data and revisions, which have made this article better. However, I still have the following questions which should be addressed before consideration for publication in Nature Communications.

(1) Detailed analyses of transgenic lines with altered expression of *DROTI* demonstrated that *DROTI* is a drought tolerance regulator in rice. However, I feel that the data presented here are insufficient to conclude that the ‘natural variation’ of *DROTI* confers drought adaptation in upland rice. A major reason for this is that there are no tolerant haplotypes in rice. It may be possible that the expression level of *DROTI* is more critical for drought tolerance than its haplotype.

Response: We agree with you that the expression level of *DROTI* is critical for drought tolerance. However, based on the new results, we also conclude that the different expression level of *DROTI* may be attributed to the difference in haplotypes.

In another study, we performed transcriptome analysis for 175 rice accessions on seedlings grown under normal conditions for 30 days. The FPKM value of *DROTI* were compared among accessions with different haplotypes. The results showed that accessions with Hap3 had a significant higher expression of *DROTI* than that of Hap2 and Hap4 (Fig. R1a). It also showed that *DROTI*^T-type (Hap3) accessions had a significant higher expression of *DROTI* than that of *DROTI*^C-type (non Hap3) accessions (Fig. R1b). All these suggest that Hap3 is associated with higher expression of *DROTI*. However, we found that the expression level of *DROTI* in upland rice shows no difference with that of lowland rice under normal growth condition (Fig. R1c). (As the transcriptome data has not been published, we cannot present it in the paper, but would like to share with you in response).

Fig.R1

To further clarify the relationship between *DROTI* haplotypes and their expressions under drought conditions, we selected 144 rice accessions including 57 upland rice and 87 lowland rice for experiments. The germinated seeds were grown under nutrient soil in lab condition with normal water supply for 3 weeks, then seedlings were subjected to drought stress without water supply for 5 days. The leaf samples were collected for RNA extraction and gene expression analysis. We found that the expression level of *DROTI* was significantly higher in accessions with Hap3 than those with Hap2 and Hap4 (new Supplementary Fig.17a). It has been clarified that the causal SNP from C to T in the promoter of *DROTI* can significantly enhance its expression (Fig.3e). It showed that accessions with *DROTI*^T-type had significant higher expression of *DROTI* than that of *DROTI*^C-type accessions (new Fig.7d). Not like that under normal conditions, the expression of *DROTI* is significantly higher in upland rice than that in lowland rice under drought stress (new

Supplementary Fig.17b). All these indicate the expression of *DRO1* should be important for drought resistance in rice. It also implies that the expression of *DRO1* is diverse among haplotypes, and Hap3 enables *DRO1* higher expression and stronger drought resistance (higher proportion of upland rice in germplasm with Hap3).

Through expression analysis of germplasm, we also found that the expression of *DRO1* was significantly higher in upland rice with Hap3 than that in lowland rice with Hap3 (new Supplementary Fig.17c). The accessions with same haplotype of *DRO1* showed different expression levels, which reminding us that upstream regulators may diverse between upland and lowland rice. Therefore, we investigated the haplotypes of *ERF3* and *ERF71*. There are 6 haplotypes for *ERF3*, among which Hap1-2 are mainly *japonica* and Hap3-6 are *indica*. It seems that *ERF3* has no specific haplotype for upland rice (new Supplementary Fig.18a). *ERF71* has 10 haplotypes, among which Hap1/2/3/6 are *japonica*-specific haplotypes and Hap5/7 are *indica*-specific haplotypes (new Supplementary Fig.18b). We found that Hap6 of *ERF71* is mainly composed of upland rice, and Hap4 also has a high proportion of upland rice. For combined haplotype analysis of *DRO1-ERF3-ERF71*, 402 rice accessions were obtained based on the high-quality SNPs for all three genes. By classification, 12 main combined haplotypes were found, with a total of 335 accessions of which DEE¹⁻¹⁻⁶ and DEE³⁻¹⁻⁴ were upland rice specific haplotypes (Fig. 7e). The rest 67 accessions were rare types with less than 5 accessions and were not listed. Rice accessions with Hap3 of *DRO1* were further divided into four types: DEE³⁻¹⁻⁴, DEE³⁻¹⁻⁶, DEE³⁻³⁻⁵ and DEE³⁻⁵⁻⁵. The expression of *DRO1* showed no significant difference between upland and lowland rice accessions in each subgroup of Hap3 (new Supplementary Fig.17d-f). This indicates that the differential expression of *DRO1* between upland and lowland rice with Hap3 is caused by different haplotypes of *ERF* regulators. If upland and lowland rice shared the same haplotype of *ERFs* (*ERF3* and *ERF71*), there was no significant difference on *DRO1* expression level between them. Based on these results, we consider that the expression of *DRO1* confers drought resistance in rice, and its expressing level is determined by the combined haplotype of *DRO1* and *ERFs*.

To further determine whether HAP3 is a drought resistance haplotype, we compared the expressions of *DRO1* and survival rates among accessions with the same haplotypes of *ERF3/71* but different haplotypes of *DRO1*. In *japonica*, accessions with DEE¹⁻¹⁻⁴ and DEE³⁻¹⁻⁴ were used for analysis. It was found that accessions with DEE³⁻¹⁻⁴ had a significantly higher expression level and survival rate than that of DEE¹⁻¹⁻⁴ (Fig.7f, g). In *indica*, accessions with DEE²⁻³⁻⁵, DEE³⁻³⁻⁵ and DEE⁴⁻³⁻⁵ were used for analysis. We also found that *DRO1* expression in DEE³⁻³⁻⁵ type accessions was significantly higher than that in DEE²⁻³⁻⁵ and DEE⁴⁻³⁻⁵ type accessions. The survival rate of DEE³⁻³⁻⁵ type accessions was also significantly higher than those of DEE²⁻³⁻⁵ and DEE⁴⁻³⁻⁵ type accessions (Fig.7f, g). These indicate that Hap3 has a stronger promoter activity and drought resistance than the other haplotypes. Combining these results, we consider that Hap3 should be a potential drought resistance haplotype.

In conclusion, we speculate that the expression of *DRO1* is essential for drought resistance of upland rice, which can be enhanced by the promoter of Hap3 and determined by the combined haplotype of the *ERFs-DRO1* regulatory module.

Related results have been added in the main text in line 412-446, new Fig. 7d-g, and Supplementary Fig. 17, 18. The detailed methods for trails were added in the Methods in line 684-692 and 835-838.

(2) Expression levels of *DRO1* are compared among upland and lowland rice accessions in

Supplemental Figure 18. The expression levels of *DROTI* in many lowland rice accessions are higher than those in upland rice accessions. Is the expression level of *DROTI* positively correlated with the degree of drought tolerance among these accessions? If so, drought tolerance is determined by the expression level of *DROTI*, rather than its haplotype.

Response: As mentioned in response to your first review, we speculate that the expression of *DROTI* confers drought resistance of upland rice, which determined by the combined haplotype of the *ERFs-DROTI* regulatory module. In former Supplemental Fig. 18, the drought resistance of the accessions was not investigated, so we do not know whether the expression level of *DROTI* is positively correlated with the degree of drought tolerance among these accessions. In fact, this is a severe drought stress method, which can only reflect the stress response of gene expression under water-deficit conditions, but cannot truly reflect the expression changes and drought resistance of accessions under field-drought stress.

Compared to dehydration, the expression data under soil drought stress is more informative and help us understand the relationship between *DROTI* expression, haplotype and drought resistance of rice. As shown in the response to your first comment (the 4th paragraph), we concluded that the differential expression of *DROTI* between upland and lowland rice with Hap3 is caused by different haplotypes of *ERF* regulators. Inspired by this, we re-checked the combined haplotypes of the accessions used for dehydration stress and *DROTI* expression analysis as shown in former Supplementary Fig.18. We found that most upland rice belong to DEE³⁻¹⁻⁴ type, while lowland rice mainly consists of DEE³⁻¹⁻⁴ and DEE³⁻³⁻⁵ types accessions (Fig.R2). It also demonstrated that different expression level of *DROTI* between upland and lowland rice with Hap3 were associated with different haplotypes of *ERF* regulators. Since the results of dehydration are consistent with those of soil-drought stress in this revision, we think that the former Supplementary Fig.18 can be removed and used the results of soil-drought stress instead.

Upland	DROT1	ERF3	ERF71	Lowland	DROT1	ERF3	ERF71
IRIS 313-11004	3	1	4	CX77	3	1	1
IRIS 313-10710	3	1	4	B212	3	1	2
IRIS 313-10828	3	1	4	B188	3	1	4
IRIS 313-10953	3	1	4	B054	3	1	4
IRIS 313-10832	3	1	4	B190	3	1	4
IRIS 313-11424	3	1	4	B053	3	1	4
IRIS 313-8768	3	1	4	B233	3	3	5
IRIS 313-8694	3	1	4	B234	3	3	5
IRIS 313-11527	3	1	4	B033	3	3	5
CX220	3	1	4	CX162	3	3	5
CX304	3	3	4	CX20	3	3	5
IRIS 313-11528	3	3	5	CX120	3	3	5
B097	3	5	5	B176	3	3	5
B164	3	3	10	CX228	3	3	10
IRIS 313-12147	3	3	5	CX227	3	4	4
				B243	3	4	10
				B244	3	5	5
				B006	3	5	7

Fig.R2

(3) The data about the origin and spread of *DROTI* are insufficient to conclude the significance of Hap3 in drought tolerance in upland rice. I agree that *DROTI* is a drought tolerance regulator in rice, but it appears that its haplotype is not directly associated with drought tolerance. Instead of its haplotype, the factors regulating *DROTI*'s expression may characterize drought tolerance in upland rice. I believe that this part is critical for the quality and uniqueness of this study, but it will still need extensive work to make some conclusive statement regarding a determinant that distinguishes between upland and lowland rice.

Response: Thank you very much for your insightful comments. We think the response to your first review also answers the concerns you raised here. We provide the evidence that the haplotypes of

DROT1 and its regulators *ERF3* and *ERF71* is associated with divergent expression of *DROT1*. In conclusion, we speculate that the expression of *DROT1* is essential for drought resistance of upland rice, which can be enhanced by the promoter of Hap3 and determined by the combined haplotype of the *ERFs-DROT1* regulatory module.

Related results have been added in the main text in line 412-446.

(4) In Fig. 6b, the phenotype of OE-ERF3 has been rescued by crossing the transgenic line with OEI. The data demonstrate that ERF3 acts upstream of *DROT1*. However, the phenotype of *erf71* is not crossed with OEI in Fig. 6e.

Response: In fact, we have crossed *erf71* with the OEI lines, but due to bad weather, it delayed getting hybrid seeds. Currently, we do not have enough homozygous seeds for drought treatment, and thus cannot provide any experimental evidence. This will be further verified in our future work.

In the present results, the drought resistance phenotype of the *erf71/drot1-n* double mutant was similar to that of *drot1-n* mutant. This clarified that *ERF71* and *DROT1* may regulate drought resistance in the same genetic pathway.

Reviewer #2 (Remarks to the Author):

The authors have tried hard to address the points that all reviewers raised, including the major concerns and minor points in my review. The revised version is improved in numerous areas by adding new data and analyses as well as interpretations. Particularly, I appreciate very much the hard work and detailed explanations provided by the authors in responding to the major concerns and minor points by all reviewers. Despite these, I am still not satisfied with the revisions on the last section of Results, as I mentioned in my original reviewing report.

1. The new Supplementary Fig. 17a (the haplotype network) is very informative and more important than the Fig. 7c. The figure indicates that Hap3 originated from wild rice because the haplotype nearest to Hap3 consists mainly of wild rice. So, I agree that “Hap3 evolved from wild rice-specific haplotypes” (please indicate clear all the haplotypes in supplementary Fig. 17 not just Hap3!). However, the descriptions from lines 413-420 in revised text, alongside with the explanations in responding letter did not solve my concerns. In responding letter regarding origin of Hap3, the authors mentioned “We found that Hap3 is mainly composed of Or-I wild rice, indica and tropical japonica. This indicates that Hap3 originated from indica-type wild rice. . . ., we speculate that Hap3 of indica-type wild rice was simultaneously introduced into indica of China and SEA.” In fact, Or-II (japonica-type wild rice) also contained Hap3 (though low frequency) and so could not be ruled out as ancestral lineage. Moreover, it’s hard to understand why Hap3 was present in LR with pretty high frequency (36/148=24.3%) (Figs. 7a and 7b), as I curious in my original reviewing report.

Response: Thank you for your concern on this issue. As you suggested, we have labeled all the haplotypes in the haplotype network. Since this figure is more informative and valuable than the old Fig. 7c, we moved it to the main figures as new Fig.7i and old Fig.7c was changed as new Fig.7h.

To our knowledge, Or-II is not *japonica*-type wild rice, but an intermediate type between *indica*-type and *japonica*-type wild rice (Huang et al., 2012). Because the intermediate types of wild rice cannot be clearly distinguished as *japonica*- or *indica*-type wild rice from the genome, and the proportion of intermediate type wild rice is very low in Hap3, we speculate that Hap3 originated from Or-I wild rice.

To further explain why Hap3 was present in LR with “pretty high frequency (24.3%)”, we

provided new evidence and reorganized our interpretations. We would like to explain it from the following aspects:

1. The upland and lowland rice shows a much lower differentiation than that between *indica* and *japonica* with a relative clear differentiation in *japonica*, but not in *indica*. In previous study, 95 rice landraces including 13 upland- and 43 lowland-*indica* accessions and 12 upland- and 27 lowland-*japonica* accessions were clustered into two major groups, corresponding to the two subspecies *indica* and *japonica*. The upland-*japonica* were grouped in a single sub-clade. However, no distinct subclades were shown in the *indica* group, all upland-*indica* were mixed with lowland-*indica* (Fig.R3a). The *Fst* between upland and lowland rice (0.004 in *indica* and 0.085 in *japonica*) is substantially lower than that between two subspecies (0.528–0.617) (Wang et al., 2020). In another study, 84 upland and 82 irrigated accessions from all over the world were selected for phylogenetic analysis. It showed that all the upland *japonica* accessions clustered together, while upland *indica* scattered and mixed with lowland *indica* in the tree (Fig.R3b). The *Fst* value for the two ecotypes calculated based on genome-wide SNPs is only 0.06. Even in *japonica*, which the ecotype differentiation is clear-cut, the upland and irrigated ecotypes still have a limited genetic differentiation (*Fst* = 0.13), much lower than the reported differentiation between *indica* and *japonica* (*Fst* = 0.55) (Iyu et al., 2014). The substantially lower *Fst* between upland and lowland rice than that between *japonica* and *indica*, indicates the two ecotypes diversified after *japonica-indica* differentiation. In our study, 541 rice cultivars (including both upland and lowland rice) and 446 wild rice were selected to construct a phylogenetic tree using genomic SNPs. As shown in Fig.R3c, the purple strips in the inner circle represent upland rice, and the pink strips represent lowland rice. The results showed that upland and lowland *japonica* were clearly clustered into different branches, but both the two ecotypes were mixed in *indica*. The above results suggest that upland *japonica* likely have a single origin, but the formation and differentiation of upland and lowland ecotypes in *indica* underwent different environmental selection and genomic evolution from that in *japonica*.

Fig.R3

In terms of the haplotype of *DRO1*, Hap3 is a typical drought resistance haplotype in *japonica* with a higher proportion of upland rice (64/73=87.7%). However, in *indica*, due to the unclear differentiation between upland and lowland rice, half of the accessions with Hap3 are lowland rice which accounts for 75% of the lowland rice with Hap3. It also shows no distinct differentiation between upland and lowland ecotypes in *indica* with Hap3. Owing to the complex genomic

background of two ecotypes in *indica*, it is difficult to distinguish these lowland *indica* from upland ecotype.

2. Upland rice may be derived by the accumulation of multi drought resistance loci. Previous studies showed that, compared with lowland rice, upland rice may not have specific drought resistance or adaptation genes, but rather the accumulation of functional variations (or drought resistance haplotypes) of many drought resistance or drought adaptation genes (Zhao et al., 2018). The proportion of wild rice with Hap3 in *Oryza rufipogon* is 10.2% (26/254). In lowland rice population, the proportion of accessions with Hap3 is 12.4% (36/290), which is similar to that in wild rice population. However, in upland rice population, the proportion of accessions with Hap3 is 44.2% (88/199), which is much higher than that in lowland rice (new Fig.7c). In contrast, the proportion of other haplotypes did not increase greatly in upland rice population (Hap1: 41.4% in lowland and 33.2% in upland; Hap2: 15.5% in lowland and 2% in upland; Hap4: 28.3% in lowland and 14.1% in upland; Hap5: 2% in lowland and 5% in upland). This suggests that Hap3 is gradually accumulated in upland rice during evolution. In this sense, upland rice must contain higher proportion of drought resistance genes or haplotypes due to the “drought selection” in this process. Meanwhile, lowland rice may also contain a certain proportion of drought resistance genes or haplotypes which is inherited from wild rice. For Hap3 of *DROTI*, it originated from Or-I wild rice and firstly spread in *indica*, and then gradually accumulated in upland ecotype to a higher proportion during domestication. However, it may also retain in lowland rice because *DROTI* did not show adverse effects on rice growth and development under well-watered conditions.

As you mentioned in the previous review “*DROTI* may be an aerobic drought adaptation gene in upland ecotype rice, why not be the same outcome of this gene in lowland rice that has Hap3 too”. We know that the genetic mechanism of drought resistance in rice is very complex, which is determined by multiple minor-effect genes (Fukao et al., 2013). One or few drought resistance genes may not significantly improve the drought resistance of lowland rice, so it does not develop into typical upland rice in breeding. Perhaps due to the limited genetic effect of *DROTI* on drought resistance, not all accessions with Hap3 can greatly enhance drought resistance and be considered as upland rice, especially for germplasm with complex and diverse genomic backgrounds.

Taken these into account, we consider that it is acceptable for Hap3 presenting in LR with a frequency of 24.3%. At least, to some extent, it can reflect the advantages of this haplotype for higher expression of *DROTI* and better drought resistance. We sincerely hope you understand this. We have lowered the tone and made some changes in the main text in line 403-405 and line 444-446. Related figures have been changed as Fig.7b, c, h and i.

Ref:

- Huang X, et al. A map of rice genome variation reveals the origin of cultivated rice. *Nature*. 2012;490(7421):497-501.
- Wang, M. et al. Genomic landscape of parallel domestication of upland rice and its implications. *J. Syst. Evol.* 00, 1-11 (2020)
- Lyu J, et al. A genomic perspective on the important genetic mechanisms of upland adaptation of rice. *BMC Plant Biol.* 2014 Jun 11; 14:160. doi: 10.1186/1471-2229-14-160.
- Zhao, Y. et al. Loci and natural alleles underlying robust roots and adaptive domestication of upland ecotype rice in aerobic conditions. *PLoS Genet.* 14, e1007521 (2018).
- Fukao, T., and Xiong, L. Genetic mechanisms conferring adaptation to submergence and drought in rice: simple or complex? *Curr. Opin. Plant Biol.* 16, 196-204 (2013).

2. To explore the origin and spread of a haplotype or allele, haplotype network analysis is one of effective approaches. In this sense, the new Supplementary Fig. 17a (the haplotype network) is very informative and more important than the Fig. 7c. However, to look at the components of Hap3 is not enough because introgression of Hap3 from cultivated rice into wild rice cannot be ruled out. On the other hand, identification of the components of the wild haplotype nearest to Hap3 is critical. In this sense, I agree that “Hap3 evolved from wild rice-specific haplotypes” because the haplotype nearest to Hap3 consists mainly of wild rice in Supplementary Fig. 17a (please indicate clear all the haplotypes in supplementary Fig. 17 not just Hap3!). Nevertheless, all three wild lineages (Or-I, Or-II Or-III) are present in the haplotype nearest to Hap3 (Supplementary Fig. 17a) and Hap3 was found in all areas (Fig. 7d). In a word, where the Hap3 originated exactly remains unclear. The sentence “Phylogenetic analysis showed that the tropical japonica accessions with Hap3 were grouped with indica but not with temperate japonica, implying that this haplotype had introgressed from indica (Fig. 7e).” (Lines 424-426) is problematic. I would stress that the UR was scattered across the entire tree (Fig. 7e) and no evidence is available so far to support that Hap3 emerged in tropical japonica! (Figs. 7f, g were not relevant to this issue). To summarize, it’s not appropriate to say “The elite haplotype of *DROTI* in upland rice was introgressed into tropical japonica from indica” (Abstract). “this haplotype had introgressed from indica (Fig. 7e)” (lines 425-426).

Response: Thank you for raising these concerns. We think it is unlikely that Hap3 of wild rice was introgressed from cultivated rice. **1.** Gene flow and feralization from cultivar into wild rice should possibly occur in specific populations which are grown in the same region and with high allele frequency in cultivar. As reported, 21.6% of wild rice contain the non-shattering allele of *sh4*, and 26% of wild rice carrying the domesticated allele of *progl*. What we should know is that nearly all cultivars (95%) possess domesticated allele of these two genes due to artificial selection during domestication (Wang, et.al 2017). According to the results of *DROTI* haplotype analysis, we found that wild rice of Hap3 are mainly distributed in Southeast Asia (13 accessions), South Asia (11 accessions) and China (2 accessions). However, in Southeast Asia, there are 67 accessions with Hap3 in cultivated rice, accounting for 48.6% (67/138) of all cultivated rice in this region. While the proportion of wild rice with Hap3 in Southeast Asia is 12.5% (13/104). In South Asia, cultivated rice accessions with Hap3 accounts for only 19.2% (5/26) of all cultivars in this region. Moreover, the proportion of wild rice with Hap3 in South Asia was 10.6% (11/104), which was close to the proportion of cultivated rice with Hap3 in South Asia. It is less likely that Hap3 of cultivars from a lower frequency introgressed into its surrounding wild rice. In addition, if Hap3 was introgressed from cultivar into wild rice, it would have occurred in both Southeast Asia and South Asia, which is unlikely to happen in two distinct regions simultaneously. **2.** If Hap3 of wild rice introgressed from cultivar, where does the Hap3 in cultivated rice come from? The first exclusion is that Hap3 of cultivar did not directly evolved from the neighboring haplotype-Hap8, because this subgroup has no cultivar. According to these speculations, we consider that it is unlikely that Hap3 of wild rice was introgressed from cultivated rice. Since we do not have much direct evidence to prove this hypothesis, we have modified the sentence in the main text in line 461 as “we propose that *DROTI*^{Hap3} of *indica* **MIGHT** originate from *indica*-type wild rice”.

To elucidate where the Hap3 originated exactly, we investigate the original place of accessions with Hap3 and Hap8 in wild rice species. In our haplotype analysis, there are 26 wild rice of Hap3, including 11 from South Asia (42.3%), 13 from Southeast Asia (50%), and 2 from China (7.7%). We agree with you that identification of the components of the wild haplotype nearest to Hap3 is

critical. Through investigating the originated place of accessions with Hap8, we found that most of the accessions (94%) were wild rice, including 26 from Southeast Asia (56.5%), 11 from South Asia (23.9%) and 9 from China (19.6%). Moreover, 55.7% (68/122) of cultivars with Hap3 are from Southeast Asia, while 6.6% (8/122) from South Asia and 20.5% from China. The rest are from other regions around the world. Clearly, Hap3 is more densely distributed in Southeast Asia (new Fig.7j). Therefore, we speculate that Hap3 may originated from Southeast Asia. We have added this in the main text in line 451-454 and Fig.7j.

We also thanks for your attention on introgressions of Hap3 in tropical *japonica*. Whether Hap3 of tropical *japonica* is introgressed from *indica* only be inferred from phylogenetic analysis and allele frequencies. Cluster analysis showed that *DROTI* is a typical *indica-japonica* differentiation gene (As shown in new Supplementary Fig.19a, Or-I wild rice was closer to *indica*, Or-III was closer to *japonica*). As revealed by neighbor-joining tree based on 3,010 samples, tropical *japonica* was clustered in GJ group (Wang et al. 2018). Temperate *japonica* and tropical *japonica* rice share a common ancestor and diversified during a global cooling event about 4,200 years ago (Gutaker et al. 2020). Therefore, tropical *japonica* is genetically closer to temperate *japonica* than to *indica*. However, in this study, nearly two third of tropical *japonica* was clustered into a branch with *indica*. At the same time, because three quarters of upland rice with Hap3 are tropical *japonica*, so how Hap3 of tropical *japonica* evolved is an interesting issue to us. Calculating allele frequency is an effect way to test whether introgressions occurred between two populations. We know that the exact *indica*-specific or tropical *japonica*-specific SNPs are difficult to be identified. Based on our experience, we set the SNP frequency > 0.9 should be a specific SNP for a certain population in this analysis. However, in other study, it is considered that if a genotype was uniquely or highly detected in one ecotype with SNP frequency > 0.7, it was defined as an ecotype-specific or ecotype-preferential recombinant genotype (Xia et al., 2019). It is more stringent in our analysis. Nonetheless, it is clear that tropical *japonica* contains *indica*-specific alleles in this region. Through allele frequencies analysis for tropical *japonica* and *indica* of Hap3, we preliminary speculate its possible dispersal route from *indica* into tropical *japonica*. In order to ensure sufficient evidence for the main figures, we moved these results to Supplementary Fig.19a-c. According to your suggestion, we have deleted the sentence “was introgressed into tropical japonica from indica” (Abstract) and described it appropriately in the main text in line 456- 467.

Ref:

Wang H, Vieira FG, Crawford JE, Chu C, Nielsen R. Asian wild rice is a hybrid swarm with extensive gene flow and feralization from domesticated rice. *Genome Res.* 2017 Jun;27(6):1029-1038.

Wang et al. Genomic variation in 3,010 diverse accessions of Asian cultivated rice. *Nature.* 2018 May;557(7703):43-49.

Gutaker, R. et al. Genomic history and ecology of the geographic spread of rice. *Nat. Plants* 6, 492-502 (2020).

Xia, H. et al. Bi-directional selection in upland rice leads to its adaptive differentiation from lowland rice in drought resistance and productivity. *Mol. Plant* 12, 170-184 (2019).

3. In responding my second concern, the authors presented “three aspects”, which, as a matter of fact, all are sorts of speculations and questionable. Say, where is the evidence to prove “This indicates that the upstream regulation pathways of *DROTI* may have diversified between upland and lowland rice, which caused the differential expression of *DROTI* in upland and lowland rice

accessions with Hap3 under drought stress.” To my knowledge, these rebuttals or interpretation are largely speculations or pretty weak without solid evidence. In a word, I still do not understand why Hap3 was present or what function Hap3 has in lowland rice if it contributes to drought tolerance although the authors mentioned “The above analysis helps us understand why Hap3 is present in lowland rice.”

Response: Thank you again for your attentions on this issue. The response to your first review also explains the concerns you raised here “why Hap3 was present or what function Hap3 has in lowland rice if it contributes to drought tolerance”.

Specially, as you mentioned in this comment, to prove “upstream regulators may have diversified between upland and lowland rice”, we carried out additional experiments to provide new evidence for further explaining it. As Reviewer 1 also suggested, in order to further study the relationship between *DROTI* expression, haplotype and drought resistance of rice, we selected 144 rice accessions including 57 upland rice and 87 lowland rice for experiment. Seedlings were grown under nutrient soil in lab condition with normal water supply for 3 weeks, and then were subjected to drought stress without water supply for 5 days. Then leaf samples were collected for RNA extraction and expression analysis. We found the expression level of *DROTI* is significantly higher in accessions with Hap3 than those with Hap2 and Hap4 (new Supplementary Fig.17a). By dividing 144 rice germplasm into 54 *DROTI*^T-type (Hap3) and 90 *DROTI*^C-type accessions, it showed that *DROTI* expression was significantly higher in *DROTI*^T-type accessions than that of *DROTI*^C-type accessions (new Fig.7d). These suggest that Hap3 enables *DROTI* higher expression. Besides, *DROTI* had a significant higher expression in upland rice with Hap3 than in lowland rice with Hap3 (new Supplementary Fig.17c). The fact that same haplotype of *DROTI* showing different expression level of the gene reminds us that upstream regulators may diverse among upland and lowland rice.

Therefore, we investigated the haplotype of *ERF3* and *ERF71*. There are 6 haplotypes for *ERF3*, among which Hap1-2 are mainly *japonica* and Hap3-6 are *indica*. It seems that there is no distinct haplotype of upland rice (new Supplementary Fig.18a). *ERF71* has 10 haplotypes, among which Hap1/2/3/6 are *japonica*-specific haplotypes and Hap5/7 are *indica*-specific haplotypes (new Supplementary Fig.18b). We found that Hap6 of *ERF71* is mainly composed of upland rice, and Hap4 also has a high proportion of upland rice. For combined haplotype analysis of *DROTI*-*ERF3*-*ERF71*, 402 rice accessions were obtained based on high quality SNPs for all three genes. By classification, 12 main combined haplotypes were found, with a total of 335 accessions of which DEE¹⁻¹⁻⁶ and DEE³⁻¹⁻⁴ were upland specific haplotypes (Fig. 7e). Rice accessions with Hap3 of *DROTI* were further divided into four types: DEE³⁻¹⁻⁴, DEE³⁻¹⁻⁶, DEE³⁻³⁻⁵ and DEE³⁻⁵⁻⁵. The expression of *DROTI* showed no significant difference between upland and lowland rice accessions in each subgroup of Hap3 (new Supplementary Fig.17d-f). This indicates that the differential expression of *DROTI* between upland and lowland rice with Hap3 is caused by different haplotypes of *ERF* regulators. If upland and lowland rice shared the same haplotype of *ERFs* (*ERF3* and *ERF71*) regulators, there was no significant difference on *DROTI* expression level between them.

To further determine whether HAP3 is a drought resistance haplotype, we selected accessions with the same haplotypes of *ERF3/71* and compared the expressions of *DROTI* and survival rates among accessions with different haplotypes of *DROTI*. It was found that, in *japonica*, accessions with DEE³⁻¹⁻⁴ had a significantly higher expression level and survival rate than that of DEE¹⁻¹⁻⁴ type accessions (Fig.7f, g). In *indica*, the expression of *DROTI* and survival rate of DEE³⁻³⁻⁵ type accessions were significantly higher than that in DEE²⁻³⁻⁵ and DEE⁴⁻³⁻⁵ type accessions (Fig.7f, g).

These indicate that Hap3 has stronger promoter activity and drought resistance than the other haplotypes. Combining these results, we consider that Hap3 should be a potential drought resistance haplotype.

Revised results have been added in the main text in line 412-446, Fig.7d-g and Supplementary Fig. 17 and 18.

Reviewer #3 (Remarks to the Author):

The authors have addressed my concerns to a large degree, and I commend them for their careful revision of the manuscript.

Response: Thank you for your approval.

Reviewer #4 (Remarks to the Author):

In the new manuscript, the authors verified that *DRO1* is directly regulated by ERF3 and ERF71 through yeast one-hybrid, ChIP-qPCR, and EMSA experiments, and further confirmed that ERF3/ERF71 acts with *DRO1* in a common genetic pathway. Although the molecular mechanism between ERF3/ERF71 and *DRO1* has been supplemented, there are still some concerns based on the current results. My main concerns and uncertainties are listed below.

1. I am confused by the inconsistent results presented by the ChIP experiment and the EMSA experiment. ChIP analysis revealed that ERF3 enrich two of the three GCC box containing fragments (Fig. 5k-i), while EMSA analysis demonstrated that ERF3 could bind to the three GCC box-containing fragments in the *DRO1* promoter (Fig. 5n, and Supplementary Fig. 15a).

Response: Thank you for pointing this out. We believe the result is convincing as the ChIP assay for ERF3-Flag has been done at least three times. ChIP analysis revealed that ERF3 could not enrich the F1 fragments, while EMSA analysis indicated that ERF3 could also bind to P3 site. This inconsistent result may be due to the different conditions of binding activity between *in vivo* and *in vitro*. Similar results were found in other published papers. For example, in the manuscript entitled “KIRA1 and ORESARA1 terminate flower receptivity by promoting cell death in the stigma of *Arabidopsis*”, the authors described “Both ORE1 and KIR1 could physically interact *in vitro* with promoter fragments of RNS3, BFN1 and EX11” according to EMSA results (Fig.R4a). However, in the ChIP assay (Fig.R4b, c), it mentioned “the immunoprecipitated ORE1-GFP and KIR1-GFP proteins were enriched with promoter fragments of BFN1 and RNS3, but not of EX11” (Gao, et al., 2018). It shows inconsistent results by the ChIP and the EMSA for ORE1 and KIR1 interacting with the promoter of *EX11*.

Fig.R4

Take another example, *SNB* controls seed shattering through direct regulation of *qSH1* and *SH5* (Jiang et. al, 2019). The results showed that *SNB*-GFP enriched Q4 of *qSH1* and H1 of *SH5* more than 3-fold in ChIP (Fig. R5H, I). However, the His-*SNB* recombinant protein was unable to bind to these two sites in EMSA (see below of Fig. R5). It also presented inconsistent results by ChIP assay and EMSA.

Fig.R5

Nevertheless, we ensured that *ERF3* directly binds to the F2 and F3 (contains the causal SNP s18975900) sites through combined analysis of ChIP and EMSA results. Therefore, we think that the evidence of *ERF3* directly regulates *DRO1* is solid.

Ref:

Gao, Z., Daneva, A., Salaneka, Y. et al. *KIRA1* and *ORESARA1* terminate flower receptivity by promoting cell death in the stigma of *Arabidopsis*. *Nature Plants* 4, 365–375 (2018).

Jiang L, Ma X, Zhao S, Tang Y, Liu F, Gu P, Fu Y, Zhu Z, Cai H, Sun C, Tan L. The APETALA2-Like Transcription Factor SUPERNUMERARY BRACT Controls Rice Seed Shattering and Seed Size. *Plant Cell*. 2019 Jan;31(1):17-36.

2. Given that the authors “*DROT1* is directly repressed by *ERF3* and activated by *ERF71*” (p. 37-38), how to explain the OE-*ERF3*/OEI lines showed a similar survival rate to OEI (new Fig. 6a, b). To more rigorously verify whether *ERF3* and *DROT1* function in a common genetic pathway, *erf3* and *erf3/DROT1* double mutants should be generated.

Response: The OE-*ERF3*/OEI lines showed a similar survival rate to OEI, indicating that *ERF3* and *DROT1* may function in a common pathway to regulate drought resistance. Because *DROT1* was overexpressed in the OE-*ERF3*/OEI lines, the overexpressed *ERF3* could only repress the native expression of *DROT1* (*DROT1* of Nipponbare haplotype), but has no effects on the transgene expression of *DROT1* (*DROT1* of IRAT109 haplotype). Thus, in the OE-*ERF3*/OEI lines, the expression of *DROT1* maintained at a high level as that in the OEI lines, which resulted in a similar survival rate between them. Conversely, if *ERF3* and *DROT1* do not function in a common pathway to regulate drought resistance, the survival rate of the OE-*ERF3*/OEI lines should be significantly lower than that of OEI lines.

We agree with you that double mutant should be more critical for validating the genetic pathway between *ERF3* and *DROT1*. We successfully obtained an *erf3* mutant with a 265 bp deletion in the exon (Fig. R6a). The *erf3* mutant was further crossed with *drot1-n1*, which deleted 289 bp in the third exon of *DROT1*, to generate *erf3/drot1* double mutant. However, to our surprise, *erf3* did not show significantly enhanced drought resistance than NT (Fig. R6b, c). Although *erf3/drot1-n1* showed significantly decreased drought resistance than NT and *erf3*, we can not determine whether *ERF3* and *DROT1* function in the same genetic pathway. We hypothesized that there may be functional redundancy of *ERF3* for drought resistance, with homologous genes compensating for the phenotypic changes caused by the loss of function of *ERF3*. Based on the phylogenetic analysis of 18 cloned rice drought-resistant ERF genes, we found that *ERF3* and *AP2-39* clustered into one branch (Fig. R7). It has been reported that *AP2-39* also negatively regulates drought resistance of rice (Wan et al., 2011). Therefore, we speculate that *AP2-39* may compensate for the loss function of *ERF3*. We have not added these results in this revised manuscript, as further studies are need in the future to elucidate the genetic relationship between *DROT1*, *ERF3* and *AP2-39*.

Fig.R6

Fig.R7

Ref:

Wan, L. et al. Transcriptional activation of OsDERF1 in OsERF3 and OsAP2-39 negatively modulates ethylene synthesis and drought tolerance in rice. *PLoS one* 6, e25216 (2011).

Reviewers' Comments:

Reviewer #1:

Remarks to the Author:

The resubmitted manuscript has new data and descriptions to address the points raised by reviewers. Haplotype analysis of ERF3 and ERF71, along with expression analysis of DROT1, is interesting. However, I am still not convinced that particular haplotype combinations of DROT1 and their upstream regulators are associated with drought tolerance in rice. I agree that DROT1 is a key gene regulating drought tolerance in rice. However, the degree of drought tolerance in rice may not be determined by DROT1 and its upstream haplotypes. My specific comments are as follows.

(1) Fig. 7 and Supplemental Fig. 17 provide the haplotype data of DROT1, ERF3, and ERF71 and expression data of DROT1 in accessions with the same haplotype of ERF3 and ERF71. These data are interesting, but the drought tolerance data is limited relative to the expression data. I believe that the research question to be addressed here is whether rice accessions with Hap3 are significantly different from those with other haplotypes (Haps 1, 2, 4-8) in terms of drought tolerance. Due to the deficiency of drought tolerance data, it is unclear whether particular combinations of DROT1, ERF3, and ERF71 haplotypes are associated with drought tolerance.

(2) In Supplemental Figure 17a, Hap3 expression is not significantly different from Hap1 expression. Is drought tolerance of rice accessions with Hap3 vs. Hap1 not significantly different?

(3) In Supplemental Figure 17d-f, there is no difference in DROT1 expression between upland and lowland rice. If upland rice genotypes are generally more tolerant to drought than lowland genotypes, it seems that different drought tolerance between upland and lowland rice is not explained by the degree of DROT1 Hap3 expression.

(4) Is the degree of DROT1 expression positively correlated with the degree of drought tolerance regardless of its haplotype? Is it possible to see this correlation among rice accessions with Hap3?

(5) In the legend of Fig. 7, DROTT and DROTC must be DROT1T and DROT1C. Also, what DROT1T and DROT1C are must be explained in the legend.

Reviewer #2:

Remarks to the Author:

First of all, I would appreciate authors' hard work to improve the manuscript by addressing all the concerns and criticisms from all reviewers. As for the three points raised in my report, the authors presented a long and detailed explanation. The new experimental data and analysis by responding to reviewer 1 are also useful to address my point 3. However, honestly, the responses to my first two points do not solve my concerns entirely. Briefly, the authors responded to my first point by claiming "the genetic mechanism of drought resistance in rice is very complex, which is determined by multiple minor-effect genes". I cannot say this explanation is incorrect but again sorts of speculation. For point 2, several important results/conclusions arose from the comparisons of percentages (different types), which is not sufficient because the percentage difference depends on the sampling strategy or components of different samples in the dataset and might be misleading without statistical testing. Anyway, I suggest authors be careful to make decisive conclusions at this stage and to leave some issues open. Also, it is better for authors to raise/discuss the caveats or limitations of current analyses relevant in this section.

Reviewer #4:

Remarks to the Author:

The authors have addressed all my concerns to a large degree.

Reviewer #1 (Remarks to the Author):

The resubmitted manuscript has new data and descriptions to address the points raised by reviewers. Haplotype analysis of *ERF3* and *ERF71*, along with expression analysis of *DROT1*, is interesting. However, I am still not convinced that particular haplotype combinations of *DROT1* and their upstream regulators are associated with drought tolerance in rice. I agree that *DROT1* is a key gene regulating drought tolerance in rice. However, the degree of drought tolerance in rice may not be determined by *DROT1* and its upstream haplotypes.

Response: Thank you for your comments. By analyzing the combined haplotypes of *DROT1-ERFs*, we found that *DEE*³⁻¹⁻⁴ and *DEE*¹⁻¹⁻⁶ are mainly composed of upland rice. Thus, these two haplotype combinations may be associated with drought resistance in rice.

Through expression analysis of *DROT1* in germplasm, we concluded that the expression level of *DROT1* was different between Hap3 and other haplotypes, especially in the accessions with the same haplotypes of *ERF3* and *ERF71* (Fig.7f and Supplementary Fig.17g). Besides, the degree of drought resistance (in term of survival rate) also showed difference between Hap3 and other haplotypes (Fig.7g and Supplementary Fig.17h). However, we were not able to determine whether the expression level of *DROT1* was positively correlated with the degree of drought resistance. Therefore, this paper only shows that the expression of *DROT1* may be determined by the haplotypes of *DROT1* and its upstream regulators, and Hap3 may be a potential drought resistance haplotype. But it is insufficient to conclude that the degree of drought resistance in rice is determined by these combined haplotypes, just as you commented in this review. We have revised in line 444-450 and further discussed in line 556-565.

My specific comments are as follows.

(1) Fig. 7 and Supplemental Fig. 17 provide the haplotype data of *DROT1*, *ERF3*, and *ERF71* and expression data of *DROT1* in accessions with the same haplotype of *ERF3* and *ERF71*. These data are interesting, but the drought tolerance data is limited relative to the expression data. I believe that the research question to be addressed here is whether rice accessions with Hap3 are significantly different from those with other haplotypes (Haps 1, 2, 4-8) in terms of drought tolerance. Due to the deficiency of drought tolerance data, it is unclear whether particular combinations of *DROT1*, *ERF3*, and *ERF71* haplotypes are associated with drought tolerance.

Response: Thank you for your valuable suggestions. We agree with you that the most concern is whether rice accessions with Hap3 are significantly different from those with other haplotypes in terms of drought tolerance. The drought tolerance data is limited relative to the expression data. Because it's indeed a huge amount of work to investigate the survival rate of all 144 rice accessions. To ensure the availability of the primary data and the maximum workload that can be undertaken simultaneously, we evaluated drought resistance in 74 out of the 144 accessions selected based on the combined haplotypes of *ERFs* and *DROT1*. These 74 accessions were previously used to investigate whether rice accessions with Hap3 are significantly different from those with other haplotypes in terms of *DROT1* expression and drought resistance under the same haplotype of upstream regulators (Fig.7f, g). Regardless of the upstream gene haplotype, it contains 11 accessions with Hap1 of *DROT1*, 8 accessions with Hap2, 33 accessions with Hap3 and 22 accessions with Hap4. By comparison, we found that accessions with Hap3 had a significant higher expression of

DROT1 and higher survival rate than those with other haplotypes (new Supplementary Fig.17g, h). By dividing 74 accessions into *DROT1^C* and *DROT1^T* types, we also found that the expression of *DROT1* and survival rate of *DROT1^T* type accessions were significantly higher than that of *DROT1^C* type accessions (Fig.R1). These results demonstrate that the drought resistance of accessions with Hap3 is significantly different from those with other haplotypes.

Through the haplotype combination analysis of *DROT1-ERF3-ERF71*, we found that *DEE³⁻¹⁴* and *DEE¹⁻¹⁻⁶* should be specific haplotypes for drought resistance, since 90% of accessions in these two groups are upland rice. Combining the expression of *DROT1* and survival rate of accessions with *DEE³⁻¹⁴* and *DEE¹⁻¹⁻⁴*, we speculate that *DEE³⁻¹⁴* is an important combinational haplotype associated with drought resistance. In addition, there are relative few accessions with *DEE¹⁻¹⁻⁶* in rice germplasm, and the preserved seeds are also limited for experiments. So, we could not confirm the drought resistance of accessions with *DEE¹⁻¹⁻⁶* in this study. It will be further investigated in our future work.

Fig.R1

(2) In Supplemental Figure 17a, Hap3 expression is not significantly different from Hap1 expression. Is drought tolerance of rice accessions with Hap3 vs. Hap1 not significantly different? Response: The expression of *DROT1* in accessions with Hap3 is not significantly different from those with Hap1. This may be affected by the inconsistent haplotypes of upstream genes-*ERF3* and *ERF71* (Fig.R2). Because we did not investigate the drought resistance of all accession with Hap1 and Hap3 whose expressions were already analyzed, it is not known for us whether there is a significant difference between accessions with Hap1 and Hap3 regardless of the haplotype of upstream regulators. However, through investigate the drought resistance of 74 out of 144 accessions as shown in Supplementary Fig.17g and 17h, accessions with Hap3 had a significant higher expression of *DROT1* and higher survival rate than those with other haplotypes. This suggests that, in the case of relatively consistent haplotypes of *ERF3* and *ERF71*, the *DROT1* expression and drought resistance of accessions with Hap3 are significantly different from those with other haplotypes.

	DROT1	ERF3	ERF71	No. of accessions	Average expression level
Hap1	1	1	1	19	0.225
	1	1	4	11	0.125
	1	1	6	8	0.249
Hap3	3	1	4	21	0.354
	3	3	5	12	0.373
	3	5	5	12	0.142
	3	1	6	9	0.158

Fig.R2

(3) In Supplemental Figure 17d-f, there is no difference in *DROT1* expression between upland and lowland rice. If upland rice genotypes are generally more tolerant to drought than lowland genotypes, it seems that different drought tolerance between upland and lowland rice is not explained by the degree of *DROT1* Hap3 expression.

Response: We agree with you that if there is a difference in drought resistance between upland and lowland rice with same haplotypes of *DROT1-ERF3-ERF71* (as shown in Supplemental Fig.17d-f), it may be associated with other drought resistance genes, rather than the degree of *DROT1*^{Hap3} expression.

(4) Is the degree of *DROT1* expression positively correlated with the degree of drought tolerance regardless of its haplotype? Is it possible to see this correlation among rice accessions with Hap3?

Response: Thank you for your comments. We found no significantly positive correlation between *DROT1* expression and the degree of drought resistance, regardless of its haplotype or among accessions with Hap3. Due to the complexity of genomic background of germplasm and the difficulty of drought resistance evaluation, it is hard to determine the correlation between *DROT1* expression level and the degree of drought resistance. The most direct assessment of drought resistance of rice is the use of upland and lowland rice information classified from agricultural production, although it ignores some intermediate types between drought resistance and drought sensitivity. To evaluate the degree of drought resistance in germplasm, the survival rate of seedlings was widely adopted. However, it is just one of the indicators for evaluating drought resistance, and does not fully reflect the drought resistance of rice. The relative biomass or plant height by comparing the rice plants grown under drought field to those in paddy field should be also important indicators to evaluate the degree of drought resistance. However, it takes a long time and a lot of work to complete. On the other hand, even if all accessions were investigated for drought resistance, there would most likely be no significant correlation between expression level of *DROT1* and the degree of drought resistance, because the genetic background of germplasm is too different and the trait of drought resistance is too complicated. As an alternative, near-isogenic or other lines with similar genetic background may be ideal materials for studying the function of gene haplotypes on drought resistance in rice and other crops.

In this manuscript, it is verified that Hap3 could increase the expression of *DROT1* which may be affected by upstream ERF regulators. In addition, through genetic studies on transgenic lines, we have clarified that the expression level of *DROT1* is important for drought resistance of rice. It also showed that Hap3 is the predominant haplotype of upland rice. All these suggest that Hap3 might be a drought resistance haplotype. Whether the expression of *DROT1* is correlated with the degree of drought resistance in rice germplasm is still an open question, and needs to be further clarified. We sincerely hope you can understand the research difficulties on evaluating the degree of drought resistance in rice germplasm, and our efforts to address these issues.

(5) In the legend of Fig. 7, $DROT^T$ and $DROT^C$ must be $DROT1^T$ and $DROT1^C$. Also, what $DROT1^T$ and $DROT1^C$ are must be explained in the legend.

Response: Thank you very much for pointing this out. We have revised and explained it in the figure legend.

Reviewer #2 (Remarks to the Author):

First of all, I would appreciate authors' hard work to improve the manuscript by addressing all the concerns and criticisms from all reviewers. As for the three points raised in my report, the authors presented a long and detailed explanation. The new experimental data and analysis by responding to reviewer 1 are also useful to address my point 3. However, honestly, the responses to my first two points do not solve my concerns entirely. Briefly, the authors responded to my first point by claiming "the genetic mechanism of drought resistance in rice is very complex, which is determined by multiple minor-effect genes". I cannot say this explanation is incorrect but again sorts of speculation. For point 2, several important results/conclusions arose from the comparisons of percentages (different types), which is not sufficient because the percentage difference depends on the sampling strategy or components of different samples in the dataset and might be misleading without statistical testing. Anyway, I suggest authors be careful to make decisive conclusions at this stage and to leave some issues open. Also, it is better for authors to raise/discuss the caveats or limitations of current analyses relevant in this section.

Response: Thank you very much for your valuable comments and constructive suggestions. In response to point 2, we made conclusions by comparing the percentages of different haplotypes. We agree with you that this depends on sampling from germplasm. What is convincing is that this analysis was based on the dataset that used for haplotype analysis as shown in Fig.7a. The accessions in this dataset were randomly selected from a core collection of rice germplasm, and only those with missing SNPs in *DROT1* were removed. Therefore, these results also show relatively accurate information to us.

Based on your suggestions, we cautiously draw conclusions on this section by lowering the tone. Besides, we discussed the limitations of the analysis relevant to this section and raised some issues, as shown in line 556-565.

Reviewer #4 (Remarks to the Author):

The authors have addressed all my concerns to a large degree.

Response: Thank you very much.

Reviewer #1 (Remarks to the Author):

The revised manuscript describes the limitations of haplotype and evolutionary analyses and interprets the data obtained from these analyses appropriately. I believe the latest version has addressed my major concerns regarding data interpretation.

Response: Thank you for your comment.

Reviewer #2 (Remarks to the Author):

The authors have positively responded to my concerns and made appropriate revisions/additions accordingly. Meanwhile, I noticed the comments from reviewer 1, which were actually my own concerns too and should be addressed appropriately. Well, I am not expert in this area and unable to make reviewer 1 hope the authors to (well, should be judged by reviewer 1).

Response: Thank you for your comments.

Reviewers' Comments:

Reviewer #1:

Remarks to the Author:

The revised manuscript describes the limitations of haplotype and evolutionary analyses and interprets the data obtained from these analyses appropriately. I believe the latest version has addressed my major concerns regarding data interpretation.

Reviewer #2:

Remarks to the Author:

The authors have positively responded to my concerns and made appropriate revisions/additions accordingly. Meanwhile, I noticed the comments from reviewer 1, which were actually my own concerns too and should be addressed appropriately. Well, I am not expert in this area and unable to make reviewer 1 hope the authors to (well, should be judged by reviewer 1).